# Multiomic profiling of medulloblastoma reveals subtype-specific targetable alterations at the proteome and N-glycan level

Medulloblastomas (MBs) are malignant pediatric brain tumors that are molecularly and clinically heterogenous. The application of omics technologies—mainly studying nucleic acids—has significantly improved MB classification and stratification, but treatment options are still unsatisfactory. The proteome and their N-glycans hold the potential to discover clinically relevant phenotypes and targetable pathways. We compile a harmonized proteome dataset of 167 MBs and integrate findings with DNA methylome, transcriptome and N-glycome data. We show six proteome MB subtypes, that can be assigned to two main molecular programs: transcription/translation (pSHHt, pWNT and pG3myc), and synapses/immunological processes (pSHHs, pG3 and pG4). Multiomic analysis reveals different conservation levels of proteome features across MB subtypes at the DNA methylome level. Aggressive pGroup3myc MBs and favorable pWNT MBs are most similar in cluster hierarchies concerning overall proteome patterns but show different protein abundances of the vincristine resistance-associated multiprotein complex TriC/CCT and of N-glycan turnover-associated factors. The N-glycome reflects proteome subtypes and complex-bisecting N-glycans characterize pGroup3myc tumors. Our results shed light on targetable alterations in MB and set a foundation for potential immunotherapies targeting glycan structures.

Medulloblastomas (MBs) are aggressive pediatric brain tumors that are histomorphologically, molecularly and clinically heterogenous[1]. Four main consensus subgroups have been described: WNT pathway activated MB (WNT MB), Sonic hedgehog pathway activated MB (SHH MB), Group 3 (G3) and Group 4 (G4) MB[2]. Molecular analyses, mainly using gene expression profiling, next generation sequencing and DNA methylation analysis predict further subdivisions with distinct clinical features[3–6]. Exemplary markers for poor survival comprise anaplastic histology, *MYC* amplification status, methylation subtype II/III, or *TP53* mutations in WNT and SHH MB[7–12]. Conversely, methylation subtype VII, extensive nodularity (MBEN histology), a distinct whole chromosomal alteration signature in non-WNT/non-SHH MB and WNT activation (e.g., nuclear accumulation of β-CATENIN or CTNNB1 mutations) were associated with a favorable prognosis in MB patients[12–14]. The clinical association between certain methylation subtypes and chromosomal aberrations has been clearly described, however, the underlying molecular mechanisms remain to be resolved and targeted treatment options are lacking. In contrast to nucleic acids, the proteome reflects a tumor's phenotype in a more direct way and holds the potential to precisely dissect clinically relevant phenotypes and targetable alterations. Studies on small MB cohorts, using fresh-frozen (FF) tumor material, have shown that MBs display

✉e-mail: ju.neumann@uke.de

heterogeneity at the proteome level[15–17]. Formalin-fixed-paraffin-embedded (FFPE) material, enables the generation of larger datasets which is essential to deal with the heterogeneity but provides challenges to proteome analysis[18]. In addition to protein abundance, post-translational modifications (PTM) of proteins are important to understand cell physiology and disease-related signaling[15–17]. The most complex and common PTM, N-glycosylation, has not been targeted in MB yet. Changes in the N-glycome are considered potential hallmarks of cancer and N-glycan structures hold strong potential as biomarkers and immunotherapy targets[19–23].

In this work, we integrate MB proteome datasets[15–17] with data from 62 FFPE MB cases and establish a joint MB proteome dataset ($n = 176$) that is comprehensively compared to DNA methylome data—a current gold standard for molecular brain tumor classification[24]. Further, global N-glycosylation patterns of MB are assessed and correlated with identified proteome subtypes. Taken together, we present a large integrated study of the MB proteome, DNA-methylome and N-glycome, revealing further insights into MB phenotypes, potential biomarkers and therapeutic targets.

## Results

### Integration of in-house proteome data and publicly available datasets enables large-scale proteome analysis of MB

Proteome analysis was performed for 62 FFPE MB tumors (53 primaries, 9 recurrent cases). Additionally, 53 cases were analyzed using DNA methylation profiling. Principal component analysis (PCA) and hierarchical clustering (HCL) distinguished the four main molecular subgroups of MB (SHH, WNT, G3, G4)[2] similarly to published FF based MB proteome datasets (Fig. 1A, Supplementary Fig. 1A, Supplementary Fig. 3, Supplementary data 1c)[15–17]. Proteome data of FF and FFPE tissue from matched MB cases further showed a high correlation (Supplementary Fig. 2A). The age of used paraffine material did not impact sample clustering, detected protein numbers or abundance levels of housekeeping proteins[25] (Fig. 1B, Supplementary Fig. 4, Supplementary Fig. 17D). Proteins detected in WNT and SHH MB, showed similar tendencies in FFPE- and FF-MB datasets[16,17] (Supplementary Fig. 1B). We concluded that FFPE tissue is suitable to study proteome patterns in MB. To increase cohort size, we next integrated and harmonized FF-MB proteome datasets from public repositories[15–17] (Fig. 1D). Technical biases were reduced with HarmonizR[26], and harmonized samples of the joint cohort (main cohort) clustered according to the main MB subgroups (Fig. 1 E-G, Supplementary Fig. 5, Supplementary data 1a). Established protein biomarkers for molecular MB subtypes[27], showed expected subgroup-specific abundance patterns in individual studies and in the combined and harmonized data (Fig. 1H). 16,279 proteins were quantified across 167 samples (19xWNT; 57xSHH; 53xG4; 36xG3; 2xno initial main subgroup stated), including 156 primary tumors and 11 recurrences.

### Six proteomic MB subtypes can be assigned to two main, potentially druggable molecular profiles

To define proteome subtypes of MB, consensus clustering was applied (Supplementary Data 1b). 6 stable clusters were identified (Fig. 2 A–D). Clusters were also reflected in RNA data of matched cases ($n = 60$, Supplementary Fig. 3D–F). The assignment reliability of a sample to a respective proteome subtype was indicated as cluster certainty (Fig. 2D, Supplementary data 1c). At the proteome level, non-WNT/non-SHH MBs divided into three groups (pG4, pG3myc and pG3, p = proteome group), while SHH MBs separated into two groups (pSHHs, pSHHt, s = synaptic profile, t = transcriptional profile). WNT MB formed a homogenous cluster (pWNT, Fig. 2D). In general, a high cluster stability was given for all proteome subtypes (median 6/6), except for pG3 samples, that showed high similarity to pG4 and pG3myc respectively (median pG3 5/6, Fig. 2D). Except for one case, corresponding recurrent and primary tumors were assigned to the same proteome subtype (Fig. 2D). The case that switched subtype in

the recurrence situation (from pSHHs to pSHHt) had a low cluster certainty in the primary sample (3/6, Fig. 2D).

Proteome MB subtypes were associated with previously described DNA methylation subtypes[3,5,6] (https://www.molecularneuropathology.org/mnp/[24], Supplementary data 1c, Supplementary Fig. 6B, Fig. 2D). pG3myc patients showed reduced overall survival (Fig. 2E). pWNT patients showed the best overall survival rate (Fig. 2E). Out of 3996 proteins found in at least 30% of samples for each proteome subtype, 529 showed a characteristic abundance in at least one subtype. The top 5 proteins with the lowest $p$ value and highest mean difference were selected as biomarker candidates (Fig. 2F, Supplementary data 2a). For high-risk non-WNT/non-SHH MBs (pG3myc) PALMD, DIEXF, MCN1, TPD52 and PYCR1 were identified. Of note, hedgehog-signaling-induced proteins (MICAL1, GAB1, PDLIM3)[28] showed a higher abundance in both, pSHHt and pSHHs. Protein biomarkers were confirmed in case-matched MB cases (FF versus FFPE tissue, $n = 10$, Supplementary Fig. 2B) and on the RNA level (Supplementary Fig. 3H). Subtype assignments were confirmed in an additional published MB dataset[29] and a technical validation dataset (Supplementary data 5j,k, Supplementary Fig. 17).

The six proteome subtypes could be assigned to two superordinate clusters at the first hierarchy level in the joint (as well as all individual) datasets (Fig. 3, Supplementary Fig. 3). Comparing these two clusters revealed two main molecular profiles: profile 1, comprising of pG3, pG4 and pSHHs and profile 2, comprising of pWNT, pG3myc and pSHHt MBs (Fig. 3A). The two clusters were confirmed in a technical validation dataset (Supplementary Fig. 17). Matched RNA expression profiles also confirmed a clustering of cases according to these defined profiles ($n = 60$, Supplementary Fig. 3F). We next used gene set enrichment analysis (GSEA) to reveal potential underlying mechanisms and signaling pathways. Synaptic/immunological processes and phospholipid signaling were observed for profile 1 and a replicative/transcriptional signature was observed for profile 2 (Fig. 3A, B, $q$ value < 0.05, Supplementary data 3a, b,h, Supplementary data 10e,f). In order to find drug targets and predict downstream effects we used the Ingenuity Pathway Analyses (IPA) tool and focused on the top two upregulated genesets based on differentially abundant proteins in profile 1 (opioid signaling and SNARE complex) and profile 2 (EIF2 signaling and cell cycle control of chromosomal replication, Fig. 3 B, C, Supplementary data 3c–g)[30]. Tumors of profile 1 could potentially be targeted by several drugs, including the NMDA receptor antagonist memantine. Profile 2 tumors (replicative/transcriptional signature) could—besides others—be targeted by CDK4 or DNA polymerase inhibitors (Fig. 3B–E, Supplementary Fig. 7, Supplementary data 3c–g).

### Group-specific correlation of the DNA methylome and the proteome reveals different conservation levels of molecular characteristics across proteomic MB subtypes

Since DNA methylome data is routinely used in brain tumor diagnostics[1], we decided to integrate our proteome data with DNA methylome data to investigate 1) the general correlation between the two data types and 2) if protein biomarkers are reflected at DNA methylome level. To integrate the data modalities, multiblock data integration using sparse partial least squares discriminant analysis (sPLS-DA) was performed between DNA methylation data (115 samples, 10,000 differentially methylated CpG sites between the MNP v12.5 defined subtypes) and proteome data (115 samples, 3990 quantified proteins present in 30% of samples, Supplementary Fig. 8A–C, Supplementary data 1b,d)[31]. Only a fraction of features out of the 381,717 probes and 3990 proteins showed correlation upon data integration using DIABLO from mixOmics, discriminating mainly the WNT subtype (Fig. 4A, arrows, correlation cut-off >0.7, Supplementary data 4h, Supplementary Fig. 9A–E). To refrain from any data bias, we next performed an MB subtype-specific correlation between complete DNA methylome data (115 samples and 381,717 CpG sites) and proteome data (115 samples, 3990 proteins, Fig. 4B, C). A significantly

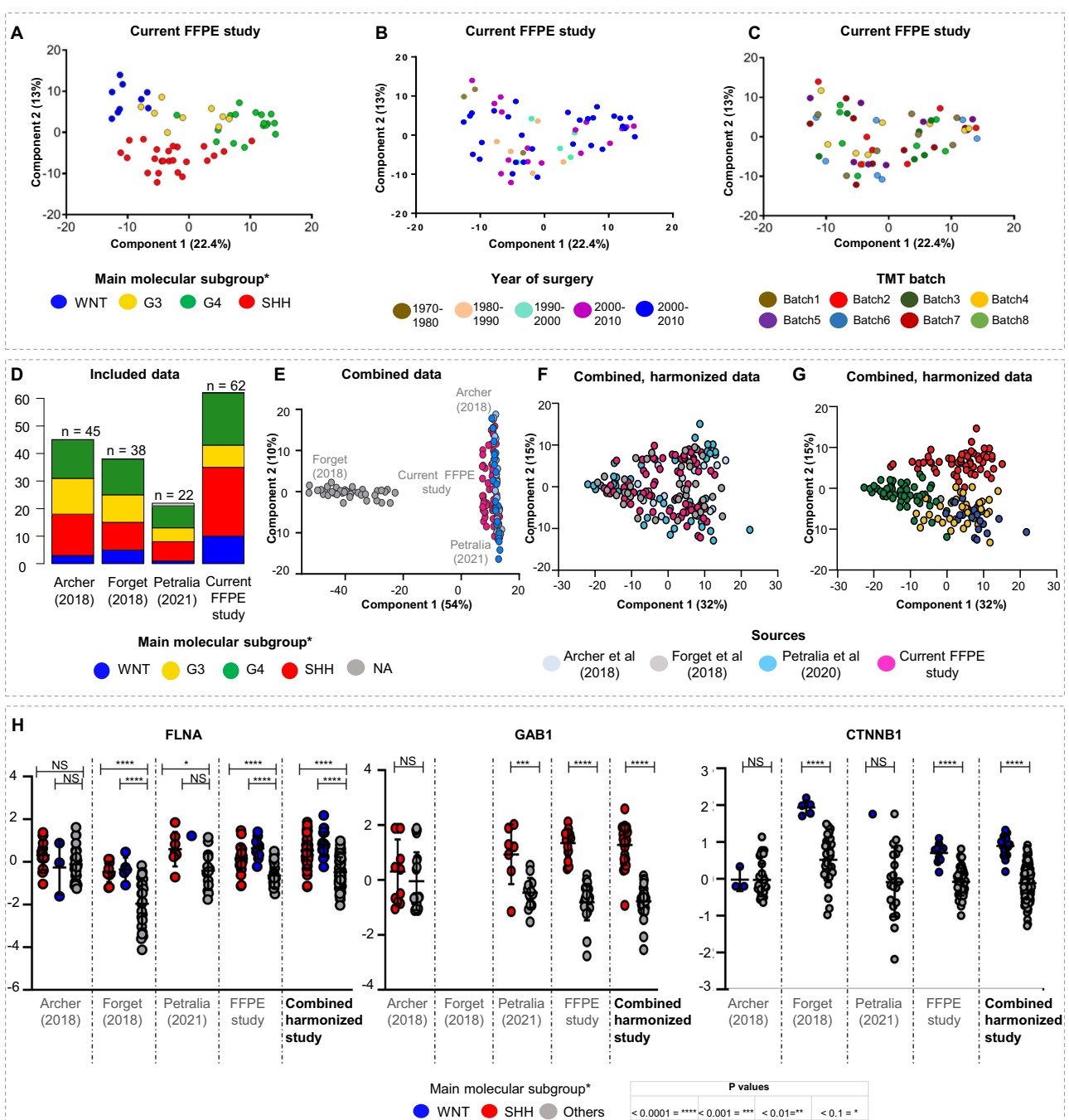

**Fig. 1 | Harmonization and integration of proteome Medulloblastoma (MB) datasets.** NIPALS principal component analyses (PCA) of measured FFPE samples ($n = 62$) with assignment to (**A**) the four main molecular MB subgroups[2], (**B**) age of measured samples, (**C**) measured TMT batch. **D** Overview of analyzed datasets. PCA of data before (**E**) and after (**F**, **G**) data harmonization using ComBat in the HarmonizR framework annotated for the source of the samples (**F**) and for main molecular MB subgroups ($n = 167$, source data file has been provided). **H** Protein abundance of the the WNT and SHH MB marker FILAMIN A ($n_{SHH\_Archer} = 15$, $n_{WNT\_Archer} = 3$, $n_{Others\_Archer} = 27$, $n_{SHH\_Forget} = 10$, $n_{WNT\_Forget} = 5$, $n_{Others\_Forget} = 23$, $n_{SHH\_Petralia} = 7$, $n_{WNT\_Petralia} = 1$, $n_{Others\_Petralia} = 14$, $n_{SHH\_FFPE} = 25$, $n_{WNT\_FFPE} = 10$, $n_{Others\_FFPE} = 27$, $n_{SHH\_combined} = 57$, $n_{WNT\_combined} = 19$, $n_{Others\_combined} = 91$, two-tailed, unpaired $t$ test, $p_{shhArchervsOtherArcher} = $ n.s., $p_{shhForgetvsOtherForget} < 0.0001$, $p_{WNTForgetvsOtherForget} < 0.0001$, $p_{shhPetraliavsOtherPetralia} = 0.02$, $p_{wntPetraliavsOtherPetralia} = $ n.s., $p_{shhFFPEvsOtherFFPE} < 0.0001$, $p_{wntFFPEvsOtherFFPE} < 0.0001$,

$p_{shhcombinedvsOthercombined} < 0.0001$, $p_{shhcombinedvsOthercombined} < 0.0001$) SHH MB marker GAB1 ($n_{SHH\_Archer} = 15$, $n_{Others\_Archer} = 30$, $n_{SHH\_Forget} = 10$, $n_{Others\_Forget} = 28$, $n_{SHH\_Petralia} = 7$, $n_{Others\_Petralia} = 15$, $n_{SHH\_FFPE} = 25$, $n_{Others\_FFPE} = 37$, $n_{SHH\_combined} = 57$, $n_{Others\_combined} = 110$, two-tailed, unpaired $t$ test, $p_{shhArchervsOtherArcher} = $ n.s., $p_{shhPetraliavsOtherPetralia} = 0.008$, $p_{shhFFPEOtherFFPE} < 0.0001$, $p_{shhcombinedOthercombined} < 0.0001$)., and the WNT MB marker CTNNB1 ($n_{WNT\_Archer} = 3$, $n_{Others\_Archer} = 42$, $n_{WNT\_Forget} = 5$, $n_{Others\_Forget} = 33$, $n_{WNT\_Petralia} = 1$, $n_{Others\_Petralia} = 21$, $n_{WNT\_FFPE} = 10$, $n_{Others\_FFPE} = 52$, $n_{WNT\_combined} = 19$, $n_{Others\_combined} = 148$, two-tailed, unpaired $t$ test, $p_{wntArchervsOtherArcher} = $ n.s., $p_{wntForgetvsOtherForget} < 0.0001$, $p_{wntPetraliavsOtherPetralia} = $ n.s., $p_{wntFFPEOtherFFPE} < 0.0001$, $p_{wntcombinedOthercombined} < 0.0001$). Data are presented as mean values $\pm$ SD in each dataset individually and in the joint dataset after harmonization PCAs are based on $\geq 70\%$ valid values, *: $p < 0.05$, **$p < 0.01$, ***$p < 0.001$, ****$p < 0.0001$, n.d. = not detected, NS = not significant, $n$ represents biologically independent human samples.

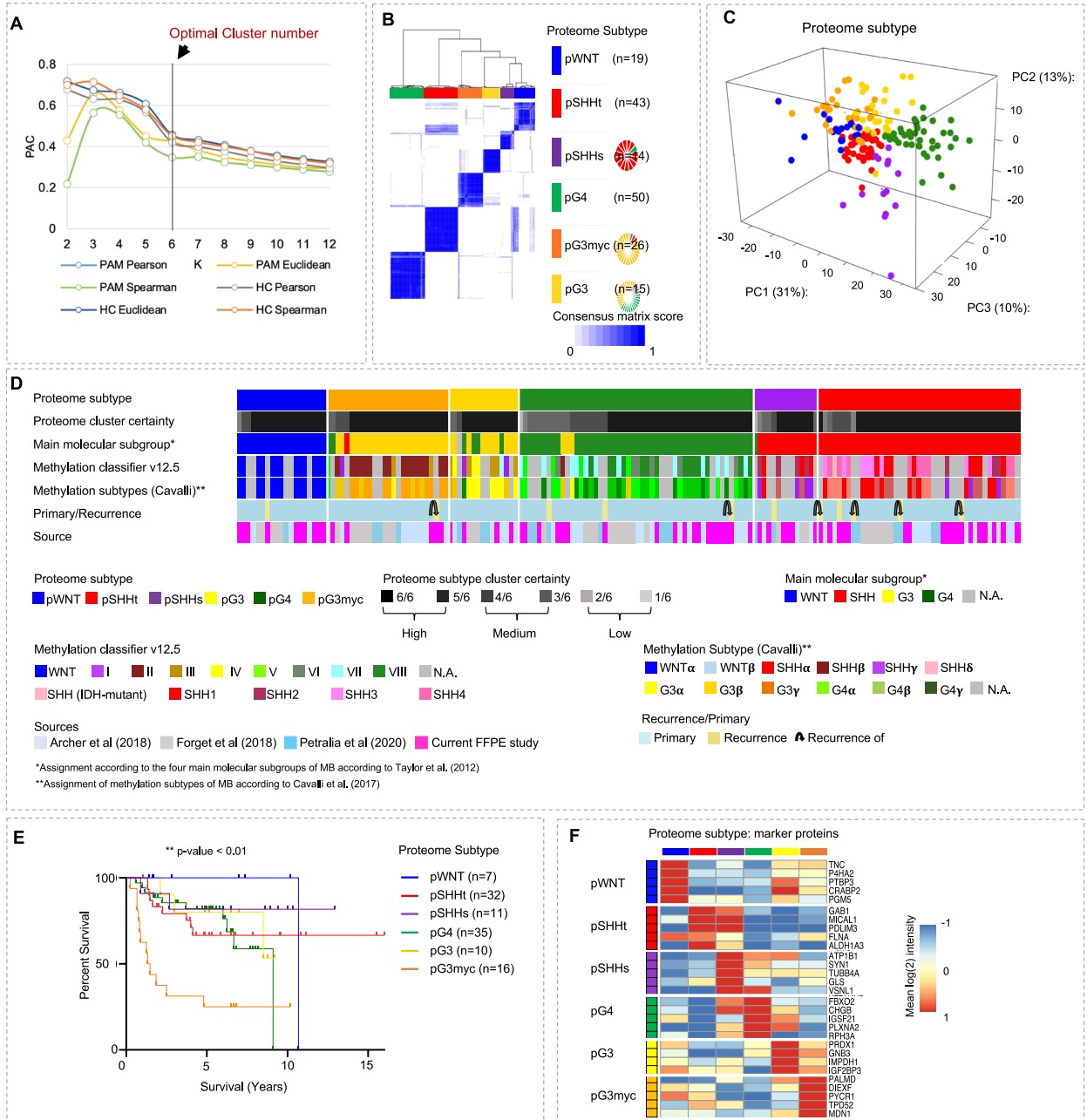

**Fig. 2 | MB segregate into six proteome subtypes. A** Proportion of ambiguous clustering (PAC) scores for $k = 2$–12 in consensus clustering of the main cohort, using different cluster algorithms ($n_{MB} = 167$, based on ≥30% valid values). **B** Optimal clustering of proteome data. Consensus scores are shown in color scale from white (samples never cluster together) to blue (samples always cluster together). Six proteome subtypes, pWNT, pSHHt, pSHHs, pG3myc, pG3 and pG4, were defined. **C** Visualization of the first three principal components. **D** Clinical sample information. **E** Log-rank (Mantel-Cox) test comparing the survival curves of proteome subtypes ($p$ value < 0.001, overall χ2-square test). **F** Group specific mean log 2 protein intensity of protein subtype marker candidate proteins. $n$ represents biologically independent human samples.

higher number of proteins of the pWNT (38.14%, 1552 proteins) and pG3 subtype (45.41%, 1812 proteins) correlated with at least one CpG site of their own gene, when compared to the other subtypes (range 1.52–6.49 %, Fig. 4B, Supplementary data 4b-g). Only 12.2–18% of protein correlating CpG sites were located at the transcriptional start site (TSS200, TSS1500, Exon1, Fig. 4B). Integrating the proteome data with DNA methylome data based on differentially methylated regions (DMR) confirmed a high correlation of features in pWNT MB (Supplementary Fig. 10 A, B). Focusing on the 31 previously selected biomarker candidates (Fig. 2F), we found 10 proteins correlating with CpG sites of their own gene across subtypes (Fig. 4C, D, Supplementary data 4a). In summary, DNA-methylation changes were only partly

reflected at the protein level, with different feature conservation levels for different proteome subtypes.

## SHH MB comprise two proteome subtypes showing a synaptic or DNA transcription/translation signature

SHH MB split into two proteome subtypes (pSHHt and pSHHs, Fig. 5A). All pSHHs cases with high cluster certainty (6/6) occurred in patients below 3 years of age. The DNA methylation subtypes SHH3 (8/29) and SHH4 (9/29) were exclusively found in pSHHt MBs (Fig. 5A). Methylation subtypes SHH1 and SHH2 were seen in both pSHHs and pSHHt (SHH1: $p = 0.43$, SHH2: $p = 0.10$, X2−test). We then analyzed the distribution of SHH pathway alterations, which are driver events in SHH

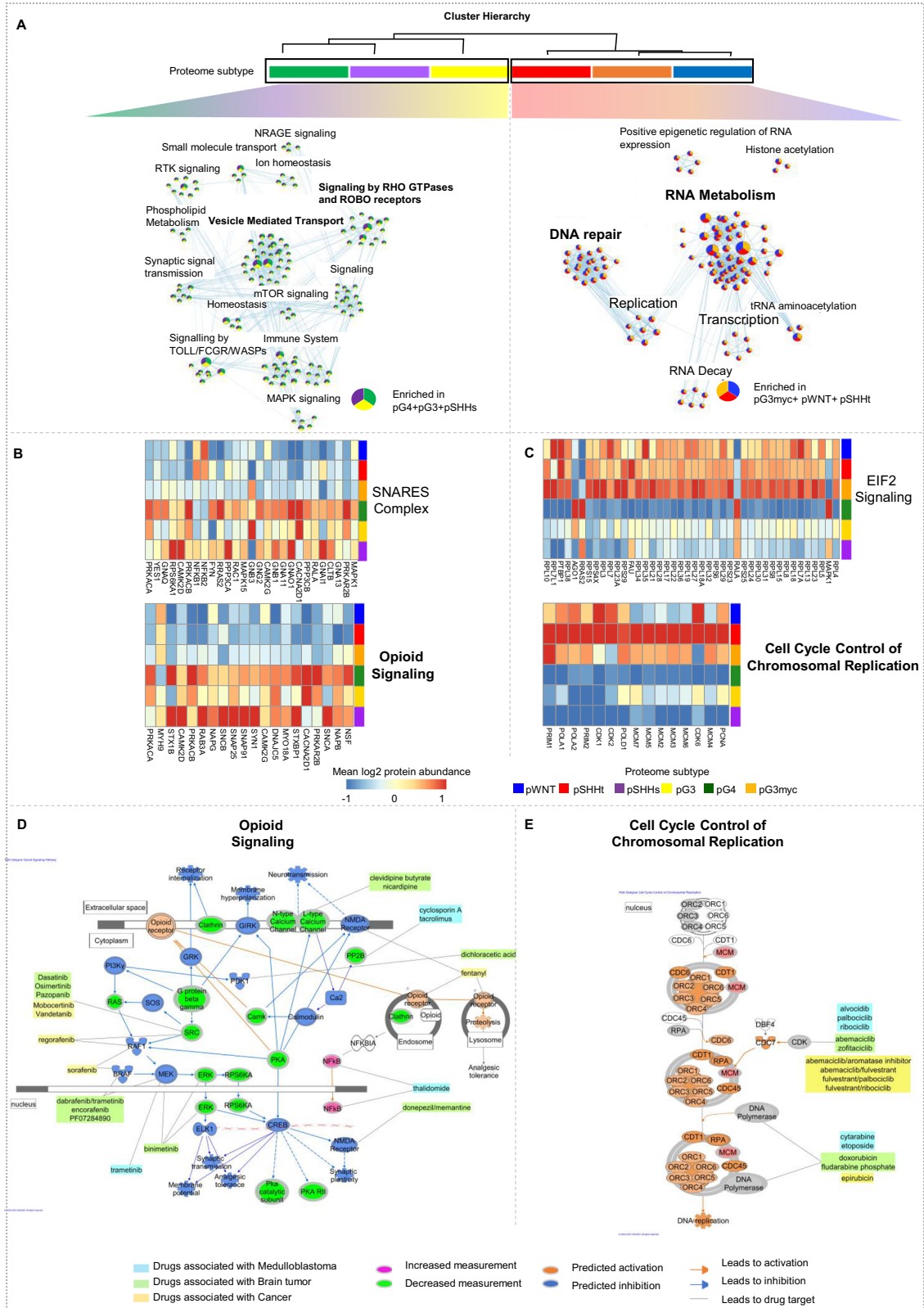

**Fig. 3 | Proteome subtypes of MB can be assigned to two main profiles.**
**A** Proteome cluster similarity hierarchy based on stepwise increasing k-means execution from $k = 2$–6 with network analyses showing gene set overlap dependent MCL clustering of enriched gene sets, comparing pG3, pG4 and pSHHs ($n = 79$, profile 1), to pWNT, pG3myc, pSHHt ($n = 88$, profile 2). Gene set enrichment analysis (GSEA) was based on REACTOME pathways for all analysis. Top two upregulated genesets based on differentially abundant proteins using Ingenuity Pathway

Analyses (IPA) in profile 1 (opioid signaling and SNARE complex (two-tailed, unpaired t-test, log2 FC > 1.5 and p value < 0.05)) (**B**)) and profile 2 (EIF2 signaling and cell cycle control of chromosomal replication (log2 FC > 1.5 and p value < 0.05) (**C**). IPA-based pathway analyses of opioid signaling (**D**) and cell cycle control of chromosomal replication (**E**) indicating therapeutic targets with respective drugs. n represents biologically independent human samples.

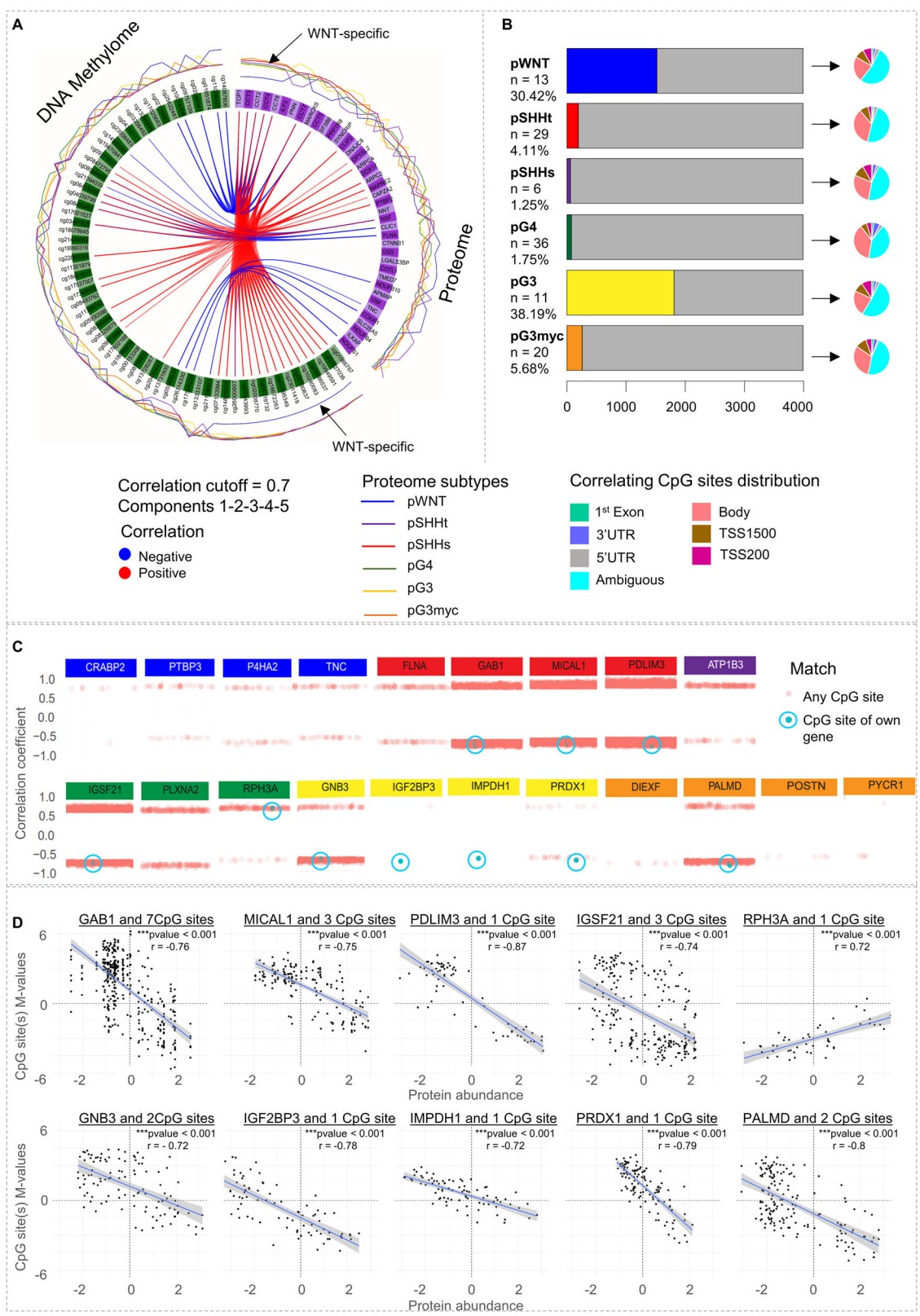

MBs[32]. *PTCH1* mutations were found exclusively but not mandatory in pSHHt tumors. *SUFU, SMO, MYCN* or *GLI2* alterations did not distribute differentially (Fig. 5A). Proteome subtypes of SHH MB were not clearly separated at the transcriptome level, which is in line with previous results[16] (matched samples *n* = 21, Supplementary Fig. 11 A-B).

MBs[32]. *PTCH1* mutations were found exclusively but not mandatory in pSHHt tumors. *SUFU, SMO, MYCN* or *GLI2* alterations did not distribute differentially (Fig. 5A). Proteome subtypes of SHH MB were not clearly separated at the transcriptome level, which is in line with previous results[16] (matched samples *n* = 21, Supplementary Fig. 11 A-B).

In order to analyze how copy number alterations might be reflected at the proteome level, the proteome abundance for each gene was mapped to chromosomal arms, which will be referred to as "proteome copy number variation (CNV)" henceforth. Both pSHHt and pSHHs groups showed a low overall correlation between calculated CNVs using matched DNA methylation data and proteome data ($r_{pSHHs}$ =0.01, $r_{pSHHt}$ = 0.20, Fig. 5D, G, Supplementary data 5g, h).

To get insights into changed pathways in pSHH subtypes, a network clustering based on gene set enrichment using pSHHs or pSHHt-

**Fig. 4 | Correlation between DNA methylome and proteome features. A** Circular plot from mixOmics analyses based on selected features of the first five components from proteome and methylome data. The plot illustrates features with correlation $r > 0.7$ represented on side quadrants. Proteome group-specific feature levels are shown in the outer circle. **B** Proteome subtype-specific Pearson correlation calculated between matched proteins and CpG methylation sites. The number of proteins correlating with CpG site methylation of their own gene ($r > 0.7$) is shown in color. The pie chart shows the distribution of correlating CpG sites concerning the position in a gene. **C** Subtype independent Pearson correlation between 3990 proteins and 381,717 methylation probes focusing on subtype-specific biomarkers. Pearson Correlations $>0.7$ are shown, CpG sites correlating with the corresponding gene are highlighted in blue. Some biomarkers correlated with more than one CpG site of their own gene (GAB1: 7, GNB3: 2, IGSF21: 3, MICAL1: 3, and PALMD: 2). **D** Scatterplot of the 10 biomarker proteins correlating with the CpG site(s) of their own gene (Pearson correlation $> 0.7$, $p < 0.001$). The linear regression line was aligned for all correlating CpG site(s), SE = 0.95.

specific proteins was performed (Fig. 5 B, C, E, F, H, Supplementary data 5a–f, $q$ value $< 0.05$). Differential proteins in pSHHs revealed differences in synaptic, mitochondrial, and immunological processes, whereas proteins in pSHHt MB were involved in post-translational protein modification, transcription/translation, DNA repair and cell cycle. In accordance with the latter profile, pSHHt showed a significantly enhanced proliferation (assessed via ki67 staining, Supplementary Fig. 11 E, F, Supplementary data 5n). ALDH1A31 was highly abundant in both pSHH groups (Fig. 2F, Fig. 5E), which could be confirmed via immunohistochemistry (Supplementary Fig. 11C, D).

Analyses of hallmark gene sets additionally revealed a distinct upregulation of proteins involved in the TCA cycle in pSHHs, indicating metabolic differences between the subtypes (Fig. 5H). Subsequent analyses of metabolites and aminoacids confirmed distinct metabolic patterns in pSHHt and pSHHs (Supplementary Fig. 12). Of note pSHHs showed a lower abundancy of Isocitrate dehydrogenases, together with a decrease of Isocitrate, alpha-Ketoglutarate and Glutamine, indicating a higher consumption of the latter three (Supplementary Fig. 12C, Supplementary data 5l-m). Alpha-Ketoglutarate and Glutamine can be further processed to Glutamate and then GABA, which are both involved in synaptic signaling. In line with these findings, we detected a significant increase of GABA target proteins in pSHHs (Supplementary Fig. 12C).

We did not detect a significant difference in survival between pSHHs and pSHHt (Fig. 5I). However, *TP53* mutations, used for stratification of high-risk SHH MB[33], mainly occurred in the pSHHt subtype (9/10, but differential distribution was not significant ($p = 0.43$, $X^2$–test)). As expected, *TP53* mutations within the pSHHt group significantly correlated with bad prognosis (Fig. 5I). *TP53* mutated MBs did not form a distinct proteome cluster. However, 134 differentially abundant proteins were detected between pSHHt-*TP53* wildtype and pSHHs-*TP53* mutated MBs (Fig. 5J, Supplementary data 5i).

### High-risk pG3myc MBs are characterized by a MYC profile and high abundance of Palmdelphin

We found three different non-WNT/non-SHH MB proteome subtypes: pG3, pG3myc and pG4 (Fig. 6A). pG4 exclusively included the main molecular subgroup G4, whereas pG3myc was dominated by G3 patients. pG3 included both molecular subgroups (Fig. 2D). pG3myc was dominated by large cell anaplastic histology (LCA). LCA histology and *MYC* amplification are used for high-risk tumor stratification in non-WNT/non-SHH MBs[34]. Accordingly, *MYC* amplifications were predominantly detected in pG3myc tumors. However, not all pG3myc classified cases were *MYC* amplified. In concordance with these high-risk characteristics, a broad fraction of pG3myc cases were assigned to the methylation subtype II (16/20 cases, 80%)[24,35] or group G3 δ[5] (13/20 cases, 65 %, Fig. 6A). Clinically, most pG3myc tumors were classified as M3 and tumors showed the worst overall survival compared to all other MB subtypes (Fig. 6A, Fig. 2E). Distinct protein abundance patterns and pathway enrichments were seen for pG3, pG4 and pG3myc each and all showed a low overall correlation between calculated proteome and DNA methylation CNV data (Fig. 6 B-J, Supplementary data 6a-l). Specifically, pG3myc MB showed a significant enrichment of MYC target proteins (FDR $< 0.25$; $p$ value $< 0.0001$, Fig. 6K). In line with this, pG3myc MB showed a high fraction of tumor cell nuclei with accumulation of MYC (Supplementary Fig. 13). Moreover, pG3myc MB

differed from pG3 and pG4 showing enhanced signaling by ROBO receptors and an underrepresentation of proteins involved in MHCII class antigen presentation (Fig. 6P, Supplementary data 6m). To establish a diagnostically useful biomarker for histological identification of high-risk pG3myc tumors, we focused on the high differentially abundant protein Palmdelphin (PALMD, Fig. 6 H). Digitally supported quantification of PALMD immunostainings confirmed a specific increase of the candidate in pG3myc tumors (Fig. 6L, M). We additionally analyzed how this biomarker is reflected at other omic levels. Indeed, a significantly higher *PALMD* mRNA expression and lower CpG site methylation was detected in pG3myc MBs compared to all other MB subtypes (Fig. 6N)[16]. High *PALMD* mRNA expression was also associated with poor survival in MB (Fig. 6O, Supplementary Fig. 14A–D). Finally, all groups displayed a low overall correlation between calculated proteome CNV and DNA methylation CNV data (Fig. 6D, G, J, Supplementary data 6j–l).

### pWNT MB show low abundance of the multiprotein complex TriC/CCT

WNT MB did not divide into further subtypes based on proteome profiles (Fig. 7A). Among differentially abundant proteins in comparison to other MB subtypes TNC showed the highest abundance (14.7 foldchange, Fig. 7B, Supplementary data 7a). A significantly high intensity of TNC in pWNT MB was confirmed using digitally supported immunostaining quantification (Fig. 7C, D, E). Using a publicly available dataset[5], a higher mRNA expression of *TNC* in WNT MB was confirmed (Fig. 7F). In contrast, CpG sites of the *TNC* gene, showed no significant difference of methylation (pWNT versus other subtypes (Fig. 7G, Supplementary Fig. 14A–C). GSEA revealed an enrichment of extracellular matrix proteins and N-glycan biogenesis and transport (FDR $< 0.25$; $p$ value $< 0.0001$, Fig. 7H, I, Supplementary data 7b, c). A high overall correlation between copy number plots extracted from proteome and DNA methylome data was observed for pWNT compared to all other subtypes (Fig. 7J, Supplementary data 7d), being in line with a general increased overall correlation of proteome and DNA methylome data (Fig. 4A, B).

The highest similarity of proteome profiles was observed for the pG3myc subtype, associated with high-risk features and the pWNT subtype-associated with relatively good overall survival (Fig. 3A). Both subtypes showed a "transcriptional/translational" profile (Fig. 3A, B) and a high abundance of MYC target proteins along with a high fraction of MYC-positive tumor cell nuclei (Fig. 6K, Supplementary Fig. 13). We therefore asked, what molecular changes could impact such diverse clinical behavior. Differentially abundant proteins between pG3myc and pWNT MBs included TriC/CCT proteins and the established WNT MB marker β-CATENIN[36] (Fig. 8A, Supplementary data 8a). Among the top discriminating gene sets was the association with TriC/CCT target proteins and asparagine-linked N-glycosylation (FDR $< 0.25$; $p$ value $< 0.0001$, Fig. 8B, Supplementary data 8b-c).

As the TriC/CCT complex has previously been reported to be associated with vincristine resistance and typical chemotherapy regimens for MB contain vincristine in the treatment combination[36], we further focused on this chaperonin containing multiprotein complex (Fig. 8C, E, Supplementary Fig. 15A). Among MB subtypes, pWNT MBs showed the lowest abundance of TriC/CCT proteins, whereas pG3myc

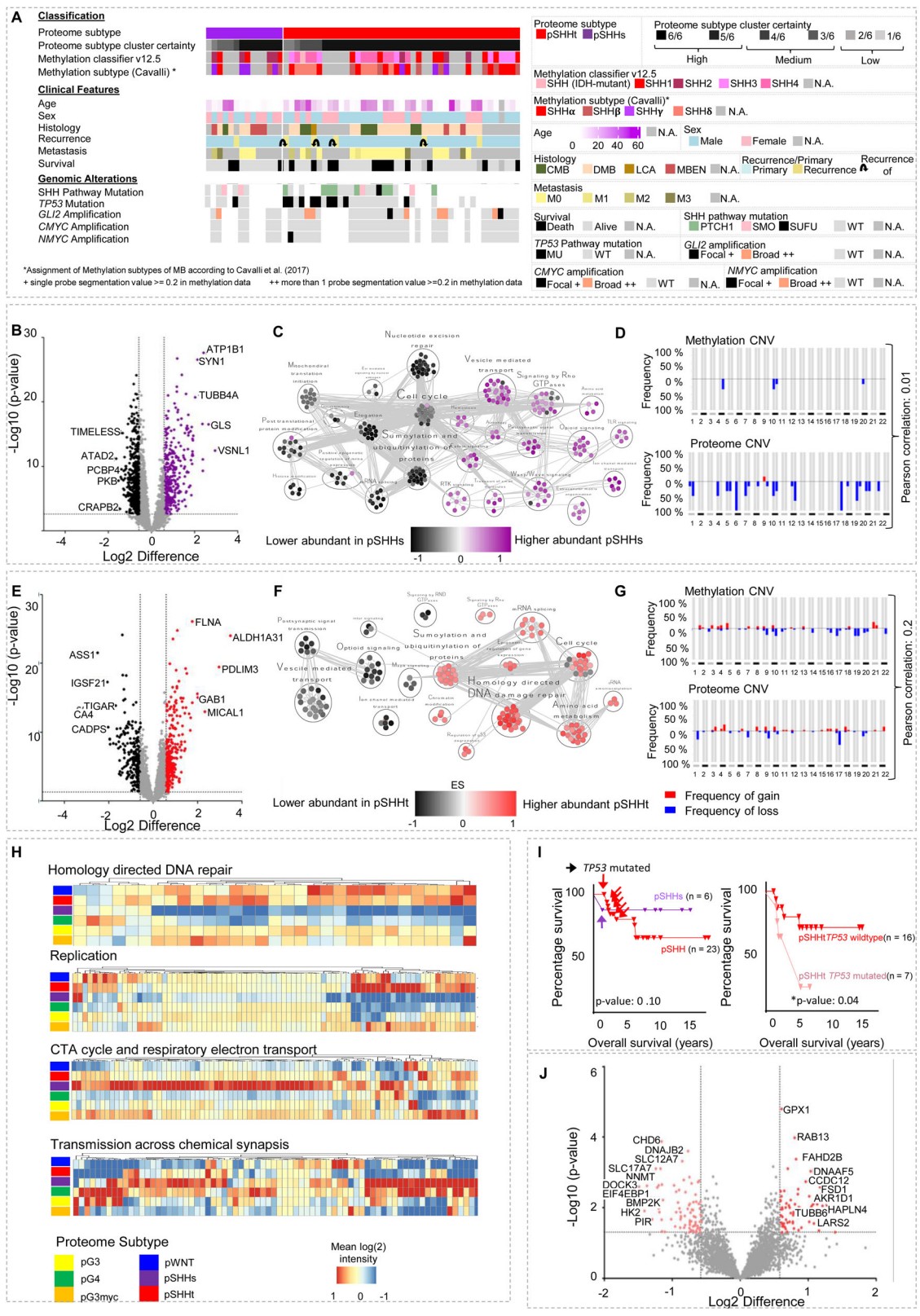

MBs displayed the highest amount. High abundance of TriC/CCt proteins in pG3myc was confirmed at mRNA level. Matched cases, as well as publicly available transcriptome data[5] did not show a statistically significant downregulation of all component mRNAs in pWNT MB when compared to other subtypes (Fig. 8C, Supplementary Fig. 15). Further, no difference of TriC/CCT gene methylation was detected among subgroups (Fig. 8C, Supplementary data 8d). Focusing on each

TriC/CCT component individually, we saw a mainly negative association between DNA methylation and RNA expression and a mainly positive one between transcriptome and proteome data–as expected (Fig. 8D). However, correlation of DNA methylome and proteome data did not point in such a clear direction (Fig. 8D). Consequently, only CCT2 showed a high association among all omic levels with a correlation score ≥0.7 (Fig. 8E, Supplementary data 8d). We therefore,

**Fig. 5 | SHH MB comprise two proteome MB subtypes. A** Histological, molecular, and clinical characteristics of the MB subtypes pSHHt ($n = 43$) and pSHHs ($n = 14$). **B** Volcano plot showing differentially abundant proteins comparing pSHHs tumors to all other proteome subtypes (two-tailed, unpaired $t$ test, $p$ value < 0.05; log2FC > 1.5). **C** MCL clustering of enriched gene sets in pSHHs MBs. **D** Copy number variations (CNV) plots of matched pSHHs MB ($n = 6$) calculated from either DNA methylation or proteome data with pearson correlation between both omic types ($r = 0.01$). **E** Differentially abundant proteins when comparing pSHHt tumors to all other proteome subtypes (two-tailed, unpaired t-test, $p$ value < 0.05; log2FC > 1.5). **F** MCL clustering of enriched gene sets in pSHHt. **G** CNV plots for matched pSHHt MBs ($n = 29$) calculated from either DNA methylation or proteome data with Pearson correlation between both omic types ($r = 0.2$) **H** Heatmaps showing mean MB subtype protein abundance hallmark genesets homology directed repair (GSEA differential expression analysis normalized enrichment score (NES), $NES_{pSHHt} = 2.2$, $p = <0.0001$, FDR < 0.25), replication ($NES_{pSHHt} = 2.2$, $p = 0.01$), TCA cycle and respiratory electron transport ($NES_{pSHHs} = 3.9$, $p = <0.0001$, FDR < 0.25) and transmission across chemical synapses ($NES_{pSHHs} = 3.2$, $p = <0.0001$, FDR < 0.25) based on differentially abundant proteins. **I** Overall survival of pSHHt MB ($n = 23$) and pSHHs MB (n = 5) and overall survival of pSHHt MB depended on TP53 mutation status. TP53 mutated cases displayed a significantly worse survival (Mantel cox test $p$ value = 0.04). **J** Volcano plot, showing differentially abundant proteins when comparing TP53 mutated cases to wildtype cases in pSHHt tumors (two-tailed, unpaired $t$ test, $p$ value < 0.05; log2FC > 1.5). $n$ represents biologically independent human samples.

identified the TriC/CCT complex as a feature discriminating pWNT and pG3myc MB.

### MB subtypes show distinct N-glycan profiles

One of the major altered genesets between pWNT and pG3myc MB was N-glycosylation (Fig. 8B), referring to a post-translational modification which is unknown in the context of MB. As glycosylation plays a major role in immune system response and might therefore enable therapeutic options[37,38], we focused on this aspect in more detail. Of note, proteins involved in all aspects of N-glycosylation (synthesis, processing, transport, and antigen presentation via MHC class II) were overrepresented in pWNT (Fig. 9C). Quantitative analysis of N-glycans revealed differential N-glycosylation patterns across proteomic MB subtypes (Fig. 9D–I). In total 302 N-Glycan species were identified (Fig. 9 E–I; Supplementary data 9a). For non-WNT/non-SHH MB a higher number of N-glycans were identified in comparison to pWNT, pSHHs and pSHHt (Fig. 9F, Supplementary data 9a). At the quantitative level, proteome MB subtypes were reflected based on their N-glycan profiles (Fig. 9G, Supplementary Fig. 16A). 92 N-glycans were differentially abundant between the proteome MB types (Supplementary Fig. 15B, Supplementary data 9b). We identified the highest number of exclusive (complex) N-glycans in the subtypes pG3myc and pG4 ($n_{pG3myc} = 22$, $n_{pG4} = 12$, Fig. 9H, I, where n represents (complex) N-glycans). Frequently described key factors in tumors are the upregulation of cancer-associated sialylated N-glycans as well as aberrant fucosylation[39]. A higher proportion of sialylated N-glycans was found in non-WNT/non-SHH tumors (non-WNT/non-SHH MB: 59.7–62.0% versus pWNT/pSHH: 49.5–51.9%). A significantly lower proportion of fucosylated N-glycans was detected in pSHHt, compared to all other subtypes (66.7 % ($n = 74$)) versus 72.1–80% ($n = 101$-174, range of the other MB subtypes, where n represents number of fucosylated N-glycans).

Taken together, integrated proteome analyses shed light on unique characteristics in MB subtypes revealing potentially druggable targets. To show the validity of results, we recapitulated the six proteome subtypes and two superordinate profiles found in the integrated cohort in a technical and biological validation dataset of FFPE samples ($n_{technical\ cohort} = 57$, $n_{biological\ validation\ cohort} = 31$, Fig. 10A-G, Supplementary Fig. 17, Supplementary data 1c, Supplementary data 10a–c,g, Supplementary data 11). We further verified the differential feature conservation between DNA methylation and protein patterns in the biological validation cohort und underlined the TriC/CCT complex as a discriminator of pWNT and pG3myc MB (Fig. 10H, I, Supplementary data 10b,h).

### Discussion

Technical variability and missing values are a general challenge of mass spectrometry-based proteome analyses implying a need for large integrated datasets with reduction of technical biases. Using the HarmonizR integration strategy[26], we could successfully identify clinically relevant proteome subtypes of MB in a large, integrated cohort of 167 MBs. Herein, we show that FFPE material,

which maintains chemical rigidity under cheap storage conditions[40], enabled the identification and differentiation of molecular subtypes, as previously described for smaller cohorts of FF tissue[16,17]. Respective results could moreover be confirmed in technical and biological FFPE validation datasets. In line with previous results[41], sample age did not impact data quality, making FFPE tissue highly suitable for large-scale analysis of rare diseases[18].

Two overriding molecular patterns were observed across MB subtypes, indicating that MB either follow a transcriptional/replicative (pWNT, pSHHt, pG3myc) or synaptic/immunological (pG4, pSHHs, pG3) profile. These profiles tempt to speculate, that MBs with a synaptic/immunological pattern (in contrast to MBs with a transcriptional/replicative pattern) may depend more on external stimuli, such as e.g., (potential) synaptic input. Further studies are therefore needed to comprehend the underlying functional background resulting in the observed patterns. To evaluate the therapeutic potential of these patterns, we used IPA[30] and identified, besides others, CDK4 inhibitors as potential drugs for targeting the groups belonging to the transcriptional profile. Various CDK inhibitors are already FDA-approved for treatment of different types of metastatic cancers and CDK4/6 inhibition has been shown to inhibit tumor growth of medulloblastoma cells in vivo[42,43]. In contrast, proteome subtypes belonging to the synaptic profile may be—besides others—targeted with the NMDA receptor antagonist memantine. Of note, memantine has neuroprotective properties and was shown to decrease cognitive dysfunction in patients receiving radiotherapy[44,45]. As radiotherapy is also applied to MB patients the drug may be of specific interest, however, further studies are needed to investigate the clinical potential of the mentioned drugs for MB patients.

We found that DNA-methylation subgroups of MB—which are used for classification of brain tumors in the clinic[1]—are associated with proteome subtypes. This underlines, that the proteome harbors a great potential for identifying subtype-specific therapy targets[3–6,24]. However, only 30% of marker proteins showed a significant correlation with their respective gene's CpG sites. In general, a low correlation between proteome and methylome data was found in MB, in line with the results of previous studies on other tumor entities[46,47]. Poor correlations might be attributed to the 850 K array design since it mostly assesses promoter methylation sites whereas CpG sites correlating well with gene expression may be located further away from transcriptional start sites[48]. Of note, correlation levels of data modalities were not evenly distributed among subtypes. Especially in pWNT tumors, proteins showed a relatively high correlation with their respective gene's CpG sites (38.9% of proteins). In addition, the commonly detected loss of chromosome 6[49] was also reflected in proteome data when mapping protein abundances to chromosomal arms. Molecular alterations may hence be more conserved for WNT MBs, whereas DNA-based methylation differences do not always result in an effective change in protein abundance, probably due to post-transcriptional and post-translational mechanisms (Supplementary Fig. 18). These findings highlight the importance of proteome analysis to detect targetable alterations.

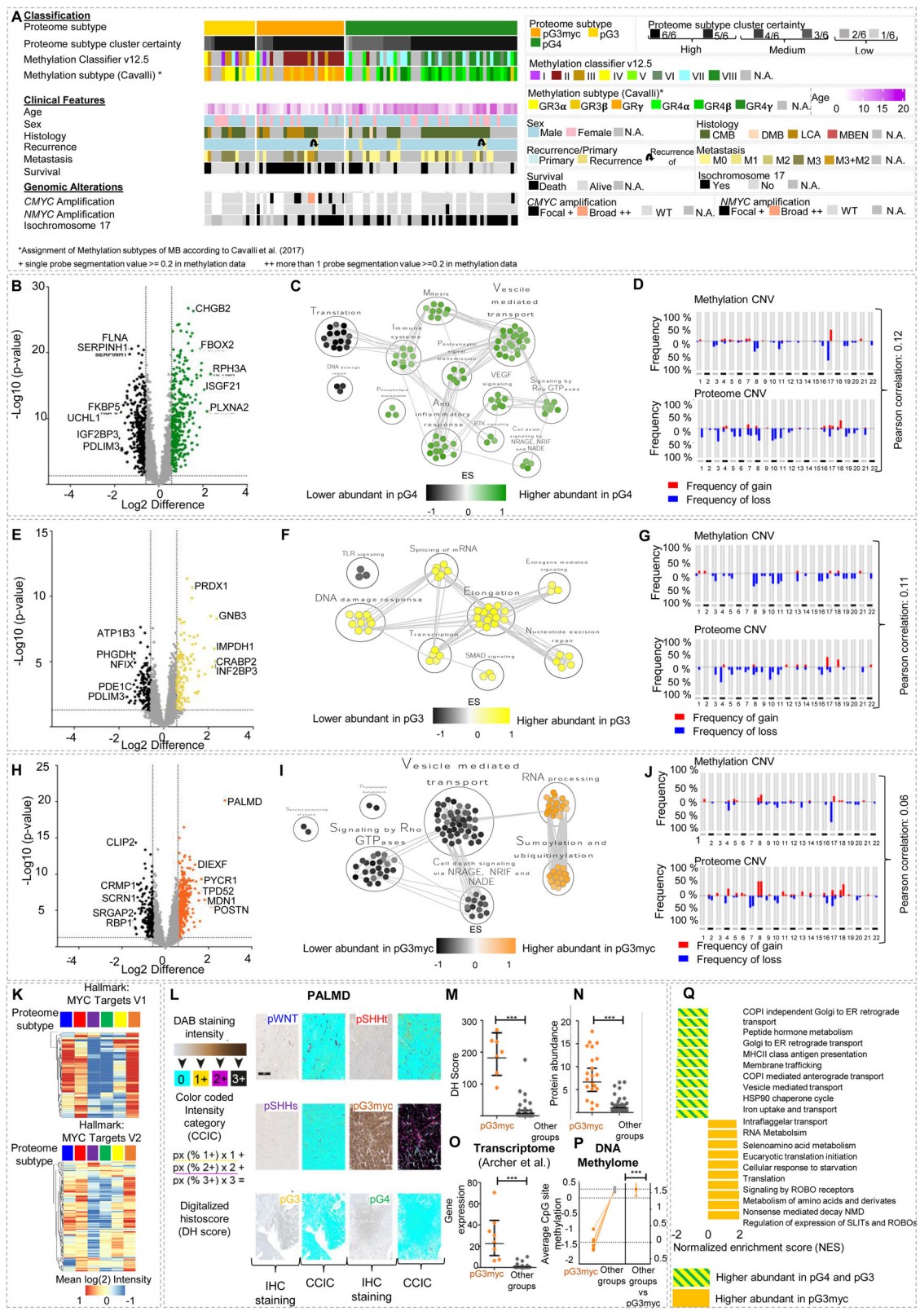

We detected two proteome SHH MB subtypes, namely pSHHs and pSHHt. While we cannot fully exclude the possibility that differences in proteome patterns could be due to variations in tissue composition, our results confirmed previously reported proteome patterns in SHH MB[16]. pSHHs tumors reflect the SHHb subgroup defined by Archer et al. [16], showing enrichment of synaptic pathways[16]. We found that pSHHs MBs are characterized by a high representation of the citric acid (TCA) cycle

and respiratory electron transport, pointing at distinct metabolic profiles of SHH proteome subtypes. Metabolic analysis confirmed significant differences with isocitrate (ISO) and α-ketoglutarate (αKET) being significantly downregulated in pSHHs MBs. As pSHHs MBs also showed a high protein abundance of isocitrate dehydrogenases, this may indicate a higher consumption of these metabolites. As both αKET as well as the amino acid glutamine were significantly downregulated in

**Fig. 6 | pG3myc tumors display an enhanced MYC target protein profile and can be identified by Palmdelphin (PALMD) staining. A** Histological, molecular, and clinical characteristics of the MB subtypes pG3myc ($n = 26$), pG3 ($n = 15$) and pG4 ($n = 40$). **B** Volcano plot showing differentially abundant proteins when comparing pG4 tumors to all other proteome subtypes (two-tailed, unpaired t-test, $p$ value < 0.05; log2FC > 1.5). **C** MCL clustering of enriched gene sets in pG4 MB. **D** CNV plots of pG4 MBs ($n = 40$) were calculated from either DNA methylation or proteome data with Pearson correlation between both omic types ($r = 0.12$). **E** Differentially abundant proteins when comparing pG3 tumors to all other proteome subtypes (two-tailed, unpaired $t$ test, $p$ value < 0.05; log2FC > 1.5). **F** MCL clustering of enriched gene sets in pG3 MB. **G** DNA methylation or proteome CNV plots of pG3 MB ($n = 11$) with Pearson correlation between both omic types ($r = 0.11$). **H** Differentially abundant proteins in pG3myc MB. Palmdelphin (PALMD) was highly abundant in pG3myc tumors (two-tailed, unpaired $t$ test, $p$ value < 0.05; log2FC > 1.5). **I** MCL clustering of enriched gene sets, in pG3myc MB. **J** DNA methylation or proteome CNV plots of pG3myc MB ($n = 20$) with pearson correlation between both omic types ($r = 0.06$). **K** Mean protein abundance in MB subtypes for hallmark gene sets MYC Targets V1 and MYC Targets V2. **L** Scheme and representative images of digitally supported immunostaining intensity quantification of PALMD immunostainings in MB. Quantified pixels of different staining intensities were used to calculate a digital Histo-score (DHS, source data file has been provided) **M** Significantly enhanced digital histoscore for PALMD in pG3myc MB ($n_{pG3myc} = 7$) compared to all other MB subtypes ($n_{Others} = 22$, $p < 0.0001$, data are presented as mean values ± SD). **N** Protein abundance for PALMD in pG3myc MB ($n_{pG3myc} = 21$) compared to all other MB subtypes ($n_{Others} = 84$, unpaired $t$ test, $p < 0.0001$, data are presented as mean values ± SD). **O** PALMD gene expression in pG3myc MBs ($n_{pG3myc} = 6$) compared to all other MB subtypes ($n_{Others} = 30$, two-tailed, unpaired t-test, $p < 0.0001$, data extracted from Archer et al. 2018, data are presented as mean values ± SD). **P** Average DNA methylation at CpG sites of the PALMD gene (Mean $M$ values of $n_{pG3myc} = 6$ CpG sites shown, two-tailed, unpaired $t$ test, $p$ value < 0.001, data are presented as mean values ± SD). pG3myc MBs show significant lower levels of methylation (two-tailed, unpaired $t$ test, $p < 0.0001$). **Q** GSEA showing the top 10 up or downregulated pathways comparing pG3myc MB to pG3/4 MB (GSEA differential expression analysis normalized enrichment score (NES), $p < 0.01$, FDR < 0.25). $n$ represents biologically independent human samples. For immunostaining, each sample was stained once.

pSHHs, we hypothesize that these factors might be further transformed to glutamate and further γ-Aminobutyric Acid (GABA), the latter both being linked to synaptic signaling[50]. In line with this, pSHHs fell into the "synaptic" profile and GABA targets were significantly upregulated in these tumors. We further speculate that pSHHs tumors might be dependent on synaptic input, a principle that has been shown for other primary brain tumors, but still has to be shown for medulloblastoma[51,52]. pSHHt MBs showed a high abundance of proteins involved in transcription/translation, DNA repair and cell cycle. In line with this, respective MB showed an increased proliferation compared to pSHHs.

*TP53*-mutated SHH cases, stratified as high-risk SHH MB[33], did not form a distinguishable cluster. However, among others, CHD6, DNAJB2 and NNMT, known to be associated with aberrant *TP53* expression and high tumor progression[53–55], showed a differential abundance comparing *TP53*-mutated to *TP53*-wildtype cases. Further, CHD6 is suggested as a potential anti-cancer target for tumors with DNA-damage repair-associated processes[55]. Mutations within the largest subunit of the elongator complex (*ELP1*) have lately been described in SHH MB[29]. These mutations were found mutually exclusive with *TP53* mutations and *ELP1* mutated SHH MBs were characterized by translational deregulation with upregulation of factors involved in transcription and translation[29]. Reanalysis of published proteome data from *ELP1* mutated SHH MB cases indeed revealed that all cases were attributed to the pSHHt MB subtype (Supplementary data 5k)[29]. As a limitation, the *ELP1* status of the SHH MB cases in our cohort was only known for $n = 3$ pSHHs and $n = 10$ pSHHt tumors (all wildtype). However, all SHH MBs with methylation subtype 3—associated with *ELP1* mutations—fell into pSHHt[24,29]. The clinical significance of the two proteome subtypes of SHH MB needs further validation in the future.

Current standard treatment approaches for MB (surgical removal, craniospinal irradiation and combinational chemotherapy) cause severe neuro-cognitive and neuroendocrine late effects. Due to their high responsiveness to therapy, WNT-type MBs are evaluated for therapy de-escalation[56]. The identification of *CTNNB1* mutations, or chromosome 6 deletion (monosomy 6) are common markers for the identification of WNT-type MB. Immunohistochemistry is used to detect nuclear ß-CATENIN staining in tumor cells that can be weak and found only a subset of cell nuclei[57,58]. Here, Tenascin C (TNC) was found elevated in pWNT MBs, in line with results of previous mRNA-based analyses[59]. *TNC* is a highly glycosylated extracellular matrix (ECM) protein, promoting or inhibiting proliferation and migration in cancer, depending on the present splice variant[60], which will be a field of further study. Besides TNC, a general enrichment of ECM proteins was detected in pWNT MBs. While the ECM has not been investigated in-depth in WNT MB, ECM components have been described to predict outcomes in MB[61]. ECM degradation was found as a hallmark of tumor invasion, metastasis development and overall bad prognosis[62]. WNT pathway activation dependent disruption of the blood-brain barrier (BBB)[62], was described to permit accumulation of high levels of intra-tumoral chemotherapy in WNT tumors, resulting in a robust therapeutic response. TNC could be another contributor to this phenotype, as high TNC levels contribute to BBB disruption[62,63]. Furthermore, other BBB contributors, such as EPLIN1, DSP and S100A4 were found differential abundant in pWNT.

In line with previous results, we found three proteome subtypes of non-WNT/non-SHH MBs[16]. pG4 (predominantly comprising G4 tumors), followed the synaptic program. These findings go in line with the literature, as synaptic signatures for G4 tumors, have been described[5,16]. In pG4 MBs, we detected a higher abundance of VEGF signalling-related proteins, previously described in the context of tumor angiogenesis. VEGF signaling can be targeted in MB using Bevacicumab or Mebendazole[64,65] and hence might be beneficial for pG4 patients. pG3 MBs (composed of both G3 and G4 tumors) showed the lowest cluster certainty and inherited the characteristics of both pG3myc and pG4. pG3myc tumors, showed a reduced survival rate and high-risk features, such as LCA histology and solid metastasis. Group 3 MB with *MYC* amplification are highly aggressive and exhibit a bad prognosis[66,67]. In our cohort, more than half of the patients showed a *CMYC* amplification, while all samples showed an upregulation of *CMYC* target genes, supporting the hypothesis that besides *CMYC* amplification, changes in its phosphorylation status result in a CMYC-driven high-risk proteome G3 subtype[16]. Therefore, proteome signatures may be additionally important for stratification of MB patients, as the current stratification scheme for high-risk MB based on (genetic) *MYC* amplification may miss these non-amplified high-risk pG3myc patients. As potential protein biomarkers for pG3myc MB, DIEXF, MDN1, POSTN, TPD52 and PALMD were selected. TPD52 has recently been suggested as an immunohistochemistry (IHC) marker for high-risk non-WNT/non-SHH patients[8]. PALMD showed the highest elevation in our cohort and was established as a suitable IHC marker for the identification of pG3myc MB. Further prospective trials need to evaluate its significance for stratification of high-risk non-WNT/non-SHH patients. Further proposed markers for proteomic MB subtypes in this study have to be tested in prospective studies to verify their potential for classification and potential therapy prediction in the future.

High-risk pG3myc MBs showed a high resemblance to pWNT tumors with favorable outcome. Comparing both groups, revealed the components of the TriC/CCT complex to be significantly different. A high abundance of CCT complex proteins has been linked to worse prognosis in cancer and was identified as a predominate driver of Vincaalcaloid resistance, including Vincristine, which is among the most frequently used drugs for MB[68]. The general low abundance of

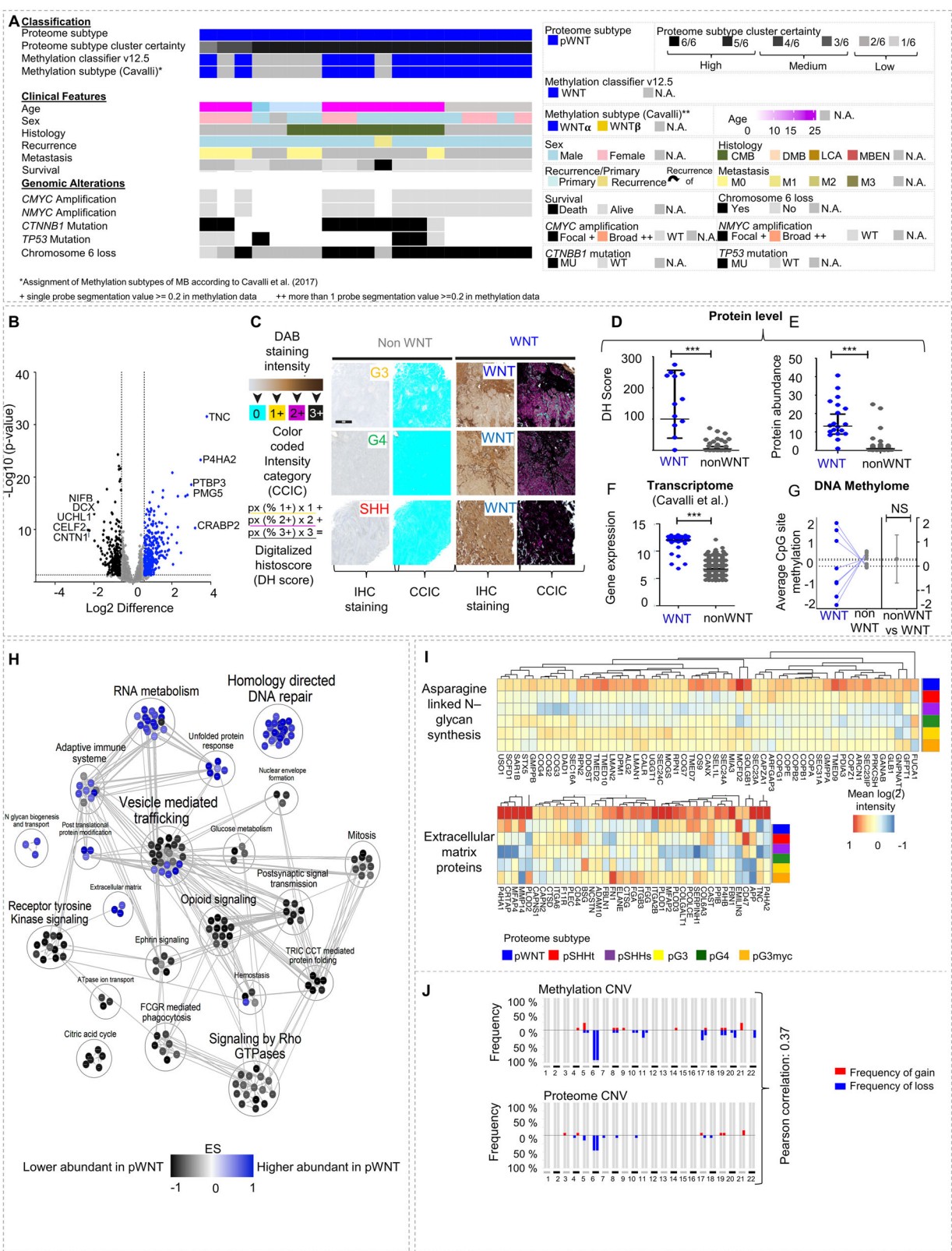

TriC/CCT proteins in pWNT MB could therefore be a BBB-phenotype-independent explanation for the relatively high response to chemotherapy[69]. The usage of CT20p, an amphipathic CCT inhibitor peptide, was described as a promising strategy for the treatment of high-risk tumors with high CCT abundance[70,71]. Based on our data, the approach should be further investigated as a potential strategy to enhance Vincristine-mediated cytotoxicity in high-risk pG3myc MBs,

which were characterized by a particularly high abundance of CCT/TriC proteins.

We further identified increased Asparagine-linked-N-glycosylation as a hallmark of WNT Medulloblastoma. Glycosylation patterns can be used as biomarkers for disease progression[19] and aberrant N-glycosylation patterns have been described for brain cancer[72]. Of note, aberrant N-glycan structures in cancer could be targeted by

**Fig. 7 | pWNT MB show high feature conservation and can be identified by Tenascin C (TNC) staining. A** Histological, molecular, and clinical characteristics of the pWNT MB subtype (n = 19). **B** Differentially abundant proteins when comparing pWNT tumors to all other proteome subtypes (two-tailed, unpaired $t$ test, $p$ value < 0.05; log2FC > 1.5). **C** Scheme and representative images of digital quantification of TNC immunostainings in MB (source data file has been provided). **D** Significantly enhanced DHS for TNC in pWNT MB ($n_{pWNT}$=9) compared to all other MB subtypes ($n_{others}$=28, two-tailed, unpaired $t$ test, $p$ < 0.0001, data are presented as mean values ± SD). **E** Protein abundance for TNC in pWNT MBs ($n_{pWNT}$ = 19) compared to all other MB subtypes ($n_{others}$=148, two-tailed, unpaired $t$ test, $p$ < 0.0001, data are presented as mean values ± SD). **F** *TNC* gene expression in WNT MBs and other MB subtypes in a published dataset of MB ($n_{WNT}$ = 70,

$n_{nonWNT}$ = 693, two-tailed, unpaired $t$ test, $p$ < 0.001, data are presented as mean values ± SD)[5]. **G** Average DNA methylation at CpG sites of the *TNC* gene (mean value for $n_{pWNT}$ = 8 CpG sites shown, two-tailed, unpaired $t$ test, $p$ = n.s., data presented as mean values ± SD). **H** MCL clustering of eEnriched gene sets, comparing pWNT to all other subtypes in GSEA. **I** Heatmaps showing mean protein abundance in MB subtypes for hallmark genesets specifically enriched in pWNT MB (GSEA differential expression analysis normalized enrichment score (NES), $NES_{Glycan}$ = 2.2, $p_{Glycan}$ = <0.001; $NES_{EMP}$ = 1.7, $p_{EMP}$ = 0.02). **J** CNV plots of pWNT MBs ($n$ = 8) were calculated from either DNA methylation or proteome data with Pearson correlation between both omic types ($r$ = 0.37). $n$ represents biologically independent human samples. For immunostaining, each sample was stained once. NS = not significant.

immunotherapy and thus provide therapeutic strategies, especially for high-risk tumors that are not sensitive to classical treatment[37,73]. As an example, chimeric-antigen-rceptor (CAR)-modified T cells, that can be specifically directed against tumor-associated carbohydrate antigens are rapidly evolving[74]. Differential, quantitative N-glycan analysis reflected proteome MB subtypes with high similarity for pSHHt and pSHHs MBs. The latter could be related to dominant SHH activation in these groups, knowingly having an impact on N-glycosylation[75]. 12 structures were identified only in high-risk pG3myc patients. Most of these structures are complex bisecting N-glycans, known to be associated with cell growth control and tumor progression[19,75] and might be related to the unfavorable outcome for pG3myc patients. pG3myc-specific N-glycans do not appear in healthy brain cells, whose N-glycome is characterized by dismissed N-glycan complexity, lack of complex N-glycans and truncated structures[76] and might serve as suitable immunotherapy targets for high-risk patients.

For pG4 patients, highest amounts of salivated N-Glycans were found, further supporting the immunological profile of pG4 MBs, observed at the proteome level[77].

Taken together, the integration of MB proteome, DNA-methylome and N-glycome data revealed (1) unique insights into MB phenotypes, (2) potential biomarkers for rapid histological subtyping and for stratification, and (3) therapeutic targets for MB. Specifically, TriC/CCT inhibitors or chimeric-antigen-receptor-modified T-cells to target tumor-specific carbohydrates may be applied for high-risk MBs. Superordinate transcription/translational or synaptic proteome profiles across subtypes further revealed targetable vulnerabilities, which may be addressed by e.g., CDK4 inhibitors or memantine.

## Methods
### Subject details
This research complies with all relevant ethical regulations. Investigations were performed in accordance with local and national ethical rules of patient's material and have, therefore, been performed in accordance with the ethical standards laid down in an appropriate version of the 1964 Declaration of Helsinki. The study protocol was approved by the Ethics Committee of the Hamburg Chamber of Physicians. All patients and parents or legal adult representatives provided informed consent in written format permitting scientific use of the data. There was no compensation provided for participation. All samples underwent anonymization.

### In house patient samples for main cohort and biological validation cohort
FFPE Medulloblastoma samples of tumors within the years 1976–2022 were obtained from tissue archives from neuropathology units in Munich (Ludwig-Maximilians-University), Heidelberg (University Hospital Heidelberg), Hannover (Hannover Medical School (MHH)), Aachen (RWTH Aachen University Hospital), Augsburg (University of Augsburg) and Hamburg (University Medical Center Hamburg-Eppendorf). Some of these samples were collected as part of the HIT-MED study, which is a registry for developing treatments in children

and adolescents with aggressive pediatric brain tumors such as Medulloblastoma and Ependymoma. The validation samples (both technical and biological validation) were a subset from all the samples collected from all the different institutions and HIT-MED. Some samples (Supplementary Data 1c, Supplementary Data 11) were part of SIOP-PNET5. SIOP-PNET5 is a clinical trial (NCT02066220) within HIT-MED, in which the primary outcomes are identification of long-term damage to disease and therapy, therapy deacceleration in low-risk patients to name a few. The PNET5 study protocol planned "comprehensive genome-wide investigations of medulloblastoma" as exploratory analyses without pre-defined methods. The present analysis was not a planned SIOP-PNET5-MB study question, but was done from archival material of PNET5-participants from the author's own institution and informed consent of the trial participants. To avoid potential interference with the analysis of SIOP-PNET5-MB trial analyses, the inclusion of these patients was discussed with the PNET5 principal investigator (Stefan Rutkowski) and the analyses were classified not to interfere with predefined SIOP-PNET5-MB study hypotheses. Included PNET5 samples were used for all the analyses in this study, but excluded from survival analysis, since this clinical trial is still not yet published. Details of the samples used in this study can be found in Supplementary Data 1c ($n$ = 6, in the main cohort) and Supplementary Data 11 ($n$ = 2, in the biological validation cohort).

Tumor samples were fixed in 4% paraformaldehyde, dehydrated, embedded in paraffin, and sectioned at 10 μm for microdissection using standard laboratory protocols. For further information on clinical details of samples, please refer to Supplementary data 1c and Supplementary Data 11 ($n_{current\ study\ main\ cohort\ with\ successful\ proteome\ subtype\ assignment}$ = 62, $n_{Forget\ et\ al.\ (PMID:\ 302050439)\ with\ successful\ proteome\ subtype\ assignment}$ = 38, $n_{Archer\ et\ al.\ (PMID:\ 30205044)\ with\ successful\ proteome\ subtype\ assignment}$ = 45, $n_{Petralia\ et\ al.\ (PMID:\ 33242424)}$ = 22, $n_{technical\ validation\ cohort}$ = 57, $n_{biological\ validation\ cohort}$ = 30). An overview of all measured protein samples can be found in Supplementary Table 3.

### Medulloblastoma cell lines
The human Medulloblastoma cell lines DAOY (Ca#HTB-186) and D283med (Ca#HTB-185) were obtained from ATCC, Manassas, VA, USA. DAOY and D283med were authenticated using Eurofins using STR-profiling analysis. UW473 was kindly provided by Michael Bobola. All lines were used as Standards for TMT batches. Cells were cultivated in DMEM (Dulbecco's Modified Eagle Medium, PAN-Biontech) supplemented with 10 % FCS at 37 °C, 5% $CO_2$.

### Publicly available datasets
For the data integration and harmonization of in-house and publicly available DNA Methylation data the following datasets were used: Archer et al.[16]: 42 FF MB samples, accessible as a subset of European Genome-phenome Archive ID: EGAS00001001953. Forget et al.[17]: 38 FF MB samples, accessible via Gene Expression Omnibus (GSE104728). For the analysis of RNA Expression data, processed and normalized data from the following datasets were used: Cavalli et al. (2017)5: 763 MB samples, accessible via Gene Expression Omnibus (GPL22286)[5]. For

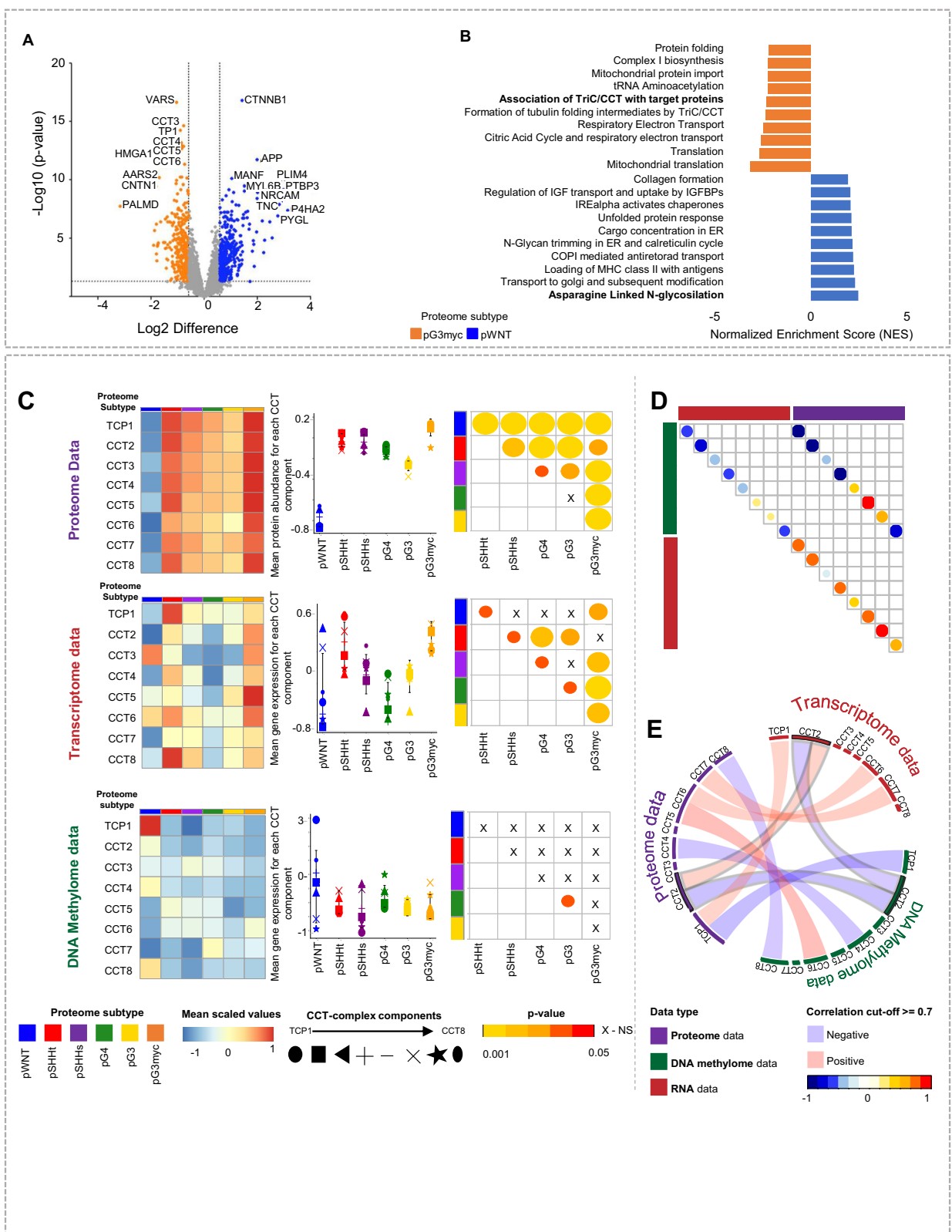

the data integration and harmonization of in-house and publicly available proteome data, the following datasets were included: Archer et al.[16]: 45 FF MB samples, available via the MassIVE online repository (MSV000082644, Tandem Mass Tag- (TMT) label-based protein quantification); Forget et al.[17]: 39 FF MB samples, available via the PRIDE archive (PXD006607, stable isotope labeling by amino acids in cell culture- (SILAC) label-based protein quantification); Petralia et al.[15],

23 FF MB samples, available through the Clinical Proteomic Tumor Analysis Consortium Data Portal (https://cptac-data-portal.georgetown.edu/cptacPublic/) and the Proteomics Data Commons (https://pdc.cancer.gov/pdc/, Tandem Mass Tag- (TMT) label-based protein quantification). For validation of determined proteome sub-types, as well as the investigation of the proteome profile of ELP1 mutated SHH MB, a dataset published by Waszak et al.[29]. was used (23

**Fig. 8 | Differential proteomics reveal low abundance of all multiprotein complex TriC/CCT components as a hallmark of pWNT MB. A** Differentially abundant proteins when comparing pWNT (*n* = 19) to pG3myc (*n* = 26) MB (two-tailed, unpaired t test, *p* value < 0.05; log2FC > 1.5). **B** GSEA showing the top 10 up or downregulated pathways comparing pG3myc MB to pWNT (GSEA differential expression analysis normalized enrichment score (NES), *p* < 0.05, FDR < 0.25). **C** Mean protein abundancies, gene expression values and methylation at CpG sites for all components of the tailless complex polypeptide 1 ring complex/Chaperonin containing tailless complex polypeptide 1 (TriC/*CCT*) per proteome subtype in matched cases ($n_{pWNT} = 4$, $n_{pSHHt} = 14$, $n_{pSHHs} = 4$, $n_{pG3} = 6$, $n_{pG4} = 17$, $n_{p3Myc} = 11$, data are presented as mean values ± SD. Left: Heatmaps. Middle: Quantification (two-tailed, unpaired *t* test). Right: *p* values when comparing subtypes ($p_{pWNTvspSHHt}$ < 0.0001, $p_{pWNTvspSHHs}$ < 0.0001, $p_{pWNTvspG3}$ < 0.0001, $p_{pWNTvspG3myc}$ < 0.0001, $p_{pWNTvspG4}$ < 0.0001, $p_{pSHHtvspSHHs}$ < 0.001, $p_{pSHHtvspG3}$ < 0.0001, $p_{pSHHtvspG3myc}$ < 0.0001, $p_{pSHHtvspG4}$ < 0.01, $p_{pSHHsvspG3}$ < 0.01, $p_{pSHHsvspG3myc}$ < 0.0001, $p_{pSHHsvspG4}$ < 0.05, $p_{pG3vsG4}$ = n.s., $p_{pG3vspG3myc}$ < 0.0001, $p_{pG4vspG3myc}$ < 0.0001). **D** Correlation plot displaying mean correlation for each component in all three omic types. **E** Circus plot displaying correlations ≥0.7 for each component's protein, gene and CpG site. Only CCT2 significantly correlated on all three levels. n represents biologically independent human samples. NS = not significant.

FF MB samples, available via the PRIDE archive (PXD016832, Data independent acquisition label free protein quantification).

## Sample preparation and data acquisition

**DNA methylation profiling.** DNA methylation data was generated from FFPE tissue samples. DNA was isolated using the ReliaPrep™ FFPE gDNA Miniprep system (Promega) following the manufacturer's instructions. 100–500 ng DNA was used for bisulfite conversation using the EZ DNA Methylation Kit (Zymo Research). Then the DNA Clean & Concentrator-5 (Zymo Research) and the Infinium HD FFPE DNA Restore Kit (Illumina) were applied. Infinium BeadChip array (EPIC) using manufacturer's instructions was then used to quantify the methylation status of CpG sites on an iScan (Illumina, San Diego, USA). Data has been deposited using accession numbers GSE222478 and GSE243768 (linked to Series GSE243796). Additionally, previously published data measured on Infinium Human Methylation 450 Bead-Chip array (450 K) were included from EGAS00001001953[16] from GSE104728[17], and GSE130051[78].

**Proteome profiling (main cohort, FFPE samples).** FFPE MB tissue sections were deparaffinized with N-heptane for 10 min and centrifuged for 10 min at 14,000 *g*. The supernatant was discarded. Proteins were extracted in 0.1 M triethyl ammonium bicarbonate buffer (TEAB) with 1% sodium deoxycholate (SDC) at 99 °C for 1 h. Sonication was performed for ten pulses at 30% power, to degrade DNA, using a PowerPac™ HC High-Current power supply (Biorad Laboratories, Hercules, USA)) probe sonicator. For cell lines, proteins were extracted in 0.1 M TEAB with 1% SDC at 99 °C for 5 min. Sonication was performed for six pulses.

The protein concentration of denatured proteins was determined by the Pierce BCA Protein assay kit (Thermo Fischer Scientific, Waltham, USA), following the manufacturer's instructions. 60 µg of protein for each tissue lysate and 30 µg protein for each cell lysate were used for tryptic digestion. Disulfide bonds were reduced, using 10 mM dithiothreitol (DTT) for 30 min at 60 °C. Alkylation was achieved with 20 mM iodoacetamide (IAA) for 30 min at 37 °C in the dark. Tryptic digestion was performed at a trypsin:protein ratio of 1:100 overnight at 37 °C and stopped by adding 1% formic acid (FA). Centrifugation was performed for 10 min at 14,000 *g* to pellet precipitated SDC. The supernatant was dried in a vacuum concentrator (SpeedVac SC110 Savant, (Thermo Fisher Scientific, Bremen, Germany)) and stored at −80 °C until further analysis.

For the main cohort, 50 µg sample per patient and internal reference, TMT-10 plex labeling (Thermo Fischer Scientific, Waltham, USA), was performed, following the manufacturer's instruction. All 70 patients were run in 8 total TMT 10-plexes. Sample assignment to batches was performed in a semi-randomized manner, according to the four main molecular subtypes. In each batch, 1–2 internal reference samples were included, composed of equal amounts of peptide material from all 70 samples and cell lines. Isobarically labeled peptides were combined and fractionated, using high pH reversed-phase chromatography (ProSwift™ RP-4H, Thermo Fischer Scientific Bremen, Germany) on an HPLC system (Agilent 12000 series, Agilent

Technologies, Santa Crara, USA). Separation was performed using buffer A (10 mM ammonium hydrogen carbonate ($NH_4HCO_3$) in$H_2O$) and buffer B (10 mM $NH_4HCO_3$ in ACN) within a 25-min gradient, linearly increasing from 3 to 35% buffer B at a flow rate of 200 nl/min. In total, 13 fractions were collected for each batch, dried in a vacuum concentrator, resuspended in 0.1% FA to a final concentration of 1 mg/ml and subjected to high pH liquid chromatography coupled mass spectrometry (LC-MS). All LC-MS measurements were performed on a UPLC system (Dionex Ultimate 3000, Thermo Fisher Scientific, Bremen, Germany, trapping column: Acclaim PepMap 100 C18 trap ((100 µm × 2 cm, 100 Å pore size,5 µm particle size); Thermo Fisher Scientific, Bremen, Germany), analytical column: Acclaim PepMap 100 C18 analytical column ((75 µm × 50 cm, 100 Å pore size, 2 µm particle size); Thermo Fisher Scientific, Bremen, Germany)), coupled to an quadrupole-orbitrap-iontrap mass spectrometer (Orbitrap Fusion, Thermo Fisher Scientific, Bremen, Germany). Separation was performed using buffer A (0.1% FA in H20) and buffer B (0.1% FA in H20) within a 60-min gradient, linearly increasing from 2-30% buffer B at a flow rate of 300 nl/min. Eluting peptides were analyzed, using a DDA-based MS3 method with synchronous precursor selection (SPS), as described by McAlister et al. [79]. For MS−raw data please refer to the PRIDE archive (PXD039319).

**Proteome profiling for biological and technical validation cohort.** The deparaffinization and quantification were conducted as previously described.

20 µg of the provided samples were dissolved to a concentration of 70% ACN. 2 µL carboxylate modified magnetic beads (GE Healthcare Sera-Mag™, Chicago, USA) at 1:1 (hydrophilic/hydrophobic) in methanol were added following the SP3-protocol workflow[80]. Samples were shaken at 1400 rpm for 18 min RT and the supernatant was removed. Beads were washed two times with 100% ACN and two times with 70% EtOH. After resuspension in 50 mM ammonium bicarbonate, disulfide bonds were reduced in 10 mM DTT for 30 min, alkylated in the presence of 20 mM IAA for 30 min in the dark and digested with trypsin (sequencing grade, Promega) at 1:100 (enzyme:protein) at 37 °C overnight while shaking at 1400 rpm. Peptides were bound in 95% ACN and shaken at 1400 rpm for 10 min RT. The supernatant was and the beads were again two times with 100% ACN. Elution of peptides was performed with 20 µL 2% DMSO in 1% formic acid (FA). Samples were dried in a vacuum centrifuge and stored at −20 °C until further use.

For the measurement samples were resuspended in 0.1% FA to a final concentration of 1 mg/ml and measured on either a Quadrupole Orbitrap hybrid mass spectrometer (QExactive, Thermo Fisher Scientific) or on a quadrupole-ion-trap-orbitrap MS (Orbitrap Fusion, Thermo Fisher) in orbitrap-orbitrap configuration. For MS−raw data please refer to the PRIDE archive (PXD048767.).

**Quadrupole Orbitrap hybrid mass spectrometer set-up.** Chromatographic separation of peptides was achieved by nano UPLC (nanoAcquity system, Waters) with a two-buffer system (buffer A: 0.1% FA in water, buffer B: 0.1% FA in ACN). Attached to the UPLC was

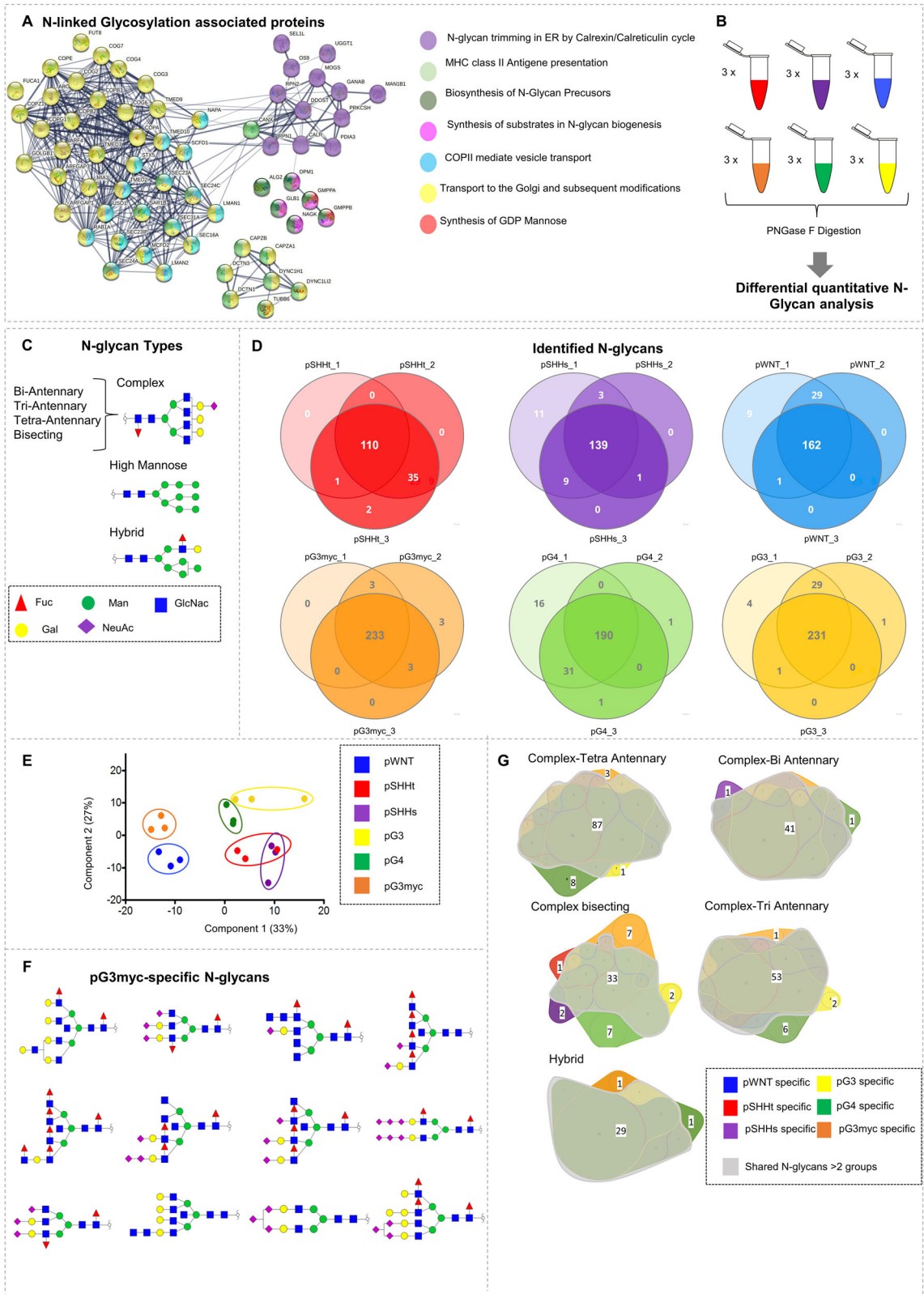

**Fig. 9 | N-glycan analysis reveals significant differences across N-glycan profiles of proteomic MB subtypes. A** STRING network analyses of differentially abundant proteins involved in N-linked glycosylation. **B** Scheme of N-glycan analyses. **C** Schematic visualization of N-glycan types. **D** Venn diagrams showing overlap of identified glycans per MB proteome subtype ($n_{pWNT} = 3$, $n_{pSHHt} = 3$, $n_{pSHHs} = 3$, $n_{pG3} = 3$, $n_{pG3myc} = 3$, $n_{pG4} = 3$). **E** PCA, based on N-glycan abundances, illustrating the separation of proteome MB subtypes at the N-glycan level. **F** 2D Structure visualization for pG3myc-specific N-glycans. GlcNAc N-Acetylglucosamine, Gal Galactose, Fuc Fucose, ManNAc N-Acetylmannosamine; Neu5AC N-Acetylneuraminic acid. **G** Venn Diagram, comparing the identified hybrid-type and complex N-glycans between proteome subtypes. n represents biologically independent human samples.

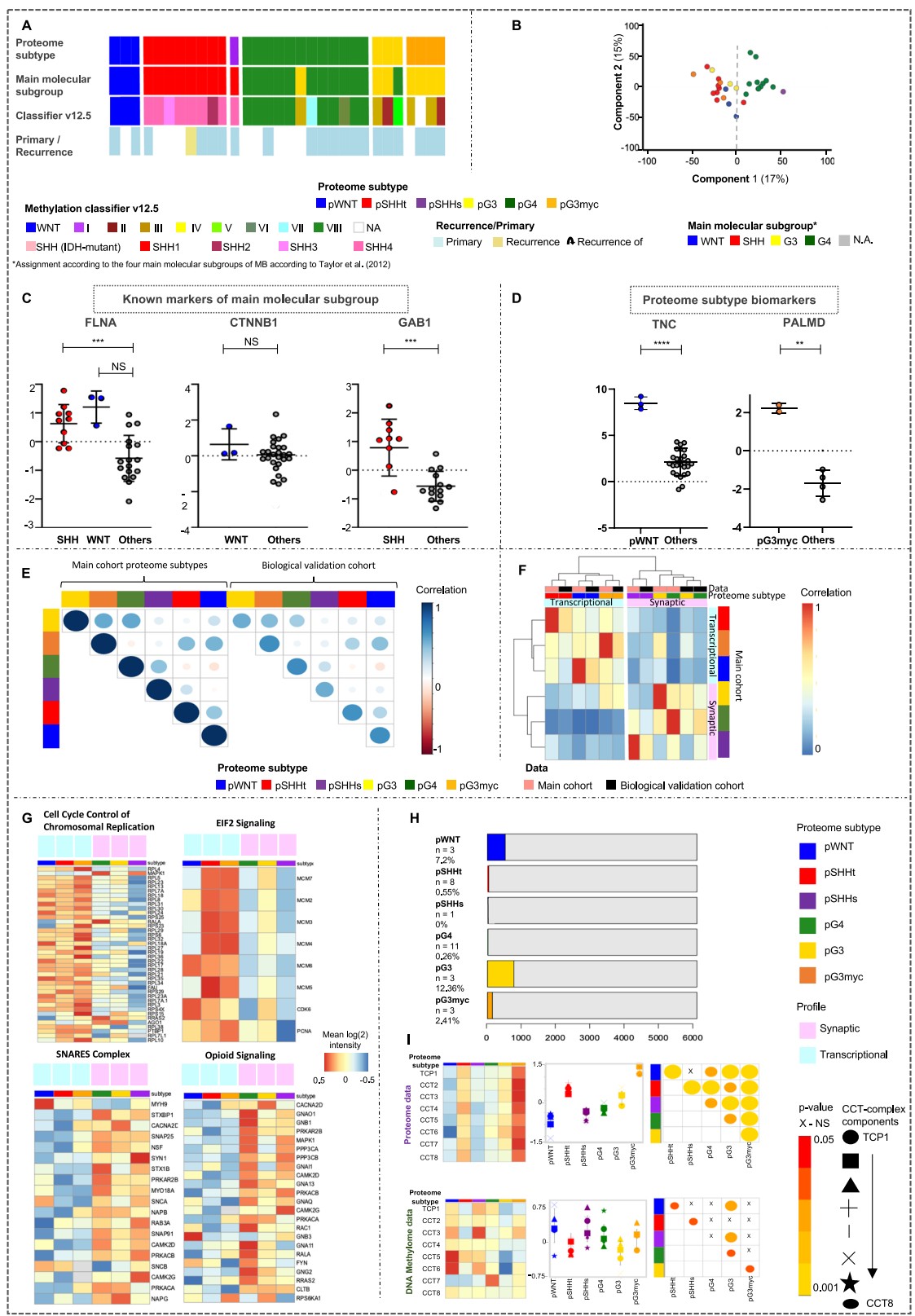

a peptide trap (100 μm × 20 mm, 100 Å pore size, 5 μm particle size, Acclaim PepMap 100 C18 trap, Thermo Fisher Scientific) for online desalting and purification followed by a 25-cm C18 reversed-phase column (75 μm × 200 mm, 130 Å pore size, 1.7 μm particle size, Peptide BEH C18, Waters). Peptides were separated using an 80-min gradient with linearly increasing ACN concentration from 2% to 30% ACN in 65 min. The eluting peptides were analyzed on a Quadrupole

Orbitrap hybrid mass spectrometer (QExactive, Thermo Fisher Scientific). Here, the ions being responsible for the 15 highest signal intensities per precursor scan ($1 \times 10^6$ ions, 70,000 Resolution, 240 ms fill time) were analyzed by MS/MS (HCD at 25 normalized collision energy, $1 \times 10^5$ ions, 17,500 Resolution, 50 ms fill time) in a range of 400–1200 m/z. A dynamic precursor exclusion of 20 s was used.

**Fig. 10 | Confirmation of proteome subtypes and differential feature conservation in an independent biological FFPE dataset. A** Clinical sample information with proteome subtype assignments using ACF based classification[94] (**B**) PCA, based on proteins found in ≥70% samples, illustrating the separation of proteome MB subtypes (source data file has been provided). **C** Protein abundances of established biomarkers WNT and SHH biomarker FLNA ($n_{WNT} = 3$, $n_{SHH} = 9$, $n_{Others} = 18$, two-tailed, unpaired t-test, $p_{pWNTvsothers} = NS$, $p_{pSHHvsothers} < 0.001$), WNT biomarker CTNNB1 ($n_{WNT} = 3$, $n_{Others} = 27$, two-tailed, unpaired t-test, $p_{pWNTvsothers} = NS.$) and SHH biomarker GAB1 ($n_{SHH} = 9$, $n_{Others} = 21$, two-tailed, unpaired $t$ test, $p_{pSHHvsothers} < 0.001$, data are represented as mean values ± SD). **D** Significant higher abundance of TNC ($n_{pWNT} = 3$, $n_{Others} = 27$, two-tailed, unpaired t-test, $p_{pWNTvsothers} < 0.0001$) and PALMD ($n_{pG3myc} = 3$, $n_{pOthers} = 27$, two-tailed, unpaired $t$ test, $p_{pG3mycvsothers} < 0.01$) in pWNT and the pG3myc subtype, respectively. Data are represented as mean values ± SD. **E** Correlation plot displaying mean Pearson correlation per subtype between the integrated cohort and the biological validation cohort. **F** Hierarchical clustering of biological validation cohort samples with samples from the main cohort (Pearson correlation and ward.D2 linkage). **G** Heatmaps showing mean protein abundance for the top hit gene sets enriched in the transcriptional (top) and synaptic profile (bottom). **H** Bar plot displaying proteome subtype-specific Pearson correlation calculated for matched samples between proteins and CpG sites ($r > 0.7$, $n = 29$, total number of samples having both DNA methylome and proteome data, 5880 proteins and 549,089 CpG sites). The number of proteins correlating with CpG site of their own gene are shown in color. **I** Left: Heatmaps for Mean protein abundances, gene expression values and methylation at CpG sites for all components of the tailless complex polypeptide 1 ring complex/Chaperonin containing tailless complex polypeptide 1 (TriC/*CCT*) per proteome subtype in matched cases ($n = 29$, $n_{pWNT} = 3$, $n_{pSHHt} = 8$, $n_{pSHHs} = 2$, $n_{pG3} = 3$, $n_{pG3myc} = 3$, $n_{pG4} = 11$) samples having both DNA methylome and proteome data). Middle: Quantification (two-tailed, unpaired $t$ test, data are presented as mean values ± SD) Right: $p$ values when comparing subtypes ($p_{pWNTvspSHHt} < 0.0001$, $p_{pWNTvspSHHs} = NS$, $p_{pWNTvspG3} < 0.0001$, $p_{pWNTvspG3myc} < 0.0001$, $p_{pWNTvspG4} < 0.001$, $p_{pSHHtvspSHHs} < 0.001$, $p_{pSHHtvspG3} < 0.001$, $p_{pSHHtvspG3myc} < 0.0001$, $p_{pSHHtvspG4} < 0.01$, $p_{pSHHsvspG3} < 0.001$, $p_{pSHHsvspG3myc} < 0.0001$, $p_{pSHHsvspG4} < 0.05$, $p_{pG3vsG4} < 0.01$, $p_{pG3vspG3myc} < 0.0001$, $p_{pG4vspG3myc} < 0.0001$). $n$ represents biologically independent human samples. NS = not significant.

## Quadrupole-ion-trap-orbitrap mass spectrometer set-up.

Chromatographic separation of peptides was achieved with a two-buffer system (buffer A: 0.1 % FA in water, buffer B: 0.1% FA in ACN). Attached to the UPLC was a peptide trap (100 μm × 200 mm, 100 Å pore size, 5 μm particle size, Acclaim PepMap 100 C18 trap, Thermo Fisher Scientific) for online desalting and purification followed by a 25 cm C18 reversed-phase column (75 μm × 250 mm, 130 Å pore size, 1.7 μm particle size, Peptide BEH C18, Waters). Peptides were separated using an 80-min gradient with linearly increasing ACN concentration from 2% to 30% ACN in 65 min. Eluting peptides were ionized using a nano-electrospray ionization source (nano-ESI) with a spray voltage of 1800, transferred into the MS, and analyzed in data-dependent acquisition (DDA) mode. For each MS1 scan, ions were accumulated for a maximum of 240 ms or until a charge density of $1 \times 10^6$ ions (AGC Target) was reached. Fourier-transformation-based mass analysis of the data from the orbitrap mass analyzer was performed, covering a mass range of 400–1200 m/z with a resolution 60,000. Peptides with charge states between 2+ and 5+ above an intensity threshold of $1 \times 10^5$ were isolated within a 2 m/z isolation window from each precursor scan and fragmented with a normalized collision energy of 25% using higher energy collisional dissociation (HCD). MS2 scanning was performed at a resolution of 17,500 on the quadrupole-ion-trap-orbitrap MS in orbitrap-orbitrap configuration, covering a mass range from 100 m/z and accumulated for 50 ms or to an AGC target of $1 \times 10^5$. Already fragmented peptides were excluded for 15 s.

## Histology and Immunohistochemistry

FFPE tissue samples were sectioned into 2 μm thick slices, according to standard laboratory protocols. Immunohistochemical stainings were performed on an automated staining machine (Ventana BenchMark XT, Roche Diagnostics, Mannheim, Germany). The following primary antibodies were used: ALDH1A3 (NBP2-15339, Novus Biologicals, 1:1000), c-myc (Z2734RL, Zeta Corporation, 1:25), TENASCIN C (SAB4200782, Sigma-Aldrich, 1:1000), PALMD (NBP2-55156, Novus Biologicals, 1:750). Further information on the antibodies and staining program can be found in Supplementary Table 1.

## Transcriptome profiling

Maxwell RSC RNA FFPE Kit was used to isolate RNA from 10 × 10 μm sections of FFPE tissue (PROMEGA Maxwell RSC RNA FFPE kit). RNA 6000 Nano Chip on an Agilent 2100 Bioanalyzer (Agilent Technologies) was used to analyse RNA integrity. From 400 ng total per sample, ribosomal RNA was depleted with the help of the RiboCop rRNA Depletion Kit (Lexogen) followed by RNA sequencing library generation using the CORALL Total RNA-Seq Library Prep Kit (Lexogen), followed by the Lexogen CORALL total RNA-Seq V2 Library Prep Kit with UDIs (according to manufacture protocol, short insert size version). Illumina NextSeq2000 machine using the P3 Reagents/100 cycle kit as paired-end sequencing 2 × 57 bp (+2× index read 12 bp). Data have been deposited under accession number GSE243795.

## Metabolic and amino acid profiling

13C-Labeled Metabolite Yeast Extract (Catalog No. ISO-1, ISOtopic solutions e.U.) LOT: 20211007 and Canonical Amino Acid Mix (Catalog No. MSK-CAA-1, Cambridge Isotope Laboratories, Inc. (CIL)) were prepared according to instructions. Tissue sections of sSHH and tSHH medulloblastoma samples were deparaffinized by two 5 min washes in xylene. 20 μL of 13C-Labeled Metabolite Yeast Extract and 1 μL of diluted 0.1 M Canonical Amino Acid Mix were added, and samples were then homogenized in 180 μL water using the TissueLyser (Qiagen N.V., Netherlands) at 20 Hz for 2 min. Afterwards, protein precipitation and metabolite extraction were achieved by adding ice-cold methanol twice (800 μL and 400 μL) and 80% methanol (200 μL). The supernatant was combined and dried in a vacuum concentrator centrifuge, and stored at −20 °C until further use.

Polar and polar ionic metabolites were analyzed by single ion monitoring (SIM) mass spectrometry coupled to ion chromatography and IC-SIM-MS raw data processing was performed as described by van Pijkeren and Egger et al. [81]. using a quadrupole orbitrap mass spectrometer (Exploris 480, Thermo Fisher Scientific) and an ICS-6000 (Thermo Fisher Scientific).

Amino acids were analyzed by multiple reaction monitoring (MRM) mass spectrometry using a triple quadrupole mass spectrometer coupled to ultra-high performance liquid chromatography (UPLC). Amino acids were separated using an Acquity Premier UPLC system (Waters) equipped with an Atlantis Premier BEH C18 AX column (1.7 μm, 2.1 × 150mm, Waters) heated to 45 °C. A gradient of mobile phase A (water, 0.1% formic acid (FA)) and mobile phase B (acetonitrile, 0.1% FA) was applied as followed: 1% B at 0.350 mL/min for 1 min, to 20% B in 1 min at 0.350 mL/min, to 40% B in 0.5 min at 0.350 mL/min, to 95% B in 1.5 min at 0.450 mL/min, hold for 0.5 min, for re-equilibration, switch to 1% B in 0.1 min at 0.450 mL/min, hold for 0.1 min at 0.450 mL/min and hold for 1.3 min at 0.350 mL/min. Samples were measured on a Xevo-TQ XS Mass spectrometer (Waters) equipped with an electrospray ionization source operated in positive ion mode. The mass spectrometer was operated in multiple reaction monitoring (MRM) mode using individual cone and collision voltages for each amino acid and its internal standard (Supplementary data 1). Raw files were analyzed by MS Quan in Waters Connect (Waters, V1.7.0.7). Details on MRM settings per metabolite and internal standard can be found in Supplementary Table 2.

For MS raw data of the metabolites and amino acids please refer to MetaboLights repository[82] MTBLS9830 and MTBLS9836, respectively.

## N-Glycan profiling

100 μg of protein for 18 samples was denatured, reduced, and alkylated as described above. Samples were concentrated by 3 kDa Amicon Ultra centrifugal filters (Merck Millipore, R0NB30416) with 100 mM $NH_4HCO_3$ to exchange the buffer and retain globular particles above 3 kDa. Thirty units of PNGase F were added to each sample and incubated in a 37 °C Thermomixer for 24 h. After PNGase F digestion, purified N-glycans were eluted by Sep-Pak C18 cartridges (Water, WAT023590) with 5% acetic acid and dried in a speed vacuum. The purified N-glycans were then permethylated using an optimized solid-phase permethylation method and analyzed via LC-MS measurement as mentioned here[83]. Glycan data has been deposited at GlycoPOST[84] with the identifier GPST000414.

## Raw data processing

**Processing of DNA methylation array data.** Idat files generated using the above protocol were processed in R (Version 4.0.5). The files were read using the minfi package (Version 1.36.0)[85]. Differentially methylated probes/CpG sites were found using the limma package (Version 3.46.0)[86], corrected for multiple testing using Benjamini Hochberg (cut-off 5% FDR). M-values of 10,000 differentially methylated CpG sites which could cluster subtypes based on biological differences were selected for further analysis. Similarly, DMR analysis was performed using DMRcate package (V4.30.0). For DMR analysis, we set a min of 10 CpGs per DMR (<1000 nt from each other) to minimize gene overlap, which resulted in ~9000 DMRs with each DMR having 10–200 CpGs.

## Processing of Proteome raw data for main cohort

**Processing of Proteome raw data for the integrated cohort.** Obtained raw data from in-house generated and publicly available (Archer et al. [16], TMT 10-Plex; Petralia et al. [15], TMT 11-Plex). TMT-based LC-MS measurements were processed with the Andromeda algorithm, implemented in the MaxQuant software (Max Plank Institute for Biochemistry, Version 1.6.2.10)[87] and searched against a reviewed human database (downloaded from Uniprot February 2019, 26,659 entries).). The Carboxymethylation of cysteine residues was set as a fixed modification. Methionine oxidation, N-terminal protein acetylation and the conversion of glutamine to pyroglutamate were set as variable modifications. Peptides with a minimum length of 6 amino acids and a maximum mass of 6000 Da were considered. The mass tolerance was set to 10 ppm. The maximum number of allowed missed cleavages in tryptic digestion was two. A false discovery rate (FDR) value threshold <0.01, using a reverted decoy peptide databases approach, was set for peptide identification. Quantification was performed, based on TMT reporter intensities at MS3 level for LC-MS3 in-house data and at MS2 level for LC-MS2 data, acquired by Archer et al. [16] and Petralia et al. [15]. All studies were searched separately. Fractions for each TMT batch were searched jointly.

For stable isotope labeling by amino acids in cell culture (super-SILAC) data, acquired by Forget et al. [17], log2 transformed SILAC ratios were directly obtained from the MassIVE online repository (MSV000082644).

For the external validation the dataset published by Waszak et al. [29]. was used. The DIA raw data spectra were downloaded from PRIDE and processed using Data Independent Acquisition with Neural Networks (DIA-NN, version 1.8.1)[88]. The spectra were searched against a peer-reviewed human FASTA database (downloaded from UniProt April 2020, 20,365 entries). A spectral library was generated in silico by DIA-NN using the same FASTA database. Smart profiling was enabled for library generation. Methionine oxidation, carboxymethylation of cysteine residues as well as N-terminal methionine excision were set as

variable modifications. The maximum number of variable modifications was set to three, the maximum number of missed cleavages was two. The peptide length range was set from 7 to 30. Mass accuracy, MS1 accuracy, and the scan window were optimized by DIA-NN. An FDR < 0.01 was applied at the precursor level—decoys were generated by mutating target precursors' amino acids adjacent to the peptide termini. Interference removal from fragment elution curves as well as normalization were disabled. Neural network classifier was set to single-pass mode and the fixed-width center of each elution peak was used for quantification.

**Processing of the biological and technical validation cohorts.** The spectra were searched with the Sequest algorithm integrated in the Proteome Discoverer software (v 3.0.0.757), Thermo Fisher Scientific) against a reviewed human database (downloaded from Uniprot in June 2021, Containing 20,683 entries)). Carbamidomethylation was set as fixed modification for cysteine residues and the oxidation of methionine, and pyro-glutamate formation at glutamine residues at the peptide N-terminus, as well as acetylation of the protein N-terminus were allowed as variable modifications. A maximum number of 2 missing tryptic cleavages was set. Peptides between 6 and 144 amino acids where considered. A strict cutoff (FDR < 0.01) was set for peptide and protein identification. Quantification was performed using the Minora Algorithm, implemented in Proteome discoverer.

## Processing of N-Glycan raw data

N-Glycan raw data were opened with Xcalibur Qual Browser (Version No 4.2.28.14). MaxQuant were used for extracting all the detected masses and $m/z$ from MS raw data of permethylated reducing N-glycans. An in-house Python-script was used to extract and calculate monosaccharide compositions based on the molecular weight of each derivatized N-glycan[89]. The N-glycan structures were identified, matched to N-glycan compositions and quantified using the Xcalibur, Glycoworkbench 2.1 and Skyline software (Version No 21.1.0.278)[83]. Further statistical analysis was performed with the Perseus software.

## Processing of raw transcriptome data

Raw fastq files of human samples were processed in usegalaxy.eu[90]. Low quality reads were detected using *FastQC* (Galaxy Version 0.73+galaxy0), and *Trimmomatic* (Galaxy Version 0.38.1) was used for trimming poor quality reads (reads with average quality <20). Reads were aligned to the GRh38 human reference genome using *STAR aligner* (Galaxy Version 2.7.8a+galaxy1). Gene expression was quantified with *featureCounts* (Galaxy Version 2.0.1+galaxy2) and VST-normalized files were generated by *DEseq2* (Galaxy Version 2.11.40.7+galaxy2). Further processing of data was performed with R (v4.2.1). Transcriptome data was combined with publicly available transcriptome data[16]. Batch corrected with HarmonizR[26].

## Processing of DNA methylation array data

Raw signal intensities for EPIC and 450 K files were read individually. Since ~93% of the loci of 450 K array are also present on EPIC array, they can be combined using minfi's combineArrays(). After combining the two arrays they can be output as a virtual array. In this study, 450 K array was the output virtual array since a greater number of samples were measured on 450 K.

The detection $P$ value was used to identify sample quality and filter out bad quality samples (none were excluded, $n = 0$). Further, probes having bad quality ($n = 49,091$), probes with single nucleotide polymorphism ($n = 12,868$) and probes present on X and Y chromosomes ($n = 8777$) were filtered out. After normalization and probe filtering, the m-values log2(M/U) where methylation intensity is denoted by M and unmethylation intensity denoted by U were used for further analysis.

## Data normalization and integration

**Normalization and integration of DNA methylation array data.**
Single-sample noob normalization (ssNoob) was performed since we combined samples from different arrays (EPIC and 450 K). The detailed method development has been mentioned[91,92].

**Normalization and integration of Proteome data.** Prior to data integration, protein abundances were handled separately for each dataset. TMT reporter intensities were log2 transformed and median normalized across columns. Technical variances between TMT batches were corrected, using HarmonizR framework (Version 0.0.0.9). As described here[26], mean subtraction across rows was applied to batch-effect corrected TMT reporter intensities to mimic SILAC ratios, prior to data integration. Log2 transformed super SILAC ratios were median normalized across columns prior to data integration.

Processed data from individual studies was combined based on the UniProt identifier, data harmonization was performed as described above. Combined, harmonized protein abundances were mean-scaled across rows. Out of 176 analyzed cases, 9 patients were excluded from further analysis, as high blood protein yields, suppressing tumor-specific signals, were detected from LC-MS/MS measurements (Supplementary data 1a).

For the external validation cohort protein abundances were log2 transformed and median normalized across columns. Samples were assigned to proteome subtypes individually. Protein abundances were reduced to the 3998 proteins, considered in the main cohort. Harmonized protein abundances from the main cohort were integrated with each individual sample. Mean row normalization was performed to adjust values from validation samples to the main cohort. Pearson correlation-based hierarchical clustering, with average linkage was applied using the Perseus software (Max Plank Institute for Biochemistry, Version 1.5.8.5)[93].

For biological and technical validation cohort the data was processed and harmonized as described above. For the biological validation, one sample had to be excluded due to high blood protein yields as described above. The proteome subtypes for the biological validation were assigned via the ACF classifier[94]. The proteome subtypes for the technical validation were taken from the main cohort. Protein abundances were treated as above.

**Normalization of N-Glycan data.** N-Glycan intensities were log2 transformed and median normalized across columns to compensate for injection amount variations.

## Quantification and statistical analysis

**Dimensionality reduction and hierarchical clustering.** Nonlinear Iterative vertical Least Squares (NIPALS) PCA and hierarchical clustering were performed in the R software environment (version 4.1.3). For Principal component calculation and visualization, the mixOmics package (Version 6.19.4.)[31] was used in Bioconductor (version 3.14). Hierarchical clustering was performed based on pheatmap package (version 1.0.12) and ComplexHeatmap (Version 2.6.2)[95]. Pearson correlation was applied as a distance metric. Ward.D linkage was used. Pairwise complete correlation was used, to enable the consideration of missing values.

**Consensus clustering.** To determine the ideal number of clusters from proteome and DNA-methylation data, Consensus Clustering was applied on normalized and integrated datasets, using the ConsensusClusterPlus package (Version 1.6)[96], in the R software environment (version 4.1.3). In correspondence with the current maximum number of suspected MB subtypes, the number of clusters was varied from 2 to 12 and calculated with 1000 subsamples for all combinations of two clustering methods (Hierarchical clustering (HC) and partition around medoids (PAM)) and three distance metrics (Euclidean, Spearman, Pearson). The Ward's method was applied for linkage. Missing value tolerant pairwise complete correlation was used, to enable the consideration of missing values. For each sample, the cluster certainty was calculated by how many times under the application of different distance metrics (Euclidean, Spearman, Pearson) and clustering approaches (k-medoids, hierarchical clustering) a sample was associated with a certain cluster, while allowing a total number of six clusters.

## Differential analysis and visualization

Statistical testing was carried out, using the Perseus software[93]. ANOVA testing was performed for the comparison across multiple subgroups/subtypes. Factors, identified with $p$ value < 0.05 were considered statistically significant differential abundant across groups. For the identification of subtype-specific biomarkers, Students t-testing was applied ($p$ value < 0.05, Foldchange difference > 1.5). Visualization of t-test results and abundance distributions across groups was performed in PRISM (GraphPad, Version 5) and Microsoft excel (Version 16.5.).

## Functional annotation of data sets

REACTOME- based[97] Gene Set Enrichment Analysis was performed by using the GSEA software (version 4.1, Broad Institute, San Diego, CA, USA)[98]. 1000 permutations were used. Permutation was performed based on gene sets. A weighted enrichment statistic was applied, using the signal-to-noise ratio as a metric for ranking genes. No additional normalization was applied within GSEA. As in default mode, gene sets smaller than 15 and bigger than 500 genes were excluded from analysis. For visualization of GSEA results, the EnrichmentMap (version 3.3)[99] application within the Cytoscape environment (version 3.8.2)[100] was used. Gene sets were considered if they were identified at an FDR < 0.25 and a $p$ value < 0.1. For gene-set-similarity filtering, data set edges were set automatically. A combined Jaccard and Overlap metric was used, applying a cutoff of 0.375. For gene set clustering, AutoAnnotate (version 1.3)[99] was used, using the Markov cluster algorithm (MCL). The gene-set-similarity coefficient was utilized for edge weighting.

## Survival curves

Kaplan-Meier curves were generated for the overall survival of 121 patients. All Kaplan-Meier curves and log-rank test $p$ values were generated with PRISM (GraphPad, Version 5). A conservative log-rank test (Mantel-Cox) was used for the comparison of survival curves. A significant difference between curves was assumed at a $p$ value < 0.05.

## Copy number frequency plots of Proteome and DNA Methylome data

Copy number analysis was performed on samples having both methylation and proteomic data ($N = 115$). Samples from 450 K and EPIC array were read in separately as mentioned above. Data were read using read.metharray.sheet() and read.metharray.exp() using the minfiData package (Version 0.36.0)[85]. For normalization, preprocessIllumina normalization using MsetEx data containing control samples for normalization of 450 K array data, while for EPIC array data minfidataEPIC (Version1.16.0)[85] was used. IlluminaHumanMethylation450kanno.ilmn12.hg19 and IlluminaHumanMethylationEPICanno.ilm10b4.hg19 were used to generate the annotation files of 450 K and EPIC array data respectively.

Individual sample CNV plots were generated as mentioned in the Conumee package (Version 1.24.0) vignette, and the segmentation information from each sample was saved and used later for generation of cumulative CNV plot using CNAppWeb tool[101](cut-off> = |0.2|) for gain or loss). The segmentation information for all samples belonging to one subtype were combined into a single file in subgroup specific manner and then read into CNAppWeb tool.

Combining the segmentation information from proteome data and methylome data in subgroup specific manner, Pearson correlation-based distance plot was generated.

To map the protein abundancies to each of the chromosomes, protein names were converted to their respective gene names and a column containing mapping information for these genes was added. Copynumber (Version 1.30.0) package in R was used to generate segmentation information for these proteins. CNAppWeb tool using the cut-off mentioned above was used to map the protein abundancies to respective chromosomes.

### Integration of proteome and DNA methylome data

DIABLO from mixOmics (Version 6.19.4)[31] was used for integration of proteome and methylome data to correlate the two data types. Proteome data (3990 proteins,115 samples) and methylome data (10,000 differentially methylated CpG sites, 115 proteins) were pre-processed as mentioned above. Steps followed were same as explained in the mixOmics vignette. Briefly, datasets were integrated, an output variable containing information about which subgroup the samples belong to was also supplied. Each data set is broken down into components (5 components for this study) or latent variables which are associated with the data. Components were selected using fivefold cross validation repeated 50 times and since the groups were imbalanced lowest overall error rate and centroid distance was used. For each dataset and for each component sparse DIABLO was applied which will select variables contributing maximally to the selected component. sPLS-DA was applied to the selected variables to generate the correlation circus plot (cut-off 0.7) which gives the variables that are either positively or negatively correlating with each other. DMRs between each methylome subtype was found in a pairwise manner, corrected for multiple testing using Benjamini Hochberg (cut-off 5% FDR) and integrated with proteome data in mixOmics.

### Global correlation of proteome and DNA methylation data

To check for overall correlation between the two datasets, subgroup-specific (pWNT = 13, pSHHt = 29, pSHHs = 6, pG4 = 36, pG3 = 11, pG3Myc = 20) pearson correlation (cut-off 0.7) was performed between the proteome (3990 proteins and 115 samples) and methylome (381,717 probes and 115 samples) in R (Version 4.0.5. The data was subsetted for correlation value ≥ 0.7 and matches of proteins to their respective probes using Python script in anaconda JupyterLab (Version 3.0.14). Non-subgroup specific pearson correlation between the proteome and methylome data was similarly performed with focus on potential biomarkers for each subgroup and their correlation with methylation probes. Scatterplots of biomarker's protein abundance and the M-values of CpG sites of its own gene (crossing the pearson correlation cut-off of 0.7) were plotted to confirm the correlations. For correlating DMRs and proteins, mean of all CpG sites belonging to each DMR was taken to find correlation between all DMRs and proteins

For correlation of CCT complex components, all samples for which we had all three datasets were considered (n = 60) and Pearson correlation ≥0.7 was plotted using circlize(Version 0.4.15) and corrplot (Version 0.92) package in R (Version 4.3.0).

### Quantification of immunohistochemical stainings

Immunostained tissue sections were digitalized using a Hamamatsu NanoZoomer 2.0-HT C9600 whole slide scanner (Hamamatsu Photonics, Tokyo, Japan). Slide images were exported using NDP view v2.7.43 software. Digital image analysis was performed using ImageJ/Fiji software[102] after white balance correction in Adobe Photoshop 2022 (Adobe Inc., San Jose, USA). Tumor areas were labeled via manually drawn regions of interest (ROIs). Tissue areas not eligible for quantification (e.g., non-tumorous tissue, technical or digital artifacts) were excluded from the analysis. Total tumor tissue areas were measured in grayscale-converted images via consistent global thresholding (0, 241) and subsequent pixel quantification within the ROIs. DAB-positive pixels (i.e., brown immunostaining) were quantified on a three-tiered intensity scale after application of the color deconvolution plugin. In detail, pixels were successively quantified within three distinct thresholds [0, 134 (strong/3+); 135, 182 (medium/2+); and 183, 203 (weak/1+)]. Based on the conventional Histo-score, pixel quantities of strong, medium and weak intensity were multiplied by three, two and one, respectively, and then summed up. The hereby generated score is referred to as a digital Histoscore (DH-score).

### Reporting summary

Further information on research design is available in the Nature Portfolio Reporting Summary linked to this article.

### Data availability

Raw proteome data have been deposited under PXD039319 (TMT data), and PXD048767 (validation cohorts). Raw DNA Methylation and RNA Seq data can be accessed via GSE243796 containing subsets GSE222478 (450 K array DNA methylation data), GSE243768 (EPIC array DNA methylation data) and GSE243795 (RNA seq data). Raw metabolomics and amino acid data have been deposited to the EMBL-EBI MetaboLights database[82] with the identifier MTBLS9830 and MTBLS9836 respectively. Raw glycan data has been deposited at GlycoPOST[84] with the identifier GPST000414. Previously published data were included from EGAS00001001953[16], from GSE104728[17], GSE130051[78], GPL22286[5], MSV000082644 (MassIVE online repository) and PXD006607[16,17], PXD016832[29], or through the Clinical Proteomic Tumor Analysis Consortium Data Portal [https://cptac-data-portal.georgetown.edu/cptacPublic/] and the Proteomics Data Commons[15] [https://pdc.cancer.gov/pdc/]. Source data are provided in this paper.

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

## Acknowledgements

We thank Tasja Lempertz, Carolina Janko, Ulrike Rumpf, Karin Gehlken, Celina Soltwedel, Ann-Kathleen Leptien, Dr. Patrick Bluemke, and Paula Nissen for skillful and kind support. H.V. was funded by the Close the Gap program, University Medical Center Hamburg-Eppendorf, Hamburg, Germany. J.N. is funded by the Deutsche Forschungsgemeinschaft (DFG, Emmy Noether program), the Hamburger Krebsgesellschaft e.V. and the Erich und Gertrud Roggenbuck-foundation. We acknowledge financial support from the Open Access Publication Fund of UKE - Universitätsklinikum Hamburg-Eppendorf.

## Author contributions

S.G., H.V., A.G. and J.N. wrote and reviewed the manuscript. J.N. planned and designed the study. S.G. and H.V. conducted experiments. H.V., S.S., B.P., T.M., H.S., C.K., N.S., B.S. and Y.G. analyzed proteome and glycosylation data. A.G. and S.G. generated and analyzed biological and technical validation data. U.S. and S.G. analyzed methylation data. S.G., Y.S. and P.N. integrated proteome and methylome data. M.D. performed digitally supported quantification of IHC. S.P., S.R., M.My., MM. D., A.K., C.H., J.W., and F.L.-S. analyzed and interpreted histological, molecular and clinical data. A.G., M.Mo, M.K. and M.H. generated and analyzed the metabolomic and amino acid data. All authors reviewed the manuscript and approved its final version.

## Funding

## Competing interests

The authors declare no competing interests.

## Additional information

Shweta Godbole [1,2,20], Hannah Voß [3,20], Antonia Gocke [1,3], Simon Schlumbohm [4], Yannis Schumann [4], Bojia Peng [3], Martin Mynarek [5,6], Stefan Rutkowski [5], Matthias Dottermusch [1,2], Mario M. Dorostkar [7,8], Andrey Korshunov [9,10], Thomas Mair [3], Stefan M. Pfister [11,12,13], Marcel Kwiatkowski [14], Madlen Hotze [14], Philipp Neumann [4], Christian Hartmann [15], Joachim Weis [16], Friederike Liesche-Starnecker [17], Yudong Guan [3], Manuela Moritz [3], Bente Siebels [3], Nina Struve [6,18], Hartmut Schlüter [3], Ulrich Schüller [2,5,19], Christoph Krisp [3] & Julia E. Neumann [1,2] ✉

[1]Center for Molecular Neurobiology (ZMNH), University Medical Center Hamburg-Eppendorf, Hamburg, Germany. [2]Institute of Neuropathology, University Medical Center Hamburg-Eppendorf, Hamburg, Germany. [3]Section of Mass Spectrometry and Proteomics, University Medical Center Hamburg-Eppendorf, Hamburg, Germany. [4]Chair for High Performance Computing, Helmut Schmidt University, Hamburg, Germany. [5]Department of Pediatric Hematology and Oncology, University Medical Center Hamburg-Eppendorf, Hamburg, Germany. [6]Mildred Scheel Cancer Career Center HaTriCS4, University Medical Center Hamburg-Eppendorf, Hamburg, Germany. [7]Center for Neuropathology, Ludwig-Maximilians-University, Munich, Germany. [8]German Center for Neurodegenerative Diseases, Munich, Germany. [9]Department of Neuropathology, University Hospital Heidelberg, Heidelberg, Germany. [10]Clinical Cooperation Unit Neuropathology, German Cancer Consortium (DKTK), German Cancer Research Center (DKFZ), Heidelberg, Germany. [11]Hopp Children's Cancer Center at the NCT Heidelberg (KiTZ), Heidelberg, Germany. [12]Division of Pediatric Neurooncology, German Cancer Consortium (DKTK), German Cancer Research Center (DKFZ), Heidelberg, Germany. [13]Department of Pediatric Hematology and Oncology, Heidelberg University Hospital, Heidelberg, Germany. [14]Institute of Biochemistry, University of Innsbruck, Innsbruck, Austria. [15]Department of Neuropathology, Hannover Medical School (MHH), Hannover, Germany. [16]Institute of Neuropathology, RWTH Aachen University Hospital, Aachen, Germany. [17]Pathology, Medical Faculty, University of Augsburg, Augsburg, Germany. [18]Department of Radiotherapy & Radiation Oncology, University Medical Center Hamburg-Eppendorf, Hamburg, Germany. [19]Research Institute Children's Cancer Center Hamburg, Hamburg, Germany. [20]These authors contributed equally: Shweta Godbole, Hannah Voß. ✉e-mail: ju.neumann@uke.de

