## [Peer Review File · Nature Communications]

REVIEWER COMMENTS

Reviewer #1 (Remarks to the Author): Expert in MS-based proteomics, multi-omics integration, and bioinformatics

In Voß et al. manuscript, the authors integrated a harmonized proteome dataset of 167 MBs with DNA methylation and N-glycome data to study MB subtypes. The analysis revealed six proteome MB subtypes, including transcription/translation (pSHHT, pWNT and 55 pGroup3-Myc), and synapses/immunological processes (pSHHs, pGroup3 and 56 pGroup4). DNA-methylome profiling further revealed different conservation levels of proteome features across MB subtypes. The N-glycome was further used to confirm proteome subtypes. Overall, the manuscript is well-described, the methods address the questions, and the data support the conclusions. However, this reviewer has several concerns.

Major comments:

1. The manuscript would benefit from including more detailed sample information, including samples used in this study and those downloaded from public databases.
2. While the harmonizR was used for integrating multiple proteomic datasets, the manuscript lacks sufficient details about the quality control measures, especially with respect to control those co-variates, such as PMI, sex, age.
3. In Figure 1F, PCA plot was used to show 5 main molecular subgroups after harmonization. However, it would be better to label data sources instead of subgroups.
4. Although the authors described that a total of 16,279 proteins were quantified across all 167 samples, it is not clear how many proteins were identified and quantified in each dataset. How exactly the data were analyzed? What FDR was used for peptide identification?
5. The methods for analyzing TMT MS data are currently not provided, such as database search and protein filtering.
6. In Figure 2B, it is not clear what protein features were used? Are they a list of protein markers from each subtype? Additionally, a color scheme should be used for the heatmap.
7. In Glycosylation analysis, differentially expressed glycan analysis was used; however, no FDR was used for identification and quantification.
8. In the methylome analysis, differentially methylated regions (DMRs) instead of methylated sites would be better to define methylation events.

Minor comments:

1. In the abstract, “holdstrong” should be “hold strong”
2. “Global” should be “global”
3. “Taken together we” should be “Taken together, We”
4. In Figure 2A, two “HC Pearson” were used. One of them should be “HC Spearman”.
5. “In each batch, 1-2 internal reference samples were 749 included, composed of equal amounts of peptide material from all 70 samples and cell lines.” What are these cell lines?
6. In the method, the authors stated, “the samples were subjected to high pH liquid chromatography coupled mass spectrometry (LC-MS).” It is not clear how many fractionations were generated.

7. In line #253, "3,990 quantified proteins present in 30% of samples" should be "3,990 quantified proteins present in >=30% of samples"
8. In line #249, "using sparse variant partial least square discriminant analysis (sPLS-DA) was performed" should be no "variant"

Reviewer #2 (Remarks to the Author): Expert in medulloblastoma genomics and epigenomics

The manuscript by Voß et al. performed proteomics on 62 formalin-fixed-paraffin-embedded (FFPE) MB tumor samples and pooled the data with previously published proteomic data from 115 fresh-frozen (FF) MBs for analysis. The authors claim that, as with FF tissues, the FFPE materials are highly suitable for molecular subtyping. Their bioinformatics analyses describe distinct proteomic patterns of the combined datasets and suggest targeted alterations in MB. However, there are concerns about the quality of their FFPE data for molecular subtyping and the lack of experimental validations for many of the bioinformatics analyses.

Major Concerns:

- 1, The authors' claim that FFPE tumor tissues are highly suitable for molecular subtyping and large-scale analysis is not convincing, which is an important basis for this study. The authors should confirm the molecular subtyping of their FFPE data before pooling it with published datasets for analysis. Otherwise, the claims from pooling "poor" quality FFPE data with previously published FF data could be misleading. They must conduct independent analysis of their 62 FFPE samples alone for proteomic subtyping and N-glycan patterns and compare them to published FF datasets. Previous studies using "smaller cohorts of FF tissue" as the author stated (PMID: 29880060; 35977718), were able to define distinct molecular clusters.
2. It is critical to determine whether the proteomic quality of FFPE matches that of FF tissues. Fig. 1A does not appear to provide a unique clustering of tumor subtypes based on their FFPE dataset. The authors should perform the subgroup analysis in Fig 1G using their own FFPE data rather than pooling it with published FF data.
3. For the markers of group 3 MB (Fig. 2F), the authors should also provide other markers, such as NRL, IMPG2, and CRX, as previously reported in proteomics studies (PMID: 36131014, 36131015). Similarly, for the group 4 MB markers, authors should show markers such as EOMES, UNCX, LMX1A, SOX4, and BARHL1 in the proteome data.
4. In Fig 2H,I, the authors claim to identify two molecular profiles: 1, pWNT, pG3myc, and pSHHt, and 2, including pG3, pG4, and pSHHs. It is confusing based on their network analysis, and they should provide specific markers and experimental support for this claim.
5. The authors state that "Only a fraction of features, discriminating mainly the WNT subtype, showed a correlation of proteome and DNA-methylation data" in line 254-255. It is not clear which subgroup is correlated with proteome and DNA-methylation data in Fig 3. Authors should show the correlation score.

6. The authors assigned SHH groups into two molecular groups: pSHHs and pSHHt. The two proteome subtypes in SHH MB in Fig 4 should be validated with specific markers. The authors should confirm whether these differences exist at the transcriptome level and which group has p53 mutations.

7. The authors state that "pWNT MBs showed the lowest abundance of TriC/CCt proteins, whereas pG3myc MBs displayed the highest amount," but there is no data to support this claim.

8. In Figure 6J, genes/proteins are not consistently represented in the proteome, methylome, and transcriptome, making the data difficult to interpret. Experimental data are needed to determine which omics (proteome, methylome, and transcriptome) best reflects the identity of the sample.

Reviewer #4 (Remarks to the Author): Expert in brain cancers, MS-based proteomics and glycomics

we read with interest the article titled "Multiomic profiling of medulloblastoma reveals subtype-1 specific targetable alterations at the proteome and N-glycan level" where the authors have performed a comprehensive analysis of phenotypically relevant MB subtypes using three layer-Omic analysis of Proteomics, DNA methylation and N-glycome analysis.

There are a couple of queries that need to be answered:

The article is written in a very complex way that renders understanding the main objective of the study or the outcomes extremely challenging to the readers. First, the abstract should define the main aim of this analysis and what are the main objective after performing all this multiomic analysis.

The discussion started by stating that the main achievement is that the work was able to successfully apply proteomics on FFPE samples followed by analyzing the molecular patterns observed across MB subtypes. I think, if the first aim is two achieve sound data with FFPE, then this should be targeting a methodology/data processing specialized journal.

The association between the different omics especially the N glycan output was not clear especially since the authors utilized the protein data from the glycoprotein data rather than the "glycan: profile for the association. This is Fine with me; however, such data can be obtained from the proteomics data itself.

The justification for using DNA methylation-Proteomics and Glycomics should be more clarified in a schematic to indicate how they relate to each other to enable the readers to grasp the main purpose of using these methodologies. since this information is absent, I assumed that the authors have performed these omics analyses due to expertise availability.

The work lacks depth in the discussion and it stops at the conclusion that the team was able to perform differential phenotypically MB subtypes; this should be concluded to highlight its importance clinically and on the translational levels. For example, the output data showing that the Six proteome MB subtypes emerged could be assigned to two main molecular programs: transcription/translation and synapses/immunological processes is not sufficient if not linked to the implications of such findings!

The work lacks validation of the findings on the three levels performed: Methylation, Proteomics, and Glycomics which need to be performed.

Point-by-point response to the reviewers' comments

All answers are marked in **blue**, the changes within the manuscript are marked in **red**.

The newly generated data was deposited under accession numbers PRD044652 (for technical and validation proteome data, Username: reviewer_pxd044652@ebi.ac.uk, Password: 10UcezrA) and GEO accession numbers GSE243768 and GSE243795 (for methylome and transcriptome data, superseries GSE243796, reviewer token evupsiuaxrmbtgx).

REVIEWER COMMENTS

Reviewer #1 (Remarks to the Author): Expert in MS-based proteomics, multi-omics integration, and bioinformatics

In Voß et al. manuscript, the authors integrated a harmonized proteome dataset of 167 MBs with DNA methylation and N-glycome data to study MB subtypes. The analysis revealed six proteome MB subtypes, including transcription/translation (pSHHt, pWNT and 55 pGroup3-Myc), and synapses/immunological processes (pSHHs, pGroup3 and 56 pGroup4). DNA-methylome profiling further revealed different conservation levels of proteome features across MB subtypes. The N-glycome was further used to confirm proteome subtypes. Overall, the manuscript is well-described, the methods address the questions, and the data support the conclusions. However, this reviewer has several concerns.

Major comments:

1. *The manuscript would benefit from including more detailed sample information, including samples used in this study and those downloaded from public databases*

Answer:

In Supplementary Table 1c we have further refined the information on all samples included, including the one's from public databases. We also added the information which samples were used for what kind of analyses (e.g., DNA methylation, RNA-seq, Proteomics, Metabolomics) which respective identifiers to facilitate reuse of the data for future studies. Information on the newly added biological validation dataset can be found in Supplementary Table 11.

2. *While the harmonizR was used for integrating multiple proteomic datasets, the manuscript lacks sufficient details about the quality control measures, especially with respect to control those co-variates, such as PMI, sex, age*

Answer:

We agree with the reviewer that controlling co-variates is challenging when combining different datasets. The influence/ or presence of co-variates may even not be known which can be seen

as a limitation of dataset integration in general. In order to analyze the distribution of the above-mentioned co-variables in the respective datasets we further performed a chi-squared test to exclude significantly different or imbalanced distributions of sex, survival, main molecular subgroup or age across the integrated datasets (see table below). We were unsure about the mentioned co-variate “PMI” as we interpreted the abbreviation as “post mortem interval”, which is not applicable in our setting. We therefore used the co-variate “survival” instead. As no significant difference in the distribution of co-variables was seen among datasets (cf. Table 1), the harmonizR approach is applicable (which the advantage of data analyses without error-prone imputation).

Table 1 Statistical analyses (chi-squared test), testing for imbalanced distributions of sex, survival, age and main molecular subgroup among the integrated datasets. No significant differences could be observed, which proves that the harmonizR algorithm is applicable in our scenario.

	Sex	Survival	Main molecular subgroup	Age
Test	Chi-square	Chi-square	Chi-square	Chi-square
Chi-square, df	6.788,6	6.981,6	14.13, 12	33.05,30
P value	0.34090	0.3226	0.2922	0.3205
P value summary	Ns	Ns	Ns	Ns

As we are aware of the challenge of co-variables in general, we are currently adapting the harmonizR approach to enable missing value tolerant data integration taking other co-variables such as imbalanced genotype distribution / molecular subgroup distribution / sex distribution / etc across different batches into account. The respective manuscript is still under review (Bioconductor package BERT submitted, issue #3070). We expect the algorithm to be available in the near future and provide preliminary results in Table 2 underneath. These findings demonstrate, that co-variables such as gender or age group ($x \leq 3$, $3 < x \leq 11$, $11 < x$) do not have impact on the quality of dataset integration in our case.

Table 2 Exemplary dataset integration of our main cohort using the novel BERT algorithm (conceptually very similar to harmonizR, but considers co-variables - currently under peer-review). The average silhouette score (ASW) after dataset integration shows no significant difference for both the dataset of origin and the proteomic subgroup when using co-variables. This indicates that the influence of covariate distribution on dataset integration with harmonizR is negligible.

Co-variate	ASW Dataset of Origin	ASW Proteomic Subgroup
None	-0,0486	0,0099
Sex	0,0473	0,0096
Age Group	-0,0457	0,0098

In summary, the statistical analyses above (Table 1 and 2) demonstrate that the co-variate distribution of the integrated datasets are suitable for the harmonizR algorithm without additional control of the mentioned covariates.

3. In Figure 1F, PCA plot was used to show 5 main molecular subgroups after harmonization. However, it would be better to label data sources instead of subgroups

Answer:

We thank the reviewer for this comment. We restructured Figure 1 in the revision process, showing in Figure 1 E - G a principal component analysis using 70% valid values on combined (1E) and then harmonized data annotated for data sources (1F) and molecular subgroups (1G), respectively. The Figure legend has been adapted accordingly.

4. Although the authors described that a total of 16,279 proteins were quantified across all 167 samples, it is no clear how many proteins were identified and quantified in each dataset. How exactly the data were analyzed? What FDR was used for peptide identification?

Answer:

We added the number of quantified proteins for each dataset in the Supplementary methods section: "Processing of Proteome raw data for the integrated cohort:" (n tabular format, see also below). The table below shows the number of identified proteins in each dataset. As identified proteins may only be meaningful when enough datapoints are present, we added the amount of proteins displaying 30%, 70% or 100% of valid values (vv) in a given dataset. Of note, the respective FF studies based their findings on 30-70 %vv^{1,2}. In our study NIPALs PCA was based on 70%vv and consensus clustering on 30%vv. The respective information is now indicated for the figures 1 and 2 in the assigned legends. For further details on data analysis, see also below (answer to point 5). A false discovery rate (FDR) value threshold <0.01, using a reverted decoy peptide databases approach, was set for peptide identification.

Study	identified	30%vv	70%vv	100%vv
Forget (PXD006607)	14885	3892	3892	2485
Archer (MSV000082644,)	13000	11638	8953	7563
Petralia (TMT)	8802	5224	5306	4291
FFPE (TMT)	2957	2379	1242	809
FFPE (DDA, technical validation)	4794	3414	2370	891
FFPE (Biological Validation)	5800	3982	2712	1417

5. The methods for analyzing TMT MS data are currently not provided, such as database search and protein filtering.

Answer:

The respective information can be now found in the Methods (and Supplementary Methods) Section "Processing of Proteome raw data for the integrated cohort". The obtained raw in-house TMT-based LC-MS data and publicly available data (Archer et al (2018)¹⁸, TMT 10-Plex; Petralia et al. (2021)¹⁷, TMT 11-Plex) were processed with the Andromeda algorithm, implemented in the MaxQuant software (Max Plank Institute for Biochemistry, Version

1.6.2.10) and searched against a reviewed human database (downloaded from Uniprot February 2019, 26,659 entries). The carboxymethylation of cysteine residues was set as a fixed modification. Methionine oxidation, N-terminal protein acetylation and the conversion of glutamine to pyroglutamate were set as variable modifications. Peptides with a minimum length of 6 amino acids and a maximum mass of 6,000 Da were considered. The mass tolerance was set to 10 ppm. The maximum number of allowed missed cleavages in tryptic digestion was 2. A false discovery rate (FDR) value threshold <0.01, using a reverted decoy peptide databases approach, was set for peptide identification. Quantification was performed, based on TMT reporter intensities at MS3 level for LC-MS3 in-house data and at MS2 level for LC-MS2 data, acquired by Archer et al.¹⁸ and Petralia et al.¹⁷. All studies were searched separately. Fractions for each TMT batch were searched jointly. We used 10 fractions for this study.

Newly added biological and technical validation cohorts were analyzed in data-dependent acquisition (DDA) mode. Raw data were searched with the Sequest algorithm integrated in the Proteome Discoverer software (v 2.4.1.15), Thermo Fisher Scientific) against a reviewed human database (downloaded from Uniprot in June 2021, Containing 20,683 entries). Respective information has been added to the Supplementary Material and Methods section "Measurement of biological and technical validation datasets"

6. In Figure 2B, it is not clear what protein features were used? Are they a list of protein markers from each subtype? Additionally, a color scheme should be used for the heatmap.

Answer:

Proteins present in $\geq 30\%$ of samples were used for consensus clustering, resulting in 3990 proteins for the joint dataset (Figure 2B). The information has been added to the figure legend). Of note no prior selection of proteins was performed to enable an unbiased approach when looking for clusters within the data. The list of proteins considered can be found in Supplementary Table 1b. A color scheme for the correlation matrix has been added.

7. In Glycosylation analysis, differentially expressed glycan analysis was used; however, no FDR was used for identification and quantification.

Answer:

Identification and quantification for N-glycans was done based on the method described by Guan et al.³. We mentioned this in the Material methods under "Processing of N-Glycan raw data:". As the process of glycan-identification was conducted manually in concordance to Guan et al.³ no FDR could be calculated. However a deviation threshold of 2.5 ppm on MS1 level was used for the identification process³.

8. In the methylome analysis, differentially methylated regions (DMRs) instead of methylated sites would be better to define methylation events.

Answer:

The brain tumor classifier (MolecularNeuropathology.org) that is implemented in the current WHO classification of brain tumors (Louis et al. 2021)⁴ was developed with differentially

methyated CpG sites (DMPs), hence we used DMPs for our analysis. To further address the reviewer's feedback we re-performed all the analysis using differentially methylated regions (DMRs) and added a Supplementary Figure 9 which showed consistency of results. We set a min of 10 CpGs per DMR (< 1000 nt from each other) to minimize gene overlap, which resulted in ~9000 DMRs with each DMR having 10-200 CpGs. We then again integrated methylation data with the proteome data using DIABLO (mixOmics R Package) and still observed a high correlation for "WNT-specific" features, as can be seen in the circus plot in Supplementary Figure 9A. We further correlated all CpGs in these DMRs with all proteins using Pearson correlation and cut-off ≥ 0.7 and again observed a high correlation for the pWNT group compared to other proteome subtypes (Supplementary Figure 9B). We added the DMR analyses in the methods part, under "Processing of DNA Methylation Array Data:" and "Global correlation of Proteome and DNA Methylation data" respectively.

Minor comments:

1. *In the abstract, "holdstrong" should be "hold strong"*

Answer:

We have corrected this accordingly.

2. *"Global" should be "global"*

Answer:

We have corrected this accordingly.

3. *"Taken together we" should be "Taken together, we"*

Answer:

We have corrected this accordingly.

4. *In Figure 2A, two "HC Pearson" were used. One of them should be "HC Spearman"*.

Answer:

Thank you for pointing it out, Figure 2A has been corrected.

5. *"In each batch, 1-2 internal reference samples were 749 included, composed of equal amounts of peptide material from all 70 samples and cell lines." What are these cell lines?*

Three different medulloblastoma cell-lines were used: DAOY (Ca#HTB-186) and D283med (Ca#HTB-185) obtained from ATCC, Manassas, VA, USA and UW473 kindly provided by Michael Bobola. The respective information can be found in the methods part ". Main heading: Subject Details, subheading: "Medulloblastoma cell-lines".

6. *In the method, the authors stated, "the samples were subjected to high pH liquid chromatography coupled mass spectrometry (LC-MS)." It is not clear how many fractionations were generated.*

Answer:

In total 32 fractions were collected and pooled according to schematic below:

New fraction	Fractions for sample pooling
-3	1, 2, 3
-2	4, 5, 6
-1	7, 8, 9
1	10, 20, 30
2	11, 21, 31
3	12, 22, 32
4	13, 23
5	14, 24
6	15, 25
7	16, 26
8	17, 27
9	18, 28
10	19, 29

This resulted in 13 fractions. We have clarified this by adding that we generated 13 fractions. The respective information has been added to the Material and Methods section.

7. In line #253, “3,990 quantified proteins present in 30% of samples” should be “3,990 quantified proteins present in $\geq 30\%$ of samples”

Answer:

We have corrected this accordingly.

8. In line #249, “using sparse variant partial least square discriminant analysis (sPLS-DA) was performed” should be no “variant”

Answer:

We have corrected this accordingly.

Reviewer #2 (Remarks to the Author): Expert in medulloblastoma genomics and epigenomics

The manuscript by Voß et al. performed proteomics on 62 formalin-fixed-paraffin-embedded (FFPE) MB tumor samples and pooled the data with previously published proteomic data from 115 fresh-frozen (FF) MBs for analysis. The authors claim that, as with FF tissues, the FFPE materials are highly suitable for molecular subtyping. Their bioinformatics analyses describe distinct proteomic patterns of the combined datasets and suggest targeted alterations in MB. However, there are concerns about the quality of their FFPE data for molecular subtyping and the lack of experimental validations for many of the bioinformatics analyses.

Major Concerns:

1. The authors' claim that FFPE tumor tissues are highly suitable for molecular subtyping and large-scale analysis is not convincing, which is an important basis for this study. The authors should confirm the molecular subtyping of their FFPE data before pooling it with published⁶ datasets for analysis. Otherwise, the claims from pooling “poor” quality FFPE data with previously published FF data could be misleading. They must conduct independent analysis

of their 62 FFPE samples alone for proteomic subtyping and N-glycan patterns and compare them to published FF datasets. Previous studies using "smaller cohorts of FF tissue" as the author stated (PMID: 29880060; 35977718), were able to define distinct molecular clusters.

Answer:

There have been several studies that compared the quality of proteomic data obtained from FFPE and FF tissue samples. For example, a study by Wiśniewski et al. (2013)⁶ found, that the two sample types had a high degree of correlation in terms of protein identification and quantification, with a Pearson correlation coefficient of 0.92, based on liver tissue⁷. Similar results were obtained by Mantsiou et al. (2020)⁸ when comparing proteome data from FFPE and FF brain tissues and in a study on prostate cancer tissues, respectively^{5,9}. Besides the studies mentioned above, we additionally reanalyzed matched FF and FFPE data of mouse medulloblastoma and cerebellar tissue showing a similarly high correlation of matched data ($r=0.80$, $n_{\text{matchedCases}}=8$ (data reanalyzed from Voss et al. 2022)¹⁰. Of note, the usage of proteome patterns from FFPE tissues has been also recently shown in the brain tumor context¹¹. Here, new biomarkers for IDH-mutant gliomas were discovered. We agree with the reviewer that proteome data of FFPE tissue contains less features as FF data, this, however, does not reflect "poorer quality" per se. FFPE tissue samples can reveal similarly distinct clusters (see also below) and similarly significant proteins (see also Supplementary Figure 1, and newly added Figure 10, Supplementary Figure 2 and 16). FFPE tissues are also more stable over the long term, as they can be stored at room temperature for years without significant degradation. We also showed that proteome quality and number of quantified proteins did not depend on sample age (Supplementary Figure 3B), which we could again verify in the technical validation cohort using a different quantification strategy (DDA instead of TMT, Supplementary Figure 16C-D). Therefore, FFPE samples are accessible in a larger quantity and have the potential to overcome cohort size limitations, that often lead to the loss of the validity and reliability of scientific studies⁵. Further, the successful bioinformatic integration of diverse proteomic datasets (including FF and FFPE) has been shown in detail by our group before¹⁰. Overall, the advantages of a larger cohort size clearly outweigh the disadvantages of a lower amount of quantified proteins.

Following the reviewer's suggestion concerning the datasets at hand, we analyzed each dataset independently (our FFPE and the 3 FF datasets, and additionally a technical and biological FFPE validation dataset, Figure 10 and Supplementary Figure 1, 2 and 16, Supplementary Table 1c, Supplementary Table 11). Here, we show that for all studies considered in our manuscript, similar patterns can be observed in PCA, enabling the clear distinction of the four main molecular subtypes of Medulloblastoma, based on the two main principal components (Supplementary Figure 1A) for FF and FFPE-derived proteome data respectively. Differentially proteomics separating samples according to the first main components also revealed a significant overlap of commonly enriched gene sets, which of note reflected either terms like "Neuronal System" or "Opioid Signalling" "or "Nucleotide biosynthesis" or "metabolism of RNA" confirming the main superordinate clusters "synaptic" versus "translational/transcriptional" that we detected in the combined cohort (Supplementary Figure 2C, Supplementary Table 3a-c). Furthermore, based on present proteins in each dataset individually, a clustering relating to main MB subgroups (and proteome subtypes) could be detected (Supplementary Figure 2B). In Figure 1 we additionally now show established IHC markers in diagnostic for WNT and SHH MB (CTNNB1, GAB1 and FLNA) in each dataset independently before integration of the data). These markers were confirmed in the technical

and biological dataset, respectively (Supplementary Figure 16, Figure 10). We included the novel analyses and data in the results and methods part of the manuscript and uploaded the data to publicly available databases (GEO and PRIDE, respectively, see also section “data availability” in the manuscript).

In summary, the integration of proteome data of FFPE samples and to FF samples is feasible, (as has been also shown for other datasets¹⁰) enabling large scale data integration in future studies and revealing a great potential for future clinical applications.

2. It is critical to determine whether the proteomic quality of FFPE matches that of FF tissues. Fig. 1A does not appear to provide a unique clustering of tumor subtypes based on their FFPE dataset. The authors should perform the subgroup analysis in Fig 1G using their own FFPE data rather than pooling it with published FF data.

Answer:

Concerning the general comparability of FFPE and FF tissue, see also our in-depth answer to point 1 above. As stated before, we have now analyzed each dataset individually, showing a similar separation of all samples into MB subtypes (PCA, HCL), a similar protein abundance of the established diagnostic markers CTNNB1, GAB1 and FLNA and did also observe a main separation into two profiles associated with synaptic and transcriptional terms in all individual studies (Figures 1, Figure 10, Supplementary Figures 1, 2,16). We included the novel analyses and data in the results and methods part of the manuscript, respectively (see also answer above).

3. For the markers of group 3 MB (Fig. 2F), the authors should also provide other markers, such as NRL, IMPG2, and CRX, as previously reported in proteomics studies (PMID: 36131014, 36131015). Similarly, for the group 4 MB markers, authors should show markers such as EOMES, UNCX, LMX1A, SOX4, and BARHL1 in the proteome data.

Answer:

We thank the reviewer for suggesting other markers previously found based on immunohistochemistry or RNA based studies. NRL, UNCX, LMX1A, CRX, EOMES and SOX4 are all transcription factors. In general, transcription factors show rather low protein abundances in comparison to other cellular proteins and are hence scarcely detected in proteome data due to the general technical limitations of mass spectrometry¹²

We searched for all the mentioned biomarkers in all individual studies (FFPE, Forget, Archer, Petralia, and the newly generated FFPE biological and technical validation data). We could identify IMPG2, and CRX, as well as EOMES, SOX4, and BARHL1 (of note presence or absence of markers was not specific to FFPE or FF datasets, but rather reflected the general technical variance in proteome datasets (independent of sample preservation)). See below a dotplot displaying every sample in which the proteins were quantified in the individual studies.

Of these, BARHL1 and EOMES showed a higher abundance in G4 MB in the FFPE datasets found in, as expected. The other markers did not show significant differences. A potential reason for that could be a difference between mRNA level (where most markers were described in) and protein level (as has been shown for several other markers in our manuscript). Another possibility - besides others - could be that intracellular localization of the factors may play a more important role (e.g. the transcription factors have to be abundant in the nucleus to execute their function).

4. In Fig 2H,I, the authors claim to identify two molecular profiles: 1, pWNT, pG3myc, and pSHHt, and 2, including pG3, pG4, and pSHHs. It is confusing based on their network analysis, and they should provide specific markers and experimental support for this claim.

Answer:

In addition to the already displayed network analyses, we added the list of the proteins which make up these two profiles in Supplementary Table 3a,b. Complementarily performed gene set enrichment analyses were performed using Ingenuity pathway analysis (IPA) and the proteins belonging to the two most enriched pathways for each pattern, namely "Opioid Signaling" and "SNARE complex" (synaptic pattern), and EIF2 Signaling and Cell Cycle control of Chromosomal Replication (transcriptional/ translational pattern) were additionally displayed in relation to protein subtypes in a newly structured Figure 3 B,C (for details on IPA analyses see also Supplementary Table 3 c-g). The data was additionally used to identify druggable targets based on therapeutics already applied in a clinical setting/clinical trials (Figure 3 D;E, Supplementary Table 3 c-g see also answer 4 to reviewer 4).

In addition, we aimed to validate our findings. As stated above we analysed each dataset independently verifying the two main profiles (see answers 1 and 2 above and Supplementary Figure 2). Moreover, we used a new technical validation dataset comprised of 57 medulloblastoma samples to reproduce results and added a new biological validation dataset comprised of 30 medulloblastoma samples to again verify the results of the main cohort. We confirmed the high abundance of proteins belonging to the above-mentioned gene sets in MB subtypes accordingly (Supplementary Figure 2BC, 16F and Figure 10G). In order to verify data using a distinct data modality, we generated new RNA-seq data of cases with known proteome subtypes and integrated these data with previous RNA and Protein data (n=53 with matched data of all three omic types: proteome-transcriptome and DNA methylome, $n_{pWNT}=4$; $n_{pSHHt}=13$, $n_{pSHHs}=4$, $n_{pG3}=6$, $n_{pG3myc}=10$, $n_{pG4}=16$). Based on overlapping mRNAs and proteins making up the two defined profiles, we could recapitulate the MB subtypes, biomarkers and two main clusters (synaptic and transcriptional/translational) – to a large extent (Supplementary Figure 2E,F). Respective analyses and results have been included in the “results” part of the manuscript.

5. The authors state that "Only a fraction of features, discriminating mainly the WNT subtype, showed a correlation of proteome and DNA-methylation data" in line 254-255. It is not clear which subgroup is correlated with proteome and DNA-methylation data in Fig 3. Authors should show the correlation score.

Answer:

Thank you for the feedback, we would like to clarify this statement. Only a fraction of features out of the 381717 probes and 3990 proteins showed correlation upon data integration using DIABLO from mixOmics. Out of these features, the features that show correlation have pWNT-specific expression as can be seen by the lines around the circus plot in Figure 4A. These lines show in which group the selected feature has high or low methylation/abundance. All correlation scores are additionally shown in Supplementary Table 4h. All correlation values ($r>0.7$) that are displayed in the circus plot of Figure 4A are shown in red colour within the table. We have adapted this in the manuscript by adding arrows and furthermore inserted clarifications on correlation in the results section.

6. The authors assigned SHH groups into two molecular groups: pSHHs and pSHHt. The two proteome subtypes in SHH MB in Fig 4 should be validated with specific markers. The authors should confirm whether these differences exist at the transcriptome level and which group has p53 mutations.

Answer:

Thanks for the valuable feedback. We generated transcriptional data for our samples and integrated it with the transcriptional data from Archer et al which resulted in matched transcriptome data for n=60 samples. After data integration (based on overlapping mRNAs and proteins), we could recreate the results from the proteome level at the transcriptome level - including representation of the six-proteome subtypes, observation of the synaptic and transcriptional profiles (to a certain extent) and confirmation of higher gene mRNA expression of most biomarkers identified at the proteome level. These results are now attached in new Supplementary Figure 2D-G. Solely, focusing on pSHHt and pSHHs at the transcriptome level, we saw a trend but no such clear separation into the two profiles see also new Supplementary

Figure 10A-B). These results are well in line with the results of Archer et al ¹ that already reported a low correlation of mRNAs and proteins in SHH MB.

Out of 10 TP53 mutated SHH cases, 9 fell into the pSHHt cluster (Figure 5A) and we detected significant abundant proteins comparing pSHHt TP53 mutant cases with pSHHt TP53 wildtype cases indicating differences on proteome level (Figure 5J). In the matched RNA/proteome data, we only had 2 confirmed TP53 mutated cases and both cases grouped to the pSHHt group (Supplementary Figure 10B).

These findings underscore, that the combination of diverse omics types is crucial for in depth characterization of MBs and for the understanding of their pathogenesis.

7. The authors state that "pWNT MBs showed the lowest abundance of TriC/Cct proteins, whereas pG3myc MBs displayed the highest amount," but there is no data to support this claim. –

Answer:

Initially, we showed a heatmap plotting protein abundances values across subtypes in the main cohort. We have now shifted this figure panel, together with the RNA data of the Cavalli cohort to Supplementary Figure 14. Instead, we now created a new main Figure 8 focusing on CCT in more detail. We now included only matched cases of DNA methylome/transcriptome and proteome data (see also answer to comment number 8) and performed quantitative analysis between subgroups to support this claim (Figure 8A). We added an additional plot with pvalues showing that all the components are significantly downregulated in the pWNT group (Figure 8A) and significantly upregulated in the pG3myc group). We further validated the findings in a biological validation cohort (n=30), where we confirmed that Tric/CCT complex proteins had low abundance in the pWNT group and a high abundance in the pG3myc group (Figure 10, Supplementary Figure 11). The manuscript has been adapted accordingly in the results part.

8. In Figure 6J, genes/proteins are not consistently represented in the proteome, methylome, and transcriptome, making the data difficult to interpret. Experimental data are needed to determine which omics (proteome, methylome, and transcriptome) best reflect the identity of the sample.

To reflect the identity of the sample and simplify interpretation we generated (based on sample availability) additional RNA transcriptome data of our characterized samples and integrated it with the matched RNA-seq data of the Archer cohort, which resulted in matched transcriptome proteome and DNA methylation data for n=60 samples. In the newly created Figure 8, we have analyzed the TRiC/CCT- complex proteins in matched samples across all different omic types - DNA methylome, transcriptome and proteome. This showed that the TRiC/CCT- complex components are all consistently downregulated in the pWNT subtype based on protein abundances. Bases on mRNA or DNA methylome level no such clear association was seen (Figure 8C). In a case matched correlation, we in general detected a mostly positive correlation for RNA and protein data, and a negative correlation for DNA methylation and RNA expression (as expected, Figure 8D). However, results were not as clear for DNA methylation and protein abundances and only CCT2 showed a significant correlation across all Omics types ($r \geq 0.7$, Figure 8E). Further, we also looked for mRNA expression of the CCT complex components in a large publicly available dataset from Cavalli et al, 2017 ¹³, which supported the fact that

differences in CCT components are not reflected at the transcriptome level (Supplementary Figure 14). In conclusion, we hypothesize that the phenotype of CCT components is best reflected at the proteome level and that this complex might represent a promising therapy target. We are further following up on this topic and aim to investigate CCT components in medulloblastoma on a functional level in the future. We have adapted the results part of the manuscript including the new figures.

Reviewer #4 (Remarks to the Author): Expert in brain cancers, MS-based proteomics and glycomics

we read with interest the article titled "Multiomic profiling of medulloblastoma reveals subtype-1 specific targetable alterations at the proteome and N-glycan level" where the authors have performed a comprehensive analysis of phenotypically relevant MB subtypes using three layer-Omic analysis of Proteomics, DNA methylation and N-glycome analysis. There are a couple of queries that need to be answered:

The article is written in a very complex way that renders understanding the main objective of the study or the outcomes extremely challenging to the readers. First, the abstract should define the main aim of this analysis and what are the main objective after performing all this multiomic analysis.

1. The discussion started by stating that the main achievement is that the work was able to successfully apply proteomics on FFPE samples followed by analyzing the molecular patterns observed across MB subtypes. I think, if the first aim is two achieve sound data with FFPE, then this should be targeting a methodology/data processing specialized journal

Answer:

Thank you for this comment. Based on the reviewer's feedback we rephrased the abstract and parts of the manuscript and the abstract to highlight the main objectives in a clearer way. E.g. in the abstract, we included the phrase "The application of omics technologies – mainly studying nucleic acids – has significantly improved MB classification and stratification, but treatment options are still unsatisfactory. The proteome and their N-glycans reflect the phenotype of a tumor in a more direct way and thus hold the potential to discover clinically relevant phenotypes and targetable pathways."

Indeed, the major aim of this study was to detect phenotypically relevant subtypes of medulloblastoma in order to detect targetable alterations in this malignant tumor. The major points are that in medulloblastoma proteome subtypes (and N-glycan patterns) associate with DNA-methylation subgroups. The latter are used for classification of brain tumors in the clinic (WHO classification of brain tumors)⁴. This underlines that the proteome harbors a great potential for identifying subtype specific therapy targets (whereas the DNA methylome is useful for diagnosis and classification). While we see a clear association of subtypes in general, gene specific correlation of both data modalities is rather low (and depends on the subtype!). This is to be expected as posttranscriptional and posttranslational mechanisms additionally impact on protein abundance, highlighting that DNA methylation data is "too far" from the actual

phenotype to reveal specific targets. As an example, we show that the CCT complex is significantly altered on the protein level – this complex was shown to be associated with chemotherapy response and might be a potential therapy target especially for high risk G3myc MB in the future. The usage of FFPE tissue, and the harmonizR algorithm for combination of proteomic datasets were a basis for our analyses but were not the main aspect of this manuscript. Our Figure 1 and Supplementary Figure 2 show that our analyses based on FFPE and data integration of multiple datasets are technically sound in the context of medulloblastoma. The usage of FFPE material for molecular subtyping of tumors per se has also recently been shown by other groups underlining the potential of the method (please refer also to point 1 of reviewer 2).

2. The association between the different omics especially the N glycan output was not clear especially since the authors utilized the protein data from the glycoprotein data rather than the “glycan: profile for the association. This is Fine with me; however, such data can be obtained from the proteomics data itself.

Answer:

In this study, we aimed to analyze the structure of N-Glycans (“glycomics”) in order detect potential targets for Immunotherapy. We agree with the reviewer, that “glycoproteomic” data, which usually refer to the site of glycosylation on glycoproteins, may be generally obtained from the same spectra acquired for proteomic analysis. However, the complexity of the glycan structures results in highly complex and ambiguous spectra, which are very hard to analyze even for state-of-the-art tools. Thus, when acquiring glycoprotein data, a higher degree of separation is required. Either specific proteins are isolated beforehand, so shorter gradients of 60 minutes can be applied, or - when trying to contain the complexity of the sample - longer gradients of at least 120 minutes may be used. According to the reviewer’s comment, we tried to analyze glycoprotein data from our spectra, but as our acquisition method for the proteomic data only used a gradient of 80 minutes for complex samples, the spectra couldn’t be used for glycoproteomic analysis. Additionally, while analysis of proteoglycans from the spectra obtained for proteomic analysis may be possible in general, there is still a loss of information, if the same chemical digest is used. Therefore, researchers investigating proteoglycans specifically tend to use different, softer methods of the enzymatic digest and measurement^{14–18}.

3. The justification for using DNA methylation-Proteomics and Glycomics should be more clarified in a schematic to indicate how they relate to each other to enable the readers to grasp the main purpose of using these methodologies. since this information is absent, I assumed that the authors have performed these omics analyses due to expertise availability.

Answer:

DNA methylation is widely used for accurate brain tumor classification and even mandatory for certain brain tumor entities^{4,19}. Therefore, we considered the data crucial for medulloblastoma subtyping. DNA methylation also impacts on gene expression. Generally, DNA methylation inhibits gene transcription, and loss of methylation is considered to be associated with gene activation^{20–22}. In this revision, we additionally added mRNA Seq data to further explore this relationship in the MB context (see for instance answer 4 to reviewer 2). The focus on proteome and N-glycan patterns was pursued because proteins reflect a tumor’s phenotype more closely than nucleic acids and might therefore represent direct therapy targets. Post-translational

modifications (PTMs) make the proteome more dynamic²³ and changes in glycan patterns play a role in development, infection and cancer²⁴. These patterns may also be controlled by epigenetic modifications²⁵. To our knowledge, Glycans have not been studied in MB before. However, Glycans are of specific interest, as they are located on the cell surface and play a major role in immune response, therefore opening the field for potential immune therapies in the future^{26,27}

Following the reviewer's suggestion, we added a schematic to explain how the different data modalities relate to each other which is now mentioned in the discussion section (the scheme has been added as Supplementary Figure 17):

4. The work lacks depth in the discussion and it stops at the conclusion that the team was able to perform differential phenotypically MB subtypes; this should be concluded to highlight its importance clinically and on the translational levels. For example, the output data showing that the Six proteome MB subtypes emerged could be assigned to two main molecular programs: transcription/translation and synapses/immunological processes is not sufficient if not linked to the implications of such findings!

Answer:

We thank the reviewer for pointing this out. The clinical significance of this work is important and in the process of rephrasing the manuscript we aimed to sharpen statements accordingly. 1) We show that DNA-methylation subgroups (used for classification) are associated with medulloblastoma proteome subtypes (and N-glycan patterns) indicating that the subtype specific proteome profile is suitable to identify therapy targets (e.g., the CCT complex or Glycan structures which we will further pursue in future studies). We have highlighted this again in the discussion section.

2) Besides subtype specific profiles, we detected superordinate profiles (synaptic versus transcriptomic) that reveal common targets (for further details see below)

3) Proteome signatures may be additionally important for stratification. We detected a poor surviving G3myc MB subtype that contained MYC amplified as well as non-MYC amplified tumors. This shows that the current stratification scheme for high-risk MB (based solely on (genetic) MYC amplification) may miss these high-risk patients. Prospective studies should evaluate the usage for MYC stratification based on proteome profiles in the future

4) Proteome profiles are also helpful for diagnostic purposes, e.g., we identified PALMD as TNC as useful immunohistochemical markers for MB subtyping in the routine histology. This is especially important for diagnostic units in the world that do not have upfront access to DNA methylation analyses

5) Data integration further helps to understand the mechanisms of tumorigenesis with the median-term aim to find additional therapy targets. We detected a significantly differential correlation between data modalities in MB subtypes highlighting a diverse feature conservation and that posttranscriptional and translational processes might be of specific importance for some subtypes (e.g. group 4 MB). Further functional studies are needed to show the potential for therapy in the future.

In order to specifically highlight therapeutic implications of superordinate synaptic and transcriptional MB profiles, we performed Ingenuity pathway analysis (IPA) and searched for

druggable targets based on proteome profiles. We focused on drugs that are in use in clinical trials (phase 1-3) and therefore reflect potential of clinical applicability. To evaluate a therapeutic potential of these patterns, we used IPA²⁸ and identified, besides others, CDK4 inhibitors as potential drugs for targeting the groups belonging to the transcriptional profile. Various CDK inhibitors are already FDA-approved for treatment of different types of metastatic cancers and CDK4/6 inhibition has been shown to inhibit tumor growth of medulloblastoma cells in vivo²⁹. In contrast, proteome subtypes belonging to the synaptic profile may be – besides others - targeted with the NMDA receptor antagonist memantine. Of note, memantine has neuroprotective properties and was shown to decrease cognitive dysfunction in patients receiving radiotherapy³⁰. As radiotherapy is also applied to MB patients the drug may be of specific interest, however, further studies are needed to investigate the clinical potential of the mentioned drugs for MB patients. Respective results are now shown in Figure 3, Supplementary Figure 6 and Supplementary Tables 3c-3g) and the results part was adapted accordingly. Furthermore, the here mentioned section was added to the discussion part of the manuscript.

5. The work lacks validation of the findings on the three levels performed: Methylation, Proteomics, and Glycomics which need to be performed.

Answer:

We generated technical and biological validation datasets, re-performed the analyses and could confirm the six-proteome subtypes, the two main profiles – synaptic and transcriptional, as well as the biomarkers namely TNC for pWNT group and PALMD for pG3myc. We could also confirm that the CCT complex has lowest abundance in the pWNT group and highest in pG3myc for all the validation datasets. Finally, for the biological validation data we have also added an additional Figure 10 and Supplementary Table 11..

In order to validate of our findings in the joint proteome cohort and confirm the stability of results:

We (1) analyzed each dataset of the joint cohort independently and showed that FFPE and FF tissue-based datasets give similar results concerning sample clustering and altered pathways. Respective results are now shown in Supplementary Figures 1 and 2 (new).

We (2) used a new technical validation dataset comprised of 57 medulloblastoma samples to reproduce results and (3) added a new biological validation dataset comprised of 30 medulloblastoma samples to again verify the results of the main cohort. Respective results are now shown in the newly added Supplementary Figure 16 and Figure 10, Supplementary Table 11.

We (4) further tested and verified two candidate biomarkers of proteome data, using immunohistochemistry (MYC and ALDH1A3). Both pSHHt and pSHHs showed a high abundance of ALDH1A3, that we could further confirm (see newly added Supplementary Figure 10B, $n_{pSHHs}=5$, $n_{pSHHt}=11$, $n_{non-SHH}=16$). Additionally, we identified a significantly higher fraction of CMYC positive tumor cell nuclei in G3myc MBs compared to all other MB subtypes, confirming the MYC profile shown in Figure 6K (new Supplementary Figure 12). Of note, a high amount of CMYC positive tumor cell nuclei was also detected in the pWNT group – as expected – as CMYC is a known WNT target and these tumors did also show a MYC profile to some extent (Figure 6K). Respective results are now shown in Supplementary Figure 12.

For a sample matched analyses of DNA Methylation data, RNA sequencing data and Proteome data, we (5) performed RNA-seq analyses of additional MB cases (n=21), which finally resulted – together with the data of Archer et al. - in n=60 matched cases, representing all proteome

MB subtypes. Integrating these data, we verified the low correlation of data modalities concerning the TriC/CCT complex components. Respective results are now shown in the newly added Figure 8.

Additionally, we re-analysed RNA-seq data concerning the found patterns of proteome data (Synaptic/transcriptional profiles and subtypes). Respective results are now shown in Supplementary Figure 2D-G and Supplementary Figure 10A,B.

Finally, we (6) additionally measured metabolites and amino acids of SHH MB samples to verify detected proteome patterns with another independent data modality. Respective results are now shown in Supplementary Figure 11.

The newly generated data was deposited under accession numbers PRD044652 (for technical and validation proteome data) and GEO accession numbers GSE243768 and GSE243795 (for methylome and transcriptome data, superseries GSE243796):

Reviewer account details:

1. PRD044652
Username: reviewer_pxd044652@ebi.ac.uk
Password: 10UcezrA
2. GSE243768 and GSE243795 (superseries GEO accession GSE243796):
Go to <https://www.ncbi.nlm.nih.gov/geo/query/acc.cgi?acc=GSE243796>
Enter token evupsiuaxrmbtgx into the box

Bibliography

1. Archer, T. C. *et al.* Proteomics, Post-translational Modifications, and Integrative Analyses Reveal Molecular Heterogeneity within Medulloblastoma Subgroups. *Cancer Cell* **34**, 396--410.e8 (2018).
2. Forget, A. *et al.* Aberrant ERBB4-SRC Signaling as a Hallmark of Group 4 Medulloblastoma Revealed by Integrative Phosphoproteomic Profiling. *Cancer Cell* **34**, 379--395.e7 (2018).
3. Guan, Y., Zhang, M., Wang, J. & Schlüter, H. Comparative Analysis of Different N-glycan Preparation Approaches and Development of Optimized Solid-Phase Permethylation Using Mass Spectrometry. *J Proteome Res* **20**, 2914--2922 (2021).
4. Louis, D. N. *et al.* The 2021 WHO Classification of Tumors of the Central Nervous System : a summary. *Neuro Oncol* **23**, 1231--1251 (2021).
5. Rivero-Hinojosa, S. *et al.* Proteomic analysis of Medulloblastoma reveals functional biology with translational potential. *Acta Neuropathol Commun* **6**, 48 (2018).
6. Wiśniewski Jacek R. Proteomic Sample Preparation from Formalin Fixed and Paraffin Embedded Tissue.
7. Thongboonkerd Visith. *Proteomics of Human Body Fluids Principles, Methods, and Applications*. (2007).
8. Mantsiou, A. *et al.* Proteomics Analysis of Formalin Fixed Paraffin Embedded Tissues in the Investigation of Prostate Cancer. *J Proteome Res* **19**, 2631--2642 (2020).

9. Leskoske, K. L. *et al.* Subgroup-Enriched Pathways and Kinase Signatures in Medulloblastoma Patient-Derived Xenografts. *J Proteome Res* **21**, 2124–2136 (2022).
10. Voß, H. *et al.* HarmonizR enables data harmonization across independent proteomic datasets with appropriate handling of missing values. *Nat Commun* **13**, 3523 (2022).
11. Felix, M. *et al.* HIP1R and vimentin immunohistochemistry predict 1p/19q status in IDH-mutant glioma. *Neuro Oncol* **24**, 2121–2132 (2022).
12. Simicevic Jovan & Deplancke Bart. Transcription factor proteomics-Tools, applications, and challenges .
13. Cavalli, F. M. G. *et al.* Intertumoral Heterogeneity within Medulloblastoma Subgroups. *Cancer Cell* **31**, 737-754.e6 (2017).
14. Sun, R., Kim, A. M. J. & Lim, S.-O. Glycosylation of Immune Receptors in Cancer. *Cells* **10**, (2021).
15. Bradberry, M. M., Peters-Clarke, T. M., Shishkova, E., Chapman, E. R. & Coon, J. J. N-glycoproteomics of brain synapses and synaptic vesicles. *Cell Rep* **42**, 112368 (2023).
16. Chang, D., Klein, J. A., Nalehua, M. R., Hackett, W. E. & Zaia, J. Data-independent acquisition mass spectrometry for site-specific glycoproteomics characterization of SARS-CoV-2 spike protein. *Anal Bioanal Chem* **413**, 7305–7318 (2021).
17. Klein, J. A., Meng, L. & Zaia, J. Deep Sequencing of Complex Proteoglycans: A Novel Strategy for High Coverage and Site-specific Identification of Glycosaminoglycan-linked Peptides. *Mol Cell Proteomics* **17**, 1578–1590 (2018).
18. Shajahan, A., Heiss, C., Ishihara, M. & Azadi, P. Glycomic and glycoproteomic analysis of glycoproteins—a tutorial. *Anal Bioanal Chem* **409**, 4483–4505 (2017).
19. Capper, D. *et al.* DNA methylation-based classification of central nervous system tumours. *Nature* **555**, 469–474 (2018).
20. Cedar, H. DNA methylation and gene activity. *Cell* **53**, 3–4 (1988).
21. Curradi, M., Izzo, A., Badaracco, G. & Landsberger, N. Molecular mechanisms of gene silencing mediated by DNA methylation. *Mol Cell Biol* **22**, 3157–3173 (2002).
22. Wang, Z. *et al.* Complex impact of DNA methylation on transcriptional dysregulation across 22 human cancer types. *Nucleic Acids Res* **48**, 2287–2302 (2020).
23. Rudd, P., Karlsson, N. & Khoo, K. Essentials of Glycobiology [Internet]. 4th edition. in *Essentials of Glycobiology [Internet]. 4th edition. Cold Spring Harbor (NY): Cold Spring Harbor Laboratory Press; 2022.* (2022).
24. Pinho, S. S. & Reis, C. A. Glycosylation in cancer: mechanisms and clinical implications. *Nat Rev Cancer* **15**, 540–555 (2015).
25. Kannagi, R. *et al.* Altered expression of glycan genes in cancers induced by epigenetic silencing and tumor hypoxia: Clues in the ongoing search for new tumor markers. *Cancer Sci* **101**, 586–593 (2010).
26. Johnson, J. L., Jones, M. B., Ryan, S. O. & Cobb, B. A. The regulatory power of glycans and their binding partners in immunity. *Trends Immunol* **34**, 290–298 (2013).

27. Amon, R., Reuven, E. M., Leviatan Ben-Arye, S. & Padler-Karavani, V. Glycans in immune recognition and response. *Carbohydr Res* **389**, 115–122 (2014).
28. Krämer, A., Green, J., Pollard, J. J. & Tugendreich, S. Causal analysis approaches in Ingenuity Pathway Analysis. *Bioinformatics* **30**, 523–530 (2014).
29. Cook Sangar, M. L. *et al.* Inhibition of CDK4/6 by Palbociclib Significantly Extends Survival in Medulloblastoma Patient-Derived Xenograft Mouse Models. *Clinical Cancer Research* **23**, 5802–5813 (2017).
30. Brown, P. D. *et al.* Memantine for the prevention of cognitive dysfunction in patients receiving whole-brain radiotherapy: a randomized, double-blind, placebo-controlled trial. *Neuro Oncol* **15**, 1429–1437 (2013).

REVIEWER COMMENTS

Reviewer #1 (Remarks to the Author):

The reviewer's comments have been fully addressed.

Reviewer #2 (Remarks to the Author):

In the revised manuscript, the authors have attempted to address some of the issues raised, however, the key concern regarding the reliability of their FFPE data for medulloblastoma subtype prediction remains unresolved.

Specifically, critique 3 requested the authors to present expression data on canonical markers for Group 3 and Group 4 tumors in their FFPE dataset. However, in their response, the authors did not provide evidence that these canonical markers can be robustly detected from their FFPE samples, while previous studies have shown they are readily detectable in fresh frozen tissues (e.g., Archer et al.). This raises concerns about whether the FFPE data can be used to draw reliable conclusions about subgroup identities.

The authors emphasize that "integration of proteome data of FFPE samples and fresh frozen samples is feasible" in their joint proteome cohort. However, if their FFPE data alone cannot recapitulate known subgroup-specific markers that have been defined in fresh frozen tissues, it remains unclear whether the FFPE dataset will be useful for clinical classification on its own.

Additionally, the proteomic markers used to define Group 3 and 4 tumors in Figure 2F have not been previously validated against transcriptomic or proteomic data from fresh frozen tissues. To prove their findings, the authors should perform immunohistochemical validation of these putative markers in bona fide Group 3 and 4 tumors defined by methylation profiling. Given the gene-specific correlation between RNA and protein levels (e.g. PMID: 19660143, PMID: 27951527), comparing the proteomic markers to established transcriptomic subgroup signatures would also help validate the FFPE findings. Without more rigorous validation, it will be difficult to assess whether the FFPE dataset alone can serve as a reliable tool for clinical subtype prediction. The lack of detectable signals from established subgroup markers raises concerns that the FFPE results may be flawed to some degree.

Moreover, the authors assigned SHH groups into pSHHs and pSHHt using a set of previously uncharacterized markers, which have not been validated with defined SHH subgroups. The clinical significance of these pSHHs and pSHHt subgroups has yet to be established. Such distinctions might arise from variations in the tissue sample compositions.

Overall, more rigorous validation is needed to determine whether the FFPE proteomic data can serve as an independent tool for clinically reliable subtyping, given the lack of expected subgroup-specific signals. The novel SHH split also requires further validation and evidence of clinical utility. These key concerns

should be addressed to support their claims.

Reviewer #4 (Remarks to the Author):

the authors have answered my queries

Point-by-point response to the reviewers' comments (Our answers are shown in blue colour)

Reviewer 1 and 3: No further concerns were raised.

Reviewer 2:

1. In the revised manuscript, the authors have attempted to address some of the issues raised, however, the key concern regarding the reliability of their FFPE data for medulloblastoma subtype prediction remains unresolved.

Answer: In order to meet the raised concern for human MB samples specifically, we now included new measurements of case matched fresh frozen human medulloblastoma samples (n=10, based on availability of material). We now directly compare proteome data of FFPE and FF tissue of the same MB case, showing that both profiles indeed reliably show the same molecular subtype of MB. We added a new Supplementary Figure 2 to the manuscript now showing these results: For the reviewer's convenience we have also attached the figure in this rebuttal letter (see also answer below to point 3, Figure 2 in this rebuttal). Additionally, the usage of proteome patterns from FFPE tissues has been also shown by other groups in the brain tumor context and in general a high correlation between proteome data from FF and FFPE samples from different tissues has been observed¹⁻⁴.

2. Specifically, critique 3 requested the authors to present expression data on canonical markers for Group 3 and Group 4 tumors in their FFPE dataset. However, in their response, the authors did not provide evidence that these canonical markers can be robustly detected from their FFPE samples, while previous studies have shown they are readily detectable in fresh frozen tissues (e.g., Archer et al.). This raises concerns about whether the FFPE data can be used to draw reliable conclusions about subgroup identities.

Answer: We agree with the reviewer that a general challenge of mass spectrometry data is the presence of missing values, as not all proteins can be detected in a sample or a measurement batch. Indeed, this implies a need for large integrated proteomic datasets. In the first revision, we had plotted the suggested canonical markers IMPG2, CRX, EOMES, SOX4 and BARHL1 among datasets. Of those, 4 markers were found in the Archer dataset (FF, however not significantly differential abundant), but none of them were found in the two other FF based datasets. Similarly, 4 markers were detected in the FFPE based datasets and BARHL1 and EOMES (TBR2) were indeed significantly higher abundant in group 4 MB (as expected). Therefore, we did not detect a specific bias relating to FF or FFPE material in relation to these canonical markers.

In order to further investigate the representation of canonical markers for MB in FF and FFPE based proteomic datasets, we researched the literature and set up a list of 80 previously described markers for the four main molecular subgroups of Medulloblastoma, as defined by gene expression profiling, Immunohistochemistry staining or proteomics (see table 1 below and for details on selected markers and respective publications see separately attached table 2 as an excel sheet -this list does not claim completeness but aims to give a broader impression of marker representation in MB). Of note, many markers have been established based on transcriptomics and do not necessarily reflect directly on proteome level (especially genes with unstable mRNAs and proteins (including mainly transcriptional

factors⁵). In order to account for potential differences between FF and FFPE based proteome analyses, we investigated each proteomic dataset individually (resulting in the analyses of 3 FF datasets (among them the dataset from Archer et al. mentioned by the reviewer) and our 3 FFPE datasets).

Tissue preservation	FF			FFPE		
Technique	TMT	SILAC	TMT	TMT	Label-free	Label-free
Study	Archer et al (2018)	Forget et al (2017)	Petralia et al (2020)	FFPE_MC (current study)	FFPE_TV	FFPE_BV
Detected	42/80 (52.5%)	6/80 (7.5%)	17/80 (21.25%)	14/80 (17.5%)	27/80 (33.75%)	33/80 (41.25%)
Detected and abundance $\geq \log_2FC(0.5)$ %	 45.23% Abundance as expected	 83.33% Abundance as expected	 64.7% Abundance as expected	 85.71% Abundance as expected	 44.44% Abundance as expected	 66.66% Abundance as expected

Table 1: Representation of canonical markers for medulloblastoma (MB) retrieved from the literature across studies (n=80, for details on markers please refer to table 2 as supplement to this rebuttal). FFPE_MC=FFPE main cohort, FFPE_TV= FFPE technical validation, FFPE_BV= FFPE biological validation

The number of biomarkers detected across datasets were quite similar. In the fresh frozen studies, the detected biomarkers ranged from 7.5% to 52.5% while in the FFPE studies, the detected biomarkers were in the range of 17.5% to 41.5%. A fraction of those markers behaved as expected from the literature (with respect to subtype specific protein abundance, $\log_2FC \geq 0.5$), independent of FF or FFPE tissue (range 45-83% for FF tissue and range 44 -56 % for FFPE tissue). Although the number of biomarkers identified in the Archer et al dataset was the highest, there was no major difference in the fraction of biomarkers with expected abundance profiles across studies. Finally, the number of quantified features does not necessarily define the discriminating quality of profiles and many studies specifically focus only on a subset of genes/proteins to subgroup MB^{6,7}.

In summary, our results indicate that FFPE tissue is reliable concerning analyses of proteome patterns in MB tissue, in general. However, our results also show that

1) protein patterns might differ from RNA patterns (e.g. due to posttranslational regulation), highlighting the additional need for proteome analyses and

2) technical variability and missing values are a general challenge of mass spectrometry-based proteome analyses (a known challenge that implies a need for large integrated datasets and that has been tackled also by our data integration approach using HarmonizR⁸).

To account for these results and the remark of the reviewer, we now start the discussion section the following way “Technical variability and missing values are a general challenge of mass spectrometry-based proteome analyses implying a need for large integrated datasets with reduction of technical biases.”.

3. The authors emphasize that "integration of proteome data of FFPE samples and fresh frozen samples is feasible" in their joint proteome cohort. However, if their FFPE data alone cannot recapitulate known subgroup-specific markers that have been defined in fresh frozen tissues, it remains unclear whether the FFPE dataset will be useful for clinical classification on its own.

Answer: We would like to thank the reviewer for their feedback and acknowledge the concerns raised. In addition to our answer to point 3, we want to emphasize that - following the reviewer's suggestion - we analyzed our main FFPE cohort and the 3 FF cohorts separately, showing a similar discrimination of MB subgroups in FFPE and FF datasets (see for the reviewer's convenience Figure 1A-B below and respective Supplementary Figure 3 in the manuscript).

Figure 1: A PCA on each individual proteome dataset with annotations of MB subtypes and the two main profiles – synaptic (pG3, pG4 and pSHHs) and transcriptional (pG3myc, pWNT, pSHHt) B Hierarchical clustering using Pearson correlation and ward.D2 linkage based on valid values in each dataset, separately. A similar separation of MB subgroups can be seen in FFPE and FF datasets.

Based on these observations, we decided to further integrate all the datasets. Additionally, we also validated all our findings in an independent FFPE dataset (biological validation, Figure 10 in the main manuscript) and in a FFPE technical validation dataset (Supplementary Figure 17).

Concerning the concerns of the representation of biomarkers in proteomic datasets we want to refer to our answer to point 2 and add that the lack of identifying single markers in mass spectrometry might be due to different reasons. The reviewer proposed the markers NRL, IMPG2 and CRX as marker for group 3 and EOMES, UNCX, LMX1A, SOX4, and BARHL1 as markers for Group 4 medulloblastoma based

on the following studies^{9,10} (PMID: 36131014, 36131015). These markers represent transcription factors which are generally low abundant and unstable on RNA and protein level⁵. and therefore they are hard to detect by mass spectrometry (independent of FF and FFPE preservation). Of note, in the study PMID 36131014, EOMES and CRX were investigated via Immunohistochemistry, whilst the identification of the biomarkers (EYS, GNB3, CRX, RPGRIP1, EOMES, BARHL1, DDX31, and LMX1A) was conducted via transcriptomics. As indicated above heterogeneity of proteome data (due to e.g. the protein extraction protocol, the technique used to label the peptides, the sensitivity of the mass spectrometer and the database search parameters and software) is a general challenge, we now address in the discussion. Additionally, its well-known in the literature that mRNAs and proteins correlate moderately and comparison between transcriptomics and proteomics is not straight forward^{5,11}. The papers mentioned by the reviewer in point 4 e.g. consider a correlation of 0.52 as significant and high, while other papers consider a correlation up to 0.6 as only moderate^{12,13}. Genes with stable mRNA and proteins (e.g. responsible for central metabolism, respiration and translation) show a higher correlation. In contrast, genes with unstable mRNAs and proteins (including mainly transcriptional factors) show lower correlation⁵.

Additionally, in order to specifically analyze the match of FF and FFPE proteome data in MB, we now generated proteome data of 10 FF MB samples that were also measured based on FFPE material (representing the subtypes pG4, pG3, pSHHt and pWNT, based on material availability). Figure 2 below shows that all FF samples on the Y-axis correlate highly with their respective FFPE sample (and samples of the same proteome subtype) on the X-axis. All FF samples grouped to the respective MB subtype as suggested by FFPE material before, and displayed expected defined biomarker profiles if detected (with the above mentioned and known challenge of missing values in proteome data (Figure 2B). This result has been added to the manuscript as Supplementary Figure 2.

Figure 2: A Corplot showing Pearson correlation between FF samples on Y-axis and their respective FFPE sample on the X-axis annotated for proteome subtype. * marks the highest match **B** Biomarkers for each proteome subtype (defined in the study). X depicts missing values.

4. Additionally, the proteomic markers used to define Group 3 and 4 tumors in Figure 2F have not been previously validated against transcriptomic or proteomic data from fresh frozen tissues. To prove their findings, the authors should perform immunohistochemical validation of these putative markers in bona fide Group 3 and 4 tumors defined by methylation profiling.

Given the gene-specific correlation between RNA and protein levels (e.g. PMID: 19660143, PMID: 27951527), comparing the proteomic markers to established transcriptomic subgroup signatures would also help validate the FFPE findings. Without more rigorous validation, it will be difficult to assess whether the FFPE dataset alone can serve as a reliable tool for clinical subtype prediction. The lack of detectable signals from established subgroup markers raises concerns that the FFPE results may be flawed to some degree.

Answer: We thank the reviewer for the proposal to further validate the suggested biomarkers. In addition to the analyses mentioned above, we again have researched the literature for every protein of the suggested 29 biomarkers in our manuscript (Figure 2F). Of those, GAB1, MICAL1 and FLNA (found highly abundant in pSHHt) are indeed known biomarkers for SHH MB^{14,15}. VSNL1 (found highly abundant in pSHHs) was also defined as biomarker for (SHH) Medulloblastoma with MBEN histology¹⁶. GNB3, PRDX1 and IGF2BP3 (found highly abundant in pG3) were already defined as biomarkers or mentioned in context of GR3 MBs^{9,17,18}. PLXNA2 (found highly abundant in pG4) was reported as a gene contributing to enriched Axonal Guidance signaling in GR4 MBs¹⁹. Finally, TPD52 (found highly abundant in the high risk pG3myc subtype) was defined as a biomarker for high risk-prediction non-WNT/non-SHH MBs, which is well in line with our results²⁰.

Interestingly we did see IGF2BP3 to be highly abundant in pG3 MB (see Figure 3A below and Figure 2F in the main manuscript). IGF2BP3 was previously described as a downstream effector gene of G3 MB¹⁸. IGF2BP3 together with CRX, OXT2 and MYC were defined as members of a super-enhancer driven transcriptional regulatory network¹⁸. Following the reviewer's suggestion, we performed additional immunohistochemical stainings on 3 cases per proteome MB subtype (based on material availability). We confirmed a strong positive staining with the IGF2BP3 antibody in pG3 MB, as expected (see Figure 3B and 3C below). We also observed that one pG3myc and one pWNT case also showed a positive signal (these two groups also showed a relatively higher abundance based on mass spectrometry in comparison to other subtypes (see also Figure 3A below). This could be due to the association of MYC with IGF2BP3 (which is highly expressed in WNT and pG3myc MB, see Figure 6K, Supplementary Figure 13A in the main manuscript). E.g. chromatin-immune precipitation assays have revealed that MYC binds to the promoter of IGF2BP3 and is further responsible for its high transcriptional activity²¹.

Figure 3: **A** IGF2BP3 protein abundance across six proteome subtypes based on mass spectrometry data and an estimation plot comparing IGF2BP3 abundance in pG3 vs other MB subtypes (pvalue < 0.001) **B** IGF2BP3 immuno histochemistry staining based digital quantification and an estimation plot comparing the digital histo score of IGF2BP3 of pG3 vs Others (pvalue < 0.01) **C** Representative images of immunohistochemical stainings against IGF2BP3. A strong signal was detected in all pG3 MB ($n_{pG3}=3$, $n_{pG3myc}=3$, $n_{pG4}=3$, $n_{pWNT}=3$, $n_{pSHHs}=3$, $n_{pSHHt}=3$).

We further validate pSHH subtypes using immunohistochemistry (see answer to point 5). Additionally, we have validated 3 other biomarkers via immunohistochemistry in our study, namely, TNC for pWNT MBs, ALDH1A3 for SHH MBs and PALMD for pG3myc MBs (see Figure 7, Supplementary Figure 11 and Figure 6, respectively). We further confirm MYC profiles using MYC antibody stainings (see Supplementary Figure 13 in the main manuscript). Taken together, we could confirm proteome abundance patterns with selected antibodies, but we agree with the reviewer that each biomarker has to be tested in prospective studies to verify it's potential for classification and potential therapy prediction in the future. We have added a respective phrase in the discussion section: "Further proposed markers for proteomic MB subtypes in this study have to be tested in prospective studies to verify their potential for classification and potential therapy prediction in the future".

Next, we thank the reviewer for the idea to correlate proteome biomarkers with established transcriptome profiles and used the defined groups from Cavalli et al.²². We have now plotted the transcripts of suggested proteome biomarkers based on defined transcriptome subtypes from Cavalli et al.²² (see Figure 4A below). All pSHHt-biomarkers were highly abundant in SHH α , SHH β and SHH δ . Of note SHH α consists of TP53-mutated SHH subtypes and a higher activation of transcription and DNA-repair pathways, matching nicely to the transcriptional/translational proteome profile. Some pSHHs

biomarkers were highly abundant in SHH γ , and this group is known to be made up of younger patients with MBEN histology, which well reflects the composition of our pSHHs group. Further, all the proposed pG4 biomarkers have similar abundance across the defined G4 transcriptome subtypes while pG3 biomarkers were highly abundant in G3 α and G3 β and pG3myc biomarkers were enriched in G3 γ - the group with worst prognosis and *MYC* amplifications, as expected (Figure 4A).

Figure 4: A Transcripts of biomarkers identified at proteome level plotted across known transcriptional defined subtypes (Cavalli et al.)⁵ **B** Transcripts of biomarkers identified at proteome level plotted across proteome subtypes **C** Heatmap showing Pearson correlation between the proteome subtypes using proteome and transcriptome data (n=57) in the main cohort and Pearson correlation between the Archer et al proteome subtypes using proteome and transcriptome data (n=38). X= missing value.

To further verify proteome data on transcriptome level we analyzed data of n=57 matched cases and could confirm that proteome patterns and biomarker patterns were generally well reflected in the transcriptome data (see Figure 4B and C in this rebuttal, and additionally Supplementary Figure 3C-F

in the manuscript). Following the reviewer's suggestion to check general correlation of proteome and transcriptome data for matched samples in our main cohort, we confirm the correlation patterns observed by Archer et al.²³ with highest correlation for pG3 and pSHHt while lowest correlation was observed for pG3myc and pG4 (mean Pearson correlation of 0.23 and 0.22 for the latter two).

5. Moreover, the authors assigned SHH groups into pSHHs and pSHHt using a set of previously uncharacterized markers, which have not been validated with defined SHH subgroups. The clinical significance of these pSHHs and pSHHt subgroups has yet to be established. Such distinctions might arise from variations in the tissue sample compositions.

Answer: We thank the reviewer for the feedback, we agree that the clinical significance of these pSHHs and pSHHt subgroups has yet to be established in the future. Indeed, we did not observe significant difference in survival in our cohort and the prognostic impact has to be further validated (see Figure 5I in the manuscript). We have now stated this in the discussion section in the following way: "The clinical significance of the two proteome subtypes of SHH MB needs further validation in the future.". Nevertheless, distinct proteome patterns may indicate distinct vulnerabilities therapy (as shown in the main Figure 3 and discussed in the manuscript). Such distinctions might, as rightfully stated by the reviewer, arise from variations in the tissue sample compositions. However, we would like to emphasize that these two proteome groups have already been described by the study of Archer et al, 2018²³ using fresh frozen tissue (labeled as SHHa and SHHb, which correspond to our pSHHt and pSHHs proteome subtypes, respectively). For FFPE tissue we also used microdissection to ensure high tumor content, and finally could confirm these 2 subtypes in the integrated proteome cohort. To account for the issue raised by the reviewer we now added a respective phrase in the discussion section "We detected two proteome SHH MB subtypes, namely pSHHs and pSHHt. While we cannot fully exclude the possibility that differences in proteome patterns could be due to variation in tissue composition, our results confirmed previously reported proteome patterns in SHH MB¹⁶".

To further validate results of mass spectrometry we used immunohistochemistry as suggested by the reviewer before. We investigated Ki67 stainings, which are commonly used to estimate the proliferation index in tumors. We hypothesized that pSHHt tumors (with higher abundance of proteins involved in DNA damage repair, mRNA splicing and cell-cycle related processes) show increased numbers of proliferating cells. Indeed, when comparing pSHHt to pSHHs MB, we detected a significantly higher percentage of Ki67 positive tumor cells (see Figure 5A-B below, which has been included in the manuscript as a new Supplementary Figure 11E-F). Thus, although the clinical significance needs further investigation, we can confirm that the biological differences are evident within the two proteome SHH subtypes.

Figure 5: A Representative image for Ki67 stainings in pSHHt and pSHHs MB. **B** pSHHt MBs show a significantly higher percentage of Ki67 positive cells compared to pSHHs MBs ($n_{\text{pSHHs}}=6$, $n_{\text{pSHHt}}=6$, $p_{\text{pSHHt_vs_pSHHs}} < 0.0001$, unpaired t-test)

Bibliography

1. Wiśniewski Jacek R. Proteomic Sample Preparation from Formalin Fixed and Paraffin Embedded Tissue. *J Vis Exp.* 2013; (79): 50589.
2. Mantsiou, A. *et al.* Proteomics Analysis of Formalin Fixed Paraffin Embedded Tissues in the Investigation of Prostate Cancer. *J Proteome Res* **19**, 2631–2642 (2020).
3. Leskoske, K. L. *et al.* Subgroup-Enriched Pathways and Kinase Signatures in Medulloblastoma Patient-Derived Xenografts. *J Proteome Res* **21**, 2124–2136 (2022).
4. Felix, M. *et al.* HIP1R and vimentin immunohistochemistry predict 1p/19q status in IDH-mutant glioma. *Neuro Oncol* **24**, 2121–2132 (2022).
5. Schwanhäusser, B. *et al.* Global quantification of mammalian gene expression control. *Nature* **473**, 337–342 (2011).
6. Northcott, P. A. *et al.* Rapid, reliable, and reproducible molecular sub-grouping of clinical medulloblastoma samples. *Acta Neuropathol* **123**, 615–626 (2012).
7. Schwalbe, E. C. *et al.* Rapid diagnosis of medulloblastoma molecular subgroups. *Clinical Cancer Research* **17**, 1883–1894 (2011).
8. Voß, H. *et al.* HarmonizR enables data harmonization across independent proteomic datasets with appropriate handling of missing values. *Nat Commun* **13**, 3523 (2022).
9. Smith, K. S. *et al.* Unified rhombic lip origins of group 3 and group 4 medulloblastoma. *Nature* **609**, 1012–1020 (2022).
10. Hendrikse, L. D. *et al.* Failure of human rhombic lip differentiation underlies medulloblastoma formation. *Nature* **609**, 1021–1028 (2022).

11. Koussounadis, A., Langdon, S. P., Um, I. H., Harrison, D. J. & Smith, V. A. Relationship between differentially expressed mRNA and mRNA-protein correlations in a xenograft model system. *Sci Rep* **5**, 10775 (2015).
12. Edfors, F. *et al.* Gene-specific correlation of RNA and protein levels in human cells and tissues. *Mol Syst Biol* **12**, 883 (2016).
13. Gry, M. *et al.* Correlations between RNA and protein expression profiles in 23 human cell lines. *BMC Genomics* **10**, 365 (2009).
14. Menyhárt, O. & Gyórfy, B. Principles of tumorigenesis and emerging molecular drivers of SHH-activated medulloblastomas. *Ann Clin Transl Neurol* **6**, 990–1005 (2019).
15. Ellison, D. W. *et al.* Medulloblastoma: clinicopathological correlates of SHH, WNT, and non-SHH/WNT molecular subgroups. *Acta Neuropathol* **121**, 381–396 (2011).
16. Korshunov, A. *et al.* Transcriptional profiling of medulloblastoma with extensive nodularity (MBEN) reveals two clinically relevant tumor subsets with VSNL1 as potent prognostic marker. *Acta Neuropathol* **139**, 583–596 (2020).
17. Sajesh, B. *et al.* MBRS-50. PEROXIREDOXIN1 IS A THERAPEUTIC TARGET IN GROUP-3 MEDULLOBLASTOMA. *Neuro Oncol* **20**, i139–i139 (2018).
18. Li, M. *et al.* Dissecting super-enhancer driven transcriptional dependencies reveals novel therapeutic strategies and targets for group 3 subtype medulloblastoma. *Journal of Experimental & Clinical Cancer Research* **41**, 311 (2022).
19. Hooper, C. M., Hawes, S. M., Kees, U. R., Gottardo, N. G. & Dallas, P. B. Gene expression analyses of the spatio-temporal relationships of human medulloblastoma subgroups during early human neurogenesis. *PLoS One* **9**, e112909 (2014).
20. Delaidelli, A. *et al.* Clinically Tractable Outcome Prediction of Non-WNT/Non-SHH Medulloblastoma Based on TPD52 IHC in a Multicohort Study. *Clinical Cancer Research* **28**, 116–128 (2022).
21. Du, M. *et al.* MYC-activated RNA N6-methyladenosine reader IGF2BP3 promotes cell proliferation and metastasis in nasopharyngeal carcinoma. *Cell Death Discov* **8**, 53 (2022).
22. Cavalli, F. M. G. *et al.* Intertumoral Heterogeneity within Medulloblastoma Subgroups. *Cancer Cell* **31**, 737–754.e6 (2017).
23. Archer, T. C. *et al.* Proteomics, Post-translational Modifications, and Integrative Analyses Reveal Molecular Heterogeneity within Medulloblastoma Subgroups. *Cancer Cell* **34**, 396–410.e8 (2018).

REVIEWERS' COMMENTS

Reviewer #2 (Remarks to the Author):

The authors have addressed my concerns adequately. I recommend it for publication.